# What Distributions are Robust to Indiscriminate Poisoning Attacks for Linear Learners?

**Fnu Suya**[1]    **Xiao Zhang**[2]    **Yuan Tian**[3]    **David Evans**[1]
[1]University of Virginia    [2]CISPA Helmholtz Center for Information Security
[3]University California Los Angeles
suya@virginia.edu, xiao.zhang@cispa.de, yuant@ucla.edu, evans@virginia.edu

## Abstract

We study indiscriminate poisoning for linear learners where an adversary injects a few crafted examples into the training data with the goal of forcing the induced model to incur higher test error. Inspired by the observation that linear learners on some datasets are able to resist the best known attacks even without any defenses, we further investigate whether datasets can be inherently robust to indiscriminate poisoning attacks for linear learners. For theoretical Gaussian distributions, we rigorously characterize the behavior of an optimal poisoning attack, defined as the poisoning strategy that attains the maximum risk of the induced model at a given poisoning budget. Our results prove that linear learners can indeed be robust to indiscriminate poisoning if the class-wise data distributions are well-separated with low variance and the size of the constraint set containing all permissible poisoning points is also small. These findings largely explain the drastic variation in empirical attack performance of the state-of-the-art poisoning attacks on linear learners across benchmark datasets, making an important initial step towards understanding the underlying reasons some learning tasks are vulnerable to data poisoning attacks.

## 1 Introduction

Machine learning models, especially current large-scale models, require large amounts of labeled training data, which are often collected from untrusted third parties [6]. Training models on these potentially malicious data poses security risks. A typical application is in spam filtering, where the spam detector is trained using data (i.e., emails) that are generated by users with labels provided often implicitly by user actions. In this setting, spammers can generate spam messages that inject benign words likely to occur in spam emails such that models trained on these spam messages will incur significant drops in filtering accuracy as benign and malicious messages become indistinguishable [37, 20]. These kinds of attacks are known as *poisoning attacks*. In a poisoning attack, the attacker injects a relatively small number of crafted examples into the original training set such that the resulting trained model (known as the *poisoned model*) performs in a way that satisfies certain attacker goals.

One commonly studied poisoning attacks in the literature are *indiscriminate poisoning attacks* [4, 54, 35, 45, 5, 46, 25, 31, 10], in which the attackers aim to let induced models incur larger test errors compared to the model trained on a clean dataset. Other poisoning goals, including targeted [42, 56, 24, 21, 18] and subpopulation [22, 46] attacks, are also worth studying and may correspond to more realistic attack goals. We focus on indiscriminate poisoning attacks as these attacks interfere with the fundamental statistical properties of the learning algorithm [45, 25], but include a summary of prior work on understanding limits of poisoning attacks in other settings in the related work section.

Indiscriminate poisoning attack methods have been developed that achieve empirically strong poisoning attacks in many settings [45, 46, 25, 31], but these works do not explain why the proposed attacks are sometimes ineffective. In addition, the evaluations of these attacks can be deficient in

some aspects [3, 4, 45, 5, 25, 31] (see Section 3) and hence, may not be able to provide an accurate picture on the current progress of indiscriminate poisoning attacks on linear models. The goal of our work is to understand the properties of the learning tasks that prevent effective attacks under linear models. An attack is considered effective if the increased error (or risk in the distributional sense) from poisoning exceeds the injected poisoning ratio [31, 25].

In this paper, we consider indiscriminate data poisoning attacks for linear models, the commonly studied victim models in the literature [3, 4, 24, 45, 10]. Attacks on linear models are also studied very recently [46, 25, 5, 9] and we limit our scope to linear models because attacks on the simplest linear models are still not well understood, despite extensive prior empirical work in this setting. Linear models continue to garner significant interest due to their simplicity and high interpretability in explaining predictions [30, 39]. Linear models also achieve competitive performance in many security-critical applications for which poisoning is relevant, including training with differential privacy [49], recommendation systems [15] and malware detection [7, 8, 41, 11, 44, 2]. From a practical perspective, linear models continue to be relevant—for example, Amazon SageMaker [1], a scalable framework to train ML models intended for developers and business analysts, provides linear models for tabular data, and trains linear models (on top of pretrained feature extractors) for images.

**Contributions.** We show that state-of-the-art poisoning strategies for linear models all have similar attack effectiveness of each dataset, whereas their performance varies significantly across different datasets (Section 3). All tested attacks are very effective on benchmark datasets such as Dogfish and Enron, while none of them are effective on other datasets, such as selected MNIST digit pairs (e.g., 6–9) and Adult, even when the victim does not employ any defenses (Figure 1). To understand whether this observation means such datasets are inherently robust to poisoning attacks or just that state-of-the-art attacks are suboptimal, we first introduce general definitions of optimal poisoning attacks for both finite-sample and distributional settings (Definitions 4.1 and 4.2). Then, we prove that under certain regularity conditions, the performance achieved by an optimal poisoning adversary with finite-samples converges asymptotically to the actual optimum with respect to the underlying distribution (Theorem 4.3), and the best poisoning performance is always achieved at the maximum allowable poisoning ratio under mild conditions (Theorem 4.5).

Building upon these definitions, we rigorously characterize the behavior of optimal poisoning attacks under a theoretical Gaussian mixture model (Theorem 5.3), and derive upper bounds on their effectiveness for general data distributions (Theorem 5.7). In particular, we discover that a larger projected constraint size (Definition 5.5) is associated with a higher inherent vulnerability, whereas projected data distributions with a larger separability and smaller standard deviation (Definition 5.6) are fundamentally less vulnerable to poisoning attacks (Section 5.2). Empirically, we find the discovered learning task properties and the gained theoretical insights largely explain the drastic difference in attack performance observed for state-of-the-art indiscriminate poisoning attacks on linear models across benchmark datasets (Section 6). Finally, we discuss the potential implications of our work by showing how one might improve robustness to poisoning via better feature transformations and defenses (e.g., data sanitization defenses) to limit the impact of poisoning points (Section 7).

**Related Work.** Several prior works developed indiscriminate poisoning attacks by *injecting* a small fraction of poisoning points. One line of research adopts iterative gradient-based methods to directly maximize the surrogate loss chosen by the victim [3, 35, 34, 24], leveraging the idea of influence functions [40]. Another approach bases attacks on convex optimization methods [45, 46, 25], which provide a more efficient way to generate poisoned data, often with an additional input of a target model. Most of these works focus on studying linear models, but recently there has been some progress on designing more effective attacks against neural networks with insights learned from attacks evaluated on linear models [31, 32]. All the aforementioned works focus on developing different indiscriminate poisoning algorithms and some also characterize the hardness of poisoning in the model-targeted setting [46, 32], but did not explain why certain datasets are seemingly harder to poison than others. Our work leverages these attacks to empirically estimate the inherent vulnerabilities of benchmark datasets to poisoning, but focuses on providing explanations for the disparate poisoning vulnerability across the datasets. Besides injection, some other works consider different poisoning settings from ours by modifying up to the whole training data, also known as unlearnable examples [19, 55, 16] and some of them, similar to us, first rigorously analyze poisoning on theoretical distributions and then generalize the insights to general distributions [48, 47].

Although much research focuses on indiscriminate poisoning, many realistic attack goals are better captured as *targeted* attacks [42, 56, 24, 21, 18], where the adversary's goal is to induce a model that misclassifies a particular known instance, or *subpopulation attacks* [22, 46], where the adversary's goal is to produce misclassifications for a defined subset of the distribution. A recent work that studies the inherent vulnerabilities of datasets to targeted data poisoning attacks proposed the Lethal Dose Conjecture (LDC) [50]: given a dataset of size $N$, the tolerable amount of poisoning points from any targeted poisoning attack generated through insertion, deletion or modifications is $\Theta(N/n)$, where $n$ is the sample complexity of the most data-efficient learner trained on the clean data to correctly predict a known test sample. Compared to our work, LDC is more general and applies to any dataset, any learning algorithm, and even different poisoning settings (e.g., deletion, insertion). In contrast, our work focuses on insertion-only indiscriminate attacks for linear models. However, the general setting for LDC can result in overly pessimistic estimates of the power of insertion-only indiscriminate poisoning attacks. In addition, the key factor of the sample complexity $n$ in LDC is usually unknown and difficult to determine. Our work complements LDC by making an initial step towards finding factors (which could be related to $n$) under a particular attack scenario to better understand the power of indiscriminate data poisoning attacks. Appendix E provides more details.

## 2 Preliminaries

We consider binary classification tasks. Let $\mathcal{X} \subseteq \mathbb{R}^n$ be the input space and $\mathcal{Y} = \{-1, +1\}$ be the label space. Let $\mu_c$ be the joint distribution of clean inputs and labels. For standard classification tasks, the goal is to learn a hypothesis $h : \mathcal{X} \rightarrow \mathcal{Y}$ that minimizes $\mathrm{Risk}(h; \mu_c) = \mathbb{P}_{(\boldsymbol{x}, y) \sim \mu_c}[h(\boldsymbol{x}) \neq y]$. Instead of directly minimizing risk, typical machine learning methods find an approximately good hypothesis $h$ by restricting the search space to a specific hypothesis class $\mathcal{H}$, then optimizing $h$ by minimizing some convex surrogate loss: $\min_{h \in \mathcal{H}} L(h; \mu_c)$. In practical applications with only a finite number of samples, model training replaces the population measure $\mu_c$ with its empirical counterpart. The surrogate loss for $h$ is defined as $L(h; \mu) = \mathbb{E}_{(\boldsymbol{x}, y) \sim \mu}[\ell(h; \boldsymbol{x}, y)]$, where $\ell(h; \boldsymbol{x}, y)$ denotes the non-negative individual loss of $h$ incurred at $(\boldsymbol{x}, y)$.

We focus on the linear hypothesis class and study the commonly considered hinge loss in prior poisoning literature [3, 4, 45, 25, 46] when deriving the optimal attacks for 1-D Gaussian distributions in Section 5.1. Our results can be extended to other linear methods such as logistic regression (LR). A *linear hypothesis* parameterized by a weight parameter $\boldsymbol{w} \in \mathbb{R}^n$ and a bias parameter $b \in \mathbb{R}$ is defined as: $h_{\boldsymbol{w}, b}(\boldsymbol{x}) = \mathrm{sgn}(\boldsymbol{w}^\top \boldsymbol{x} + b)$ for any $\boldsymbol{x} \in \mathbb{R}^n$, where $\mathrm{sgn}(\cdot)$ denotes the sign function. For any $\boldsymbol{x} \in \mathcal{X}$ and $y \in \mathcal{Y}$, the *hinge loss* of a linear classifier $h_{\boldsymbol{w}, b}$ is defined as:

$$\ell(h_{\boldsymbol{w}, b}; \boldsymbol{x}, y) = \max\{0, 1 - y(\boldsymbol{w}^\top \boldsymbol{x} + b)\} + \frac{\lambda}{2} \|\boldsymbol{w}\|_2^2, \tag{1}$$

where $\lambda \geq 0$ is the tuning parameter which penalizes the $\ell_2$-norm of the weight parameter $\boldsymbol{w}$.

**Threat Model.** We formulate indiscriminate data poisoning attack as a game between an attacker and a victim in practice following the prior work [45]:

1. A clean training dataset $\mathcal{S}_c$ is produced, where each data point is i.i.d. sampled from $\mu_c$.
2. The attacker generates a poisoned dataset $\mathcal{S}_p$ using some poisoning strategy $\mathcal{A}$, which aims to reduce the performance of the victim model by injecting $\mathcal{S}_p$ into the training dataset.
3. The victim minimizes empirical surrogate loss $L(\cdot)$ on $\mathcal{S}_c \cup \mathcal{S}_p$ and produces a model $\hat{h}_p$.

The attacker's goal is to find a poisoning strategy $\mathcal{A}$ such that the risk of the final induced classifier $\mathrm{Risk}(\hat{h}_p; \mu_c)$ is as high as possible, which is empirically estimated on a set of fresh testing data that are i.i.d. sampled from $\mu_c$. We assume the attacker has full knowledge of the learning process, including the clean distribution $\mu_c$ or the clean training dataset $\mathcal{S}_c$, the hypothesis class $\mathcal{H}$, the surrogate loss function $\ell$ and the learning algorithm adopted by the victim.

We impose two restrictions to the poisoning attack: $|\mathcal{S}_p| \leq \epsilon \cdot |\mathcal{S}_c|$ and $\mathcal{S}_p \subseteq \mathcal{C}$, where $\epsilon \in [0, 1]$ is the poisoning budget and $\mathcal{C} \subseteq \mathcal{X} \times \mathcal{Y}$ is a bounded subset that captures the feasibility constraints for poisoned data. We assume that $\mathcal{C}$ is specified in advance with respect to different applications (e.g., normalized pixel values of images can only be in range $[0, 1]$) and possible defenses the victim may choose (e.g., points that have larger Euclidean distance from center will be removed) [45, 25]. Here, we focus on undefended victim models, i.e., $\mathcal{C}$ is specified based on application constraints, so

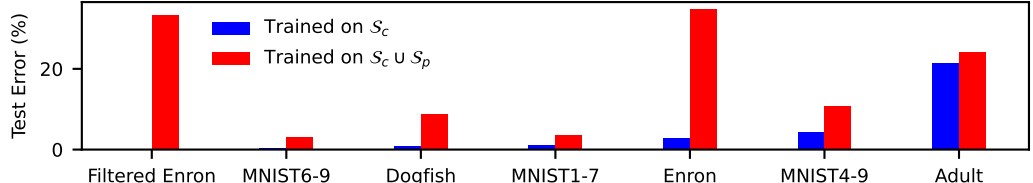

Figure 1: Performance of best current indiscriminate poisoning attacks with $\epsilon = 3\%$ across different benchmark datasets on linear SVM. Datasets are sorted from lowest to highest base error rate.

as to better assess the inherent dataset vulnerabilities without active protections. However, defense strategies such as data sanitization [12, 45, 25] may shrink the size of $\mathcal{C}$ so that the poisoned data are less extreme and harmful. We provide preliminary experimental results on this in Section 7.

## 3 Disparate Poisoning Vulnerability of Benchmark Datasets

Prior evaluations of poisoning attacks on *convex models* are inadequate in some aspects, either being tested on very small datasets (e.g., significantly subsampled MNIST 1–7 dataset) without competing baselines [3, 10, 34, 35], generating invalid poisoning points [45, 25] or lacking diversity in the evaluated convex models/datasets [31, 32]. This motivates us to carefully evaluate representative attacks for linear models on various benchmark datasets without considering additional defenses.

**Experimental Setup.** We evaluate the state-of-the-art data poisoning attacks for linear models: *Influence Attack* [24, 25], *KKT Attack* [25], *Min-Max Attack* [45, 25], and *Model-Targeted Attack (MTA)* [46]. We consider both linear SVM and LR models and evaluate the models on benchmark datasets including different MNIST [28] digit pairs (MNIST 1–7, as used in prior evaluations [45, 25, 3, 46], in addition to MNIST 4–9 and MNIST 6–9 which were picked to represent datasets that are relatively easier/harder to poison), and other benchmark datasets used in prior evaluations including Dogfish [24], Enron [36] and Adult [22, 46]. *Filtered Enron* is obtained by filtering out 3% of near boundary points (with respect to the clean decision boundary of linear SVM) from Enron. The motivation of this dataset (along with Adult) is to show that, the initial error cannot be used to trivially infer the vulnerability to poisoning attacks (Section 6). We choose 3% as the poisoning rate following previous works [45, 25, 31, 32]. Appendix D.1 provides details on the experimental setup.

**Results.** Figure 1 shows the highest error from the tested poisoning attacks (they perform similarly in most cases) on linear SVM. Similar observations are found on LR in Appendix D.2.5. At 3% poisoning ratio, the (absolute) increased test errors of datasets such as MNIST 6–9 and MNIST 1–7 are less than 4% for SVM while for other datasets such as Dogfish, Enron and Filtered Enron, the increased error is much higher than the poisoning ratio, indicating that these datasets are more vulnerable to poisoning. Dogfish is moderately vulnerable ($\approx 9\%$ increased error) while Enron and Filtered Enron are highly vulnerable with over 30% of increased error. Consistent with prior work [45, 25, 32], throughout this paper, we measure the increased error to determine whether a dataset is vulnerable to poisoning attacks. However, in security-critical applications, the ratio between the increased error and the initial error might matter more but leave its exploration as future work. These results reveal a drastic difference in robustness of benchmark datasets to state-of-the-art indiscriminate poisoning attacks which was not explained in prior works. A natural question to ask from this observation is *are datasets like MNIST digits inherently robust to poisoning attacks or just resilient to current attacks?* Since estimating the performance of optimal poisoning attacks for benchmark datasets is very challenging, we first characterize optimal poisoning attacks for theoretical distributions and then study their partial characteristics for general distributions in Section 5.

## 4 Defining Optimal Poisoning Attacks

In this section, we lay out formal definitions of optimal poisoning attacks and study their general implications. First, we introduce a notion of *finite-sample optimal poisoning* to formally define the optimal poisoning attack in the practical finite-sample setting with respect to our threat model:

**Definition 4.1** (Finite-Sample Optimal Poisoning). Consider input space $\mathcal{X}$ and label space $\mathcal{Y}$. Let $\mu_c$ be the underlying data distribution of clean inputs and labels. Let $\mathcal{S}_c$ be a set of examples sampled i.i.d. from $\mu_c$. Suppose $\mathcal{H}$ is the hypothesis class and $\ell$ is the surrogate loss function that are used for learning. For any $\epsilon \geq 0$ and $\mathcal{C} \subseteq \mathcal{X} \times \mathcal{Y}$, a *finite-sample optimal poisoning adversary* $\hat{\mathcal{A}}_{\mathrm{opt}}$ is defined to be able to generate some poisoned dataset $\mathcal{S}_p^*$ such that:

$$\mathcal{S}_p^* = \operatorname*{argmax}_{\mathcal{S}_p} \ \mathrm{Risk}(\hat{h}_p; \mu_c) \quad \text{s.t. } \mathcal{S}_p \subseteq \mathcal{C} \text{ and } |\mathcal{S}_p| \leq \epsilon \cdot |\mathcal{S}_c|,$$

where $\hat{h}_p = \operatorname{argmin}_{h \in \mathcal{H}} \sum_{(\boldsymbol{x},y) \in \mathcal{S}_c \cup \mathcal{S}_p} \ell(h; \boldsymbol{x}, y)$ denotes the empirical loss minimizer.

Definition 4.1 states that no poisoning strategy can achieve a better attack performance than that achieved by $\hat{\mathcal{A}}_{\mathrm{opt}}$. If we denote by $\hat{h}_p^*$ the hypothesis produced by minimizing the empirical loss on $\mathcal{S}_c \cup \mathcal{S}_p^*$, then $\mathrm{Risk}(\hat{h}_p^*; \mu_c)$ can be regarded as the maximum achievable attack performance.

Next, we introduce a more theoretical notion of *distributional optimal poisoning*, which generalizes Definition 4.1 from finte-sample datasets to data distributions.

**Definition 4.2** (Distributional Optimal Poisoning). Consider the same setting as in Definition 4.1. A *distributional optimal poisoning adversary* $\mathcal{A}_{\mathrm{opt}}$ is defined to be able to generate some poisoned data distribution $\mu_p^*$ such that:

$$(\mu_p^*, \delta^*) = \operatorname*{argmax}_{(\mu_p, \delta)} \ \mathrm{Risk}(h_p; \mu_c) \quad \text{s.t. } \mathrm{supp}(\mu_p) \subseteq \mathcal{C} \text{ and } 0 \leq \delta \leq \epsilon,$$

where $h_p = \operatorname{argmin}_{h \in \mathcal{H}} \left\{ L(h; \mu_c) + \delta \cdot L(h; \mu_p) \right\}$ denotes the population loss minimizer.

Similar to the finite-sample case, Definition 4.2 implies that there is no feasible poisoned distribution $\mu_p$ such that the risk of its induced hypothesis is higher than that attained by $\mu_p^*$. Theorem 4.3, proven in Appendix A.1, connects Definition 4.1 and Definition 4.2. The formal definitions of uniform convergence, strong convexity and Lipschitz continuity are given in Appendix A.1.

**Theorem 4.3.** *Consider the same settings as in Definitions 4.1 and 4.2. Suppose $\mathcal{H}$ satisfies the uniform convergence property with function $m_{\mathcal{H}}(\cdot, \cdot)$. Assume $\ell$ is b-strongly convex and $\mathrm{Risk}(h; \mu_c)$ is $\rho$-Lipschitz continuous with respect to model parameters for some $b, \rho > 0$. Let $\hat{h}_p^* = \operatorname{argmin}_{h \in \mathcal{H}} \sum_{(\boldsymbol{x},y) \in \mathcal{S}_c \cup \mathcal{S}_p^*} \ell(h; \boldsymbol{x}, y)$ and $h_p^* = \operatorname{argmin}_{h \in \mathcal{H}} \{ L(h; \mu_c) + \delta^* \cdot L(h; \mu_p^*) \}$. For any $\epsilon', \delta' \in (0, 1)$, if $|\mathcal{S}_c| \geq m_{\mathcal{H}}(\epsilon', \delta')$, then with probability at least $1 - \delta'$,*

$$\left| \mathrm{Risk}(\hat{h}_p^*; \mu_c) - \mathrm{Risk}(h_p^*; \mu_c) \right| \leq 2\rho \sqrt{\frac{\epsilon'}{b}}.$$

*Remark* 4.4. Theorem 4.3 assumes three regularity conditions to ensure the finite-sample optimal poisoning attack is a consistent estimator of the distributional optimal one (i.e., insights on poisoning from distributional settings can transfer to finite-sample settings): the uniform convergence property of $\mathcal{H}$ that guarantees empirical minimization of surrogate loss returns a good hypothesis, the strong convexity condition that ensures a unique loss minimizer, and the Lipschitz condition that translates the closeness of model parameters to the closeness of risk. These conditions hold for most (properly regularized) convex problems and input distributions with bounded densities. The asymptotic convergence rate is determined by the function $m_{\mathcal{H}}$, which depends on the complexity of the hypothesis class $\mathcal{H}$ and the surrogate loss $\ell$. For instance, if we choose $\lambda$ carefully, sample complexity of the linear hypothesis class for a bounded hinge loss is $\Omega(1/(\epsilon')^2)$, where $\epsilon'$ is the error bound parameter for specifying the uniform convergence property (see Definition A.3) and other problem-dependent parameters are hidden in the big-$\Omega$ notation (see Section 15 of [43] for details). We note the generalization of optimal poisoning attack for linear case is related to agnostic learning of halfspaces [23], which also imposes assumptions on the underlying distribution such as anti-concentration assumption [13, 17] similar to the Lipschitz continuity condition assumed in Theorem 4.3. The theorem applies to any loss function that is strongly convex (e.g., general convex losses with the $\ell_2$- or elastic net regularizers), not just the hinge loss.

Moreover, we note that $\delta^*$ represents the ratio of injected poisoned data that achieves the optimal attack performance. In general, $\delta^*$ can be any value in $[0, \epsilon]$, but we show in Theorem 4.5, proven in Appendix A.2, that optimal poisoning can always be achieved with $\epsilon$-poisoning under mild conditions.

**Theorem 4.5.** *The optimal poisoning attack performance defined in Definition 4.2 can always be achieved by choosing $\epsilon$ as the poisoning ratio, if either of the following conditions is satisfied:*

1. *The support of the clean distribution* $\mathrm{supp}(\mu_c) \subseteq \mathcal{C}$.
2. *$\mathcal{H}$ is a convex hypothesis class, and for any $h_\theta \in \mathcal{H}$, there always exists a distribution $\mu$ such that $\mathrm{supp}(\mu) \subseteq \mathcal{C}$ and $\frac{\partial}{\partial \theta} L(h_\theta; \mu) = \mathbf{0}$.*

*Remark* 4.6. Theorem 4.5 characterizes the conditions under which the optimal performance is guaranteed to be achieved with the maximum poisoning ratio $\epsilon$. Note that the first condition $\mathrm{supp}(\mu_c) \subseteq \mathcal{C}$ is mild because it typically holds for poisoning attacks against undefended classifiers. When attacking classifiers that employ some defenses such as data sanitization, the condition $\mathrm{supp}(\mu_c) \subseteq \mathcal{C}$ might not hold, due to the fact that the proposed defense may falsely reject some clean data points as outliers (i.e., related to false positive rates). The second condition complements the first one in that it does not require the victim model to be undefended, however, it requires $\mathcal{H}$ to be convex. We prove in Appendix A.3 that for linear hypothesis with hinge loss, such a $\mu$ can be easily constructed. The theorem enables us to conveniently characterize the optimal poisoning attacks in Section 5.1 by directly using $\epsilon$. When the required conditions are satisfied, this theorem also provides a simple sanity check on whether a poisoning attack is optimal. In particular, if a candidate attack is optimal, the risk of the induced model is monotonically non-decreasing with respect to the poisoning ratio. Note that the theorem above applies to any strongly convex loss function.

# 5 Characterizing Optimal Poisoning Attacks

This section characterizes the distributional optimal poisoning attacks with respect to linear hypothesis class. We first consider a theoretical 1-dimensional Gaussian mixture model and exactly characterize optimal poisoning attack and then discuss the implications on the underlying factors that potentially cause the inherent vulnerabilities to poisoning attacks for general high-dimensional distributions.

## 5.1 One-Dimensional Gaussian Mixtures

Consider binary classification tasks using *hinge* loss with one-dimensional inputs, where $\mathcal{X} = \mathbb{R}$ and $\mathcal{Y} = \{-1, +1\}$. Let $\mu_c$ be the underlying clean data distribution, where each example $(x, y)$ is assumed to be i.i.d. sampled according to the following Gaussian mixture model:

$$\begin{cases} y = -1, x \sim \mathcal{N}(\gamma_1, \sigma_1^2) & \text{with probability } p, \\ y = +1, x \sim \mathcal{N}(\gamma_2, \sigma_2^2) & \text{with probability } 1 - p, \end{cases} \tag{2}$$

where $\sigma_1, \sigma_2 > 0$ and $p \in (0, 1)$. Without loss of generality, we assume $\gamma_1 \leq \gamma_2$. Following our threat model, we let $\epsilon \geq 0$ be the maximum poisoning ratio and $\mathcal{C} := [-u, u] \times \mathcal{Y}$ for some $u > 0$ be the constraint set. Let $\mathcal{H}_\mathrm{L} = \{h_{w,b} : w \in \{-1, 1\}, b \in \mathbb{R}\}$ be the linear hypothesis class with normalized weights. Note that we consider a simplified setting where the weight parameter $w \in \{-1, 1\}$.[1] Since $\|w\|_2$ is fixed, we also set $\lambda = 0$ in the hinge loss function (1).

For clarify in presentation, we outline the connection between the definitions and the main theorem below, which will also show the high-level proof sketch of Theorem 5.3. We first prove that in order to understand the optimal poisoning attacks, it is sufficient to study the family of two-point distributions (Definition 5.1), instead of any possible distributions, as the poisoned data distribution. Based on this reduction and a specification of weight flipping condition (Definition 5.2), we then rigorously characterize the optimal attack performance with respect to different configurations of task-related parameters (Theorem 5.3). Similar conclusions may also be made with logistic loss.

**Definition 5.1** (Two-point Distribution). For any $\alpha \in [0, 1]$, $\nu_\alpha$ is defined as a *two-point distribution*, if for any $(x, y)$ sampled from $\nu_\alpha$,

$$(x, y) = \begin{cases} (-u, +1) & \text{with probability } \alpha, \\ (u, -1) & \text{with probability } 1 - \alpha. \end{cases} \tag{3}$$

---

[1]Characterizing the optimal poisoning attack under the general setting of $w \in \mathbb{R}$ is more challenging, since we need to consider the effect of any possible choice of $w$ and its interplay with the dataset and constraint set factors. We leave the theoretical analyses of $w \in \mathbb{R}$ to future work.

**Definition 5.2** (Weight-Flipping Condition). Consider the assumed Gaussian mixture model (2) and the linear hypothesis class $\mathcal{H}_L$. Let $g$ be an auxiliary function such that for any $b \in \mathbb{R}$,

$$g(b) = \frac{1}{2}\Phi\left(\frac{b + \gamma_1 + 1}{\sigma}\right) - \frac{1}{2}\Phi\left(\frac{-b - \gamma_2 + 1}{\sigma}\right),$$

where $\Phi$ is the cumulative distribution function (CDF) of standard Gaussian $\mathcal{N}(0, 1)$. Let $\epsilon > 0$ be the poisoning budget and $g^{-1}$ be the inverse of $g$, then the *weight-flipping condition* is defined as:

$$\max\{\Delta(-\epsilon), \Delta(g(0)), \Delta(\epsilon)\} \geq 0, \tag{4}$$

where $\Delta(s) = L(h_{1,g^{-1}(s)}; \mu_c) - \min_{b \in \mathbb{R}} L(h_{-1,b}; \mu_c) + \epsilon \cdot (1 + u) - s \cdot g^{-1}(s)$.

Now we are ready to present our main theoretical results. The following theorem rigorously characterizes the behavior of the distributional optimal poisoning adversary $\mathcal{A}_{\text{opt}}$ under the Gaussian mixture model (2) and the corresponding optimal attack performance:

**Theorem 5.3.** *Suppose the clean distribution $\mu_c$ follows the Gaussian mixture model* (2) *with $p = 1/2$, $\gamma_1 \leq \gamma_2$, and $\sigma_1 = \sigma_2 = \sigma$. Assume $u \geq 1$ and $|\gamma_1 + \gamma_2| \leq 2(u - 1)$. There always exists some $\alpha \in [0, 1]$ such that the optimal attack performance defined in Definition 4.2 is achieved with $\delta = \epsilon$ and $\mu_p = \nu_\alpha$, where $\nu_\alpha$ is defined by* (3). *More specifically, if $h_p^* = \operatorname{argmin}_{h \in \mathcal{H}_L}\{L(h; \mu_c) + \epsilon \cdot L(h; \nu_\alpha)\}$ denotes the induced hypothesis with optimal poisoning, then*

$$\text{Risk}(h_p^*; \mu_c) = \begin{cases} \Phi\left(\frac{\gamma_2 - \gamma_1}{2\sigma}\right) & \text{if condition (4) is satisfied,} \\ \frac{1}{2}\Phi\left(\frac{\gamma_1 - \gamma_2 + 2s}{2\sigma}\right) + \frac{1}{2}\Phi\left(\frac{\gamma_1 - \gamma_2 - 2s}{2\sigma}\right) & \text{otherwise,} \end{cases}$$

*where $s = \max\{g^{-1}(\epsilon) - g^{-1}(0), g^{-1}(0) - g^{-1}(-\epsilon)\}$ and $g(\cdot)$ is defined in Definition 5.2.*

*Remark* 5.4. The proof of Theorem 5.3 is given in Appendix B.1. We considered same variance of $\sigma_1 = \sigma_2 = \sigma$ simply for the convenience in proof without involving minor but tedious calculations, and the same conclusion can be made using different variances. Theorem 5.3 characterizes the exact behavior of $\mathcal{A}_{\text{opt}}$ for typical combinations of hyperparameters under the considered model, including distribution related parameters such as $\gamma_1$, $\gamma_2$, $\sigma$ and poisoning related parameters such as $\epsilon$, $u$. We use $\epsilon$ directly because the poisoning setup in the 1-D Gaussian distribution satisfies the second condition in Theorem 4.5. Hence, the best performance can be achieved with the maximum poisoning ratio $\epsilon$. For $u$, a larger of it suggests the weight-flipping condition (4) is more likely to be satisfied, as an attacker can generate poisoned data with larger hinge loss to flip the weight parameter $\boldsymbol{w}$. Class separability $|\gamma_1 - \gamma_2|$ and within-class variance $\sigma$ also play an important role in affecting the optimal attack performance. If the ratio $|\gamma_1 - \gamma_2|/\sigma$ is large, then we know the initial risk $\text{Risk}(h_c; \mu_c) = \Phi\left(\frac{\gamma_1 - \gamma_2}{2\sigma}\right)$ will be small. Consider the case where condition (4) is satisfied. Note that $\Phi\left(\frac{\gamma_2 - \gamma_1}{2\sigma}\right) = 1 - \Phi\left(\frac{\gamma_1 - \gamma_2}{2\sigma}\right)$ implies an improved performance of optimal poisoning attack thus an higher inherent vulnerabilities to data poisoning attacks. However, it is worth noting that there is an implicit assumption in condition (4) that the weight parameter can be flipped from $w = 1$ to $w = -1$. A large value of $|\gamma_1 - \gamma_2|/\sigma$ also implies that flipping the weight parameter becomes more difficult, since the gap between the hinge loss with respect to $\mu_c$ for a hypothesis with $w = -1$ and that with $w = 1$ becomes larger. Moreover, if condition (4) cannot be satisfied, then a larger ratio of $|\gamma_1 - \gamma_2|/\sigma$ suggests that it is more difficult to move the decision boundary to incur an increase in test error, because of the number of correctly classified boundary points will increase at a faster rate. In summary, Theorem 5.3 suggests that a smaller value of $u$ and a larger ratio of $|\gamma_1 - \gamma_2|/\sigma$ increases the inherent robustness to indiscriminate poisoning for typical configurations under our model (2). Empirical verification of the above theoretical results is given in Appendix G.

## 5.2 General Distributions

Recall that we have identified several key factors (i.e., $u$, $|\gamma_1 - \gamma_2|$ and $\sigma$) for 1-D Gaussian distributions in Section 5.1 which are highly related to the performance of an optimal distributional poisoning adversary $\mathcal{A}_{\text{opt}}$. In this section, we demonstrate how to generalize the definition of these factors to high-dimensional datasets and illustrate how they affect an inherent robustness upper bound to indiscriminate poisoning attacks for linear learners. In particular, we project the clean distribution $\mu_c$ and the constraint set $\mathcal{C}$ onto some vector $\boldsymbol{w}$, then compute those factors based on the projections.

**Definition 5.5** (Projected Constraint Size). Let $\mathcal{C} \subseteq \mathcal{X} \times \mathcal{Y}$ be the constraint set for poisoning. For any $\boldsymbol{w} \in \mathbb{R}^n$, the *projected constraint size* of $\mathcal{C}$ with respect to $\boldsymbol{w}$ is defined as:

$$\text{Size}_{\boldsymbol{w}}(\mathcal{C}) = \max_{(\boldsymbol{x}, y) \in \mathcal{C}} \boldsymbol{w}^\top \boldsymbol{x} - \min_{(\boldsymbol{x}, y) \in \mathcal{C}} \boldsymbol{w}^\top \boldsymbol{x}$$

According to Definition 5.5, $\text{Size}_{\boldsymbol{w}}(\mathcal{C})$ characterizes the size of the constraint set $\mathcal{C}$ when projected onto the (normalized) projection vector $\boldsymbol{w}/\|\boldsymbol{w}\|_2$ then scaled by $\|\boldsymbol{w}\|_2$, the $\ell_2$-norm of $\boldsymbol{w}$. In theory, the constraint sets conditioned on $y = -1$ and $y = +1$ can be different, but for simplicity and practical considerations, we simply assume they are the same in the following discussions.

**Definition 5.6** (Projected Separability and Standard Deviation). Let $\mathcal{X} \subseteq \mathbb{R}^n$, $\mathcal{Y} = \{-1, +1\}$, and $\mu_c$ be the underlying distribution. Let $\mu_-$ and $\mu_+$ be the input distributions with labels of $-1$ and $+1$ respectively. For any $\boldsymbol{w} \in \mathbb{R}^n$, the *projected separability* of $\mu_c$ with respect to $\boldsymbol{w}$ is defined as:

$$\text{Sep}_{\boldsymbol{w}}(\mu_c) = \left| \mathbb{E}_{\boldsymbol{x} \sim \mu_-}[\boldsymbol{w}^\top \boldsymbol{x}] - \mathbb{E}_{\boldsymbol{x} \sim \mu_+}[\boldsymbol{w}^\top \boldsymbol{x}] \right|.$$

In addition, the *projected standard deviation* of $\mu_c$ with respect to $\boldsymbol{w}$ is defined as:

$$\text{SD}_{\boldsymbol{w}}(\mu_c) = \sqrt{\text{Var}_{\boldsymbol{w}}(\mu_c)}, \ \text{Var}_{\boldsymbol{w}}(\mu_c) = p_- \cdot \text{Var}_{\boldsymbol{x} \sim \mu_-}[\boldsymbol{w}^\top \boldsymbol{x}] + p_+ \cdot \text{Var}_{\boldsymbol{x} \sim \mu_+}[\boldsymbol{w}^\top \boldsymbol{x}],$$

where $p_- = \text{Pr}_{(\boldsymbol{x}, y) \sim \mu_c}[y = -1]$, $p_+ = \text{Pr}_{(\boldsymbol{x}, y) \sim \mu_c}[y = +1]$ denote the sampling probabilities.

For finite-sample settings, we simply replace the input distributions with their empirical counterparts to compute the sample statistics of $\text{Sep}_{\boldsymbol{w}}(\mu_c)$ and $\text{SD}_{\boldsymbol{w}}(\mu_c)$. Note that the above definitions are specifically for linear models, but out of curiosity, we also provide initial thoughts on how to extend these metrics to neural networks (see Appendix F for preliminary results). Below, we provide justifications on how the three factors are related to the optimal poisoning attacks. Theorem 5.7 and the techniques used in its proof in Appendix B.2 are inspired by the design of Min-Max Attack [45].

**Theorem 5.7.** *Consider input space $\mathcal{X} \subseteq \mathbb{R}^n$, label space $\mathcal{Y}$, clean distribution $\mu_c$ and linear hypothesis class $\mathcal{H}$. For any $h_{\boldsymbol{w},b} \in \mathcal{H}$, $\boldsymbol{x} \in \mathcal{X}$ and $y \in \mathcal{Y}$, let $\ell(h_{\boldsymbol{w},b}; \boldsymbol{x}, y) = \ell_{\text{M}}(-y(\boldsymbol{w}^\top \boldsymbol{x} + b))$ be a margin-based loss adopted by the victim, where $\ell_{\text{M}}$ is convex and non-decreasing. Let $\mathcal{C} \subseteq \mathcal{X} \times \mathcal{Y}$ be the constraint set and $\epsilon > 0$ be the poisoning budget. Suppose $h_c = \text{argmin}_{h \in \mathcal{H}} L(h; \mu_c)$ has weight $\boldsymbol{w}_c$ and $h_p^*$ is the poisoned model induced by optimal adversary $\mathcal{A}_{\text{opt}}$, then we have*

$$\text{Risk}(h_p^*; \mu_c) \leq \min_{h \in \mathcal{H}} \left[ L(h; \mu_c) + \epsilon \cdot L(h; \mu_p^*) \right] \leq L(h_c; \mu_c) + \epsilon \cdot \ell_{\text{M}}(\text{Size}_{\boldsymbol{w}_c}(\mathcal{C})). \tag{5}$$

*Remark* 5.8. Theorem 5.7 proves an upper bound on the inherent vulnerability to indiscriminate poisoning for linear learners, which can be regarded as a necessary condition for the optimal poisoning attack. A smaller upper bound likely suggests a higher inherent robustness to poisoning attacks. In particular, the right hand side of (5) consists of two terms: the clean population loss of $h_c$ and a term related to the projected constraint size. Intuitively, the projected separability and standard deviation metrics highly affect the first term, since data distribution with a higher $\text{Sep}_{\boldsymbol{w}_c}(\mu_c)$ and a lower $\text{SD}_{\boldsymbol{w}_c}(\mu_c)$ implies a larger averaged margin with respect to $h_c$, which further suggests a smaller $L(h_c; \mu_c)$. The second term is determined by the poisoning budget $\epsilon$ and the projected constraint size, or more precisely, a larger $\epsilon$ and a larger $\text{Size}_{\boldsymbol{w}_c}(\mathcal{C})$ indicate a higher upper bound on $\text{Risk}(h_p^*; \mu_c)$. In addition, we set $h = h_c$ and the projection vector as $\boldsymbol{w}_c$ for the last inequality of (5), because $h_c$ achieves the smallest population surrogate loss with respect to the clean data distribution $\mu_c$. However, choosing $h = h_c$ may not always produce a tighter upper bound on $\text{Risk}(h_p^*; \mu_c)$ since there is no guarantee that the projected constraint size $\text{Size}_{\boldsymbol{w}_c}(\mathcal{C})$ is small. An interesting future direction is to select a more appropriate projection vector that returns a tight, if not the tightest, upper bound on $\text{Risk}(h_p^*; \mu_c)$ for any clean distribution $\mu_c$ (see Appendix D.2 for preliminary experiments). We also note that the results above apply to any (non-decreasing and strongly-convex) margin-based losses.

## 6 Experiments

Recall from Theorem 4.3 and Remark 4.4 that the finite-sample optimal poisoning attack is a consistent estimator of the distributional one for linear learners. In this section, we demonstrate the theoretical insights gained from Section 5, despite proven only for the distributional optimal attacks, still appear to largely explain the empirical performance of best attacks across benchmark datasets.

Given a clean training data $\mathcal{S}_c$, we empirically estimate the three distributional metrics defined in Section 5.2 on the clean test data with respect to the weight $\boldsymbol{w}_c$. Since $\|\boldsymbol{w}_c\|_2$ may vary across different datasets while the predictions of linear models (i.e., the classification error) are invariant to the scaling of $\|\boldsymbol{w}_c\|_2$, we use ratios to make their metrics comparable: $\text{Sep}_{\boldsymbol{w}_c}(\mu_c)/\text{SD}_{\boldsymbol{w}_c}(\mu_c)$ (denoted as Sep/SD in Table 1) and $\text{Sep}_{\boldsymbol{w}_c}(\mu_c)/\text{Size}_{\boldsymbol{w}_c}(\mathcal{C})$ (Sep/Size). According to our theoretical results, we expect datasets that are less vulnerable to poisoning have higher values for both metrics.

| | Metric | MNIST 6–9 | Robust MNIST 1–7 | Adult | Dogfish | Moderately Vulnerable MNIST 4–9 | Highly Vulnerable F. Enron | Enron |
|---|---|---|---|---|---|---|---|---|
| SVM | Error Increase | 2.7 | 2.4 | 3.2 | 7.9 | 6.6 | 33.1 | 31.9 |
| | Base Error | 0.3 | 1.2 | 21.5 | 0.8 | 4.3 | 0.2 | 2.9 |
| | Sep/SD | 6.92 | 6.25 | 9.65 | 5.14 | 4.44 | 1.18 | 1.18 |
| | Sep/Size | 0.24 | 0.23 | 0.33 | 0.05 | 0.14 | 0.01 | 0.01 |
| LR | Error Increase | 2.3 | 1.8 | 2.5 | 6.8 | 5.8 | 33.0 | 33.1 |
| | Base Error | 0.6 | 2.2 | 20.1 | 1.7 | 5.1 | 0.3 | 2.5 |
| | Sep/SD | 6.28 | 6.13 | 4.62 | 5.03 | 4.31 | 1.11 | 1.10 |
| | Sep/Size | 0.27 | 0.27 | 0.27 | 0.09 | 0.16 | 0.01 | 0.01 |

Table 1: Explaining disparate poisoning vulnerability under linear models. The top row for each model gives the increase in error rate due to the poisoning, over the base error rate in the second row. The explanatory metrics are the scaled (projected) separability, standard deviation and constraint size.

Table 1 summarizes the results where *Error Increase* is produced by the best attacks from the state-of-the-art attacks mentioned in Section 3. The Sep/SD and Sep/Size metrics are highly correlated to the poisoning effectiveness (as measured by error increase). Datasets such as MNIST 1–7 and MNIST 6–9 are harder to poison than others, which correlates with their increased separability and reduced impact from the poisoning points. In contrast, datasets such as Enron and Filtered Enron are highly vulnerable and this is strongly correlated to their small separability and the large impact from the admissible poisoning points. The results of Filtered Enron (low base error, high increased error) and Adult (high base error, low increased error) demonstrate that poisoning vulnerability, measured by the error increase, cannot be trivially inferred from the initial base error.

When the base error is small, which is the case for all tested benchmark datasets except Adult, the empirical metrics are highly correlated to the error increase and also the final poisoned error. However, when the base error becomes high as it is for Adult, the empirical metrics are highly correlated to the final poisoned error, but not the error increase, if the metrics are computed on the entire (clean) test data. For the error increase, computing the metrics on the clean and correctly classified (by $w_c$) test points is more informative as it (roughly) accounts for the (newly) induced misclassification from poisoning. Therefore, we report metrics based on correctly-classified test points in Table 1 and defer results of the whole test data to Appendix D.2. For datasets except Adult, both ways of computing the metrics produce similar results. The Adult dataset is very interesting in that it is robust to poisoning (i.e., small error increase) despite having a very high base error. As a preliminary experiment, we also extended our analysis to neural networks (Appendix F) and find the identified factors are correlated to the varied poisoning effectiveness when different datasets are compared under similar learners.

Our current results only show the correlation of the identified factors to the varied (empirical) poisoning effectiveness, not causality. However, we also speculate that these factors may *cause* different poisoning effectiveness because: 1) results in Section 5 characterize the relationship of these factors to the performance of optimal attacks exactly for 1-D Gaussian distributions and partially for the general distributions, and 2) preliminary results on MNIST 1–7 shows that gradual changes in one factor (with the other fixed) also causes gradual changes in the vulnerability to poisoning (Appendix D.2.6). We leave the detailed exploration on their causal relationship as future work.

## 7 Discussion on Future Implications

Our results imply future defenses by explaining why candidate defenses work and motivating defenses to improve separability and reduce projected constraint size. We present two ideas—using data filtering might reduce projected constraint size and using better features might improve separability.

**Reduced projected constraint size.** The popular data sanitization defense works by filtering out bad points. We speculate it works because the defense may be effectively limiting the projected constraint size of $C$. To test this, we picked the combination of Sphere and Slab defenses considered in prior works [25, 45] to protect the vulnerable Enron dataset. We find that with the defense, the test error is increased from 3.2% to 28.8% while without the defense the error can be increased from 2.9% to 34.8%. Although limited in effectiveness, the reduced *Error Increase* with the defense is highly correlated to the significantly reduced projected constraint size $\text{Size}_{w_c}(C)$ (and hence a significantly higher Sep/Size)—the Sep/Size metric jumps from 0.01 without defense to 0.11 with defense while

Sep/SD remains almost the same at 1.18 with and without defense. Similar conclusions can also be drawn for MNIST 1-7 and Dogfish (detailed experimental results are in Appendix D.2.3).

**Better feature representation.** We consider a transfer learning scenario where the victim trains a linear model on a clean pretrained model. As a preliminary experiment, we train LeNet and ResNet18 models on the CIFAR10 dataset till convergence, but record the intermediate models of ResNet18 to produce models with different feature extractors (R-$X$ denotes ResNet18 trained for $X$ epochs). We then use the feature extraction layers of these models (including LeNet) as the pretrained models and obtain features of CIFAR10 images with labels "Truck" and "Ship", and train linear models on them.

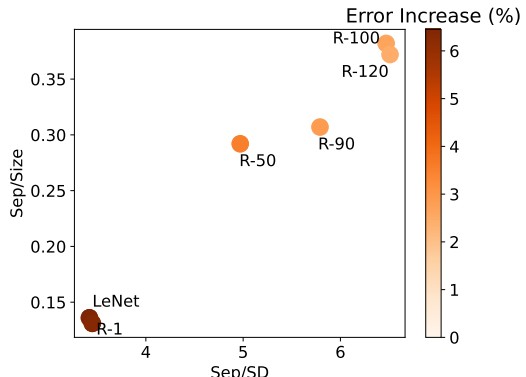

Figure 2: Improving downstream robustness to poisoning through better feature extractors.

We evaluate the robustness of this dataset against poisoning attacks and set $\mathcal{C}$ as dimension-wise box-constraints, whose values are the minimum and maximum values of the clean data points for each dimension when fed to the feature extractors. This way of configuring $\mathcal{C}$ corresponds to the practical scenario where the victim has access to some small number of clean samples so that they can deploy a simple defense of filtering out inputs that do not fall into a dimension-wise box constraint that is computed from the available clean samples of the victim. Figure 2 shows that as the feature extractor becomes better (using deeper models and training for more epochs), both the Sep/SD and Sep/Size metrics increase, leading to reduced error increase. Quantitatively, the Pearson correlation coefficient between the error increase and each of the two factors is $-0.98$ (a strong negative correlation). This result demonstrates the possibility that better feature representations free from poisoning might be leveraged to improve downstream resilience against indiscriminate poisoning attacks. We also simulated a more practical scenario where the training of the pretrained models never sees any data (i.e., data from classes of "Truck" and "Ship") used in the downstream analysis and the conclusions are similar (Appendix D.2.7).

## 8    Limitations

Our work also has several limitations. We only characterize the optimal poisoning attacks for theoretical distributions under linear models. Even for the linear models, the identified metrics cannot quantify the actual error increase from optimal poisoning attacks, which is an interesting future work, and one possible approach might be to tighten the upper bound in Theorem 5.7 using better optimization methods. The metrics identified in this paper are learner dependent, depending on the properties of the learning algorithm, dataset and domain constraints (mainly reflected through $\mathcal{C}$). In certain applications, one might be interested in understanding the impact of learner agnostic dataset properties on poisoning effectiveness—a desired dataset has such properties that any reasonable learners trained on the dataset can be robust to poisoning attacks. One likely application scenario is when the released data will be used by many different learners in various applications, each of which may be prone to poisoning. We also did not systematically investigate how to compare the vulnerabilities of different datasets under different learning algorithms. Identifying properties specific to the underlying learner that affect the performance of (optimal) data poisoning attacks is challenging but interesting future work.

Although we focus on indiscriminate data poisoning attacks in this paper, we are optimistic that our results will generalize to subpopulation or targeted poisoning settings. In particular, the specific learning task properties identified in this paper may still be highly correlated, but now additional factors related to the relative positions of the subpopulations/individual test samples to the rest of the population under the clean decision boundary will also play important roles.

## Acknowledgements

We appreciate the comments from the anonymous reviewers. This work was partially funded by awards from the National Science Foundation (NSF) SaTC program (Center for Trustworthy Machine Learning, #1804603), the AI Institute for Agent-based Cyber Threat Intelligence and Operation (ACTION) (#2229876), NSF #2323105, and NSF #2325369. Any opinions, findings and conclusions or recommendations expressed in this material are those of the authors and do not necessarily reflect the views of the National Science Foundation. Fnu Suya also acknowledges the financial support from the Maryland Cybersecurity Center (MC2) at the University of Maryland to attend the conference.

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

# A  Proofs of Main Results in Section 4

## A.1  Proof of Theorem 4.3

We first introduce the formal definitions of strong convexity and Lipschitz continuity conditions with respect to a function, and the uniform convergence property respect to a hypothesis class. These definitions are necessary for the proof of Theorem 4.3.

**Definition A.1** (Strong Convexity). A function $f : \mathbb{R}^n \to \mathbb{R}$ is $b$-strongly convex for some $b > 0$, if $f(\boldsymbol{x}_1) \geq f(\boldsymbol{x}_2) + \nabla f(\boldsymbol{x}_2)^\top (\boldsymbol{x}_1 - \boldsymbol{x}_2) + \frac{b}{2}\|\boldsymbol{x}_1 - \boldsymbol{x}_2\|_2^2$ for any $\boldsymbol{x}_1, \boldsymbol{x}_2 \in \mathbb{R}^n$.

**Definition A.2** (Lipschitz Continuity). A function $f : \mathbb{R}^n \to \mathbb{R}$ is $\rho$-Lipschitz for some $\rho > 0$, if $|f(\boldsymbol{x}_1) - f(\boldsymbol{x}_2)| \leq \rho\|\boldsymbol{x}_1 - \boldsymbol{x}_2\|_2$ for any $\boldsymbol{x}_1, \boldsymbol{x}_2 \in \mathbb{R}^n$.

**Definition A.3** (Uniform Convergence). Let $\mathcal{H}$ be a hypothesis class. We say that $\mathcal{H}$ satisfies the *uniform convergence property* with a loss function $\ell$, if there exists a function $m_{\mathcal{H}} : (0, 1)^2 \to \mathbb{N}$ such that for every $\epsilon', \delta' \in (0, 1)$ and for every probability distribution $\mu$, if $\mathcal{S}$ is a set of examples with $m \geq m_{\mathcal{H}}(\epsilon', \delta')$ samples drawn i.i.d. from $\mu$, then

$$\mathbb{P}_{\mathcal{S} \leftarrow \mu^m}\left[\sup_{h \in \mathcal{H}} \left|L(h; \hat{\mu}_{\mathcal{S}}) - L(h; \mu)\right| \leq \epsilon'\right] \geq 1 - \delta'.$$

Such a uniform convergence property, which can be achieved using the VC dimension or the Rademacher complexity of $\mathcal{H}$, guarantees that the learning rule specified by empirical risk minimization always returns a good hypothesis with high probability [43]. Similar to PAC learning, the function $m_{\mathcal{H}}$ measures the minimal sample complexity requirement that ensures uniform convergence.

Now, we are ready to prove Theorem 4.3

*Proof of Theorem 4.3.* First, we introduce the following notations to simplify the proof. For any $\mathcal{S}_p$, $\mu_p$ and $\delta \geq 0$, let

$$\hat{g}(\mathcal{S}_p, \mathcal{S}_c) = \operatorname*{argmin}_{h \in \mathcal{H}} \sum_{(\boldsymbol{x}, y) \in \mathcal{S}_c \cup \mathcal{S}_p} \ell(h; \boldsymbol{x}, y),$$

$$g(\delta, \mu_p, \mu_c) = \operatorname*{argmin}_{h \in \mathcal{H}} \{L(h; \mu_c) + \delta \cdot L(h; \mu_p)\}.$$

According to the definitions of $\hat{h}_p^*$ and $h_p^*$, we know $\hat{h}_p^* = \hat{g}(\mathcal{S}_p^*, \mathcal{S}_c)$ and $h_p^* = g(\delta^*, \mu_p^*, \mu_c)$.

Now we are ready to prove Theorem 4.3. For any $\mathcal{S}_c$ sampled from $\mu_c$, consider the empirical loss minimizer $\hat{h}_p^* = \hat{g}(\mathcal{S}_p^*, \mathcal{S}_c)$ and the population loss minimizer $g(\delta_{\mathcal{S}_p^*}, \hat{\mu}_{\mathcal{S}_p^*}, \mu_c)$, where $\delta_{\mathcal{S}_p^*} = |\mathcal{S}_p^*|/|\mathcal{S}_c|$. Then $\mathcal{S}_p^* \cup \mathcal{S}_c$ can be regarded as the i.i.d. sample set from $(\mu_c + \delta_{\mathcal{S}_p^*} \cdot \hat{\mu}_{\mathcal{S}_p^*})/(1 + \delta_{\mathcal{S}_p^*})$. According to Definition A.3, since $\mathcal{H}$ satisfies the uniform convergence property with respect to $\ell$, we immediately know that the empirical loss minimization is close to the population loss minimization if the sample size is large enough (see Lemma 4.2 in [43]). To be more specific, for any $\epsilon', \delta' \in (0, 1)$, if $|\mathcal{S}_c| \geq m_{\mathcal{H}}(\epsilon', \delta')$, then with probability at least $1 - \delta'$, we have

$$L\big(\hat{g}(\mathcal{S}_p^*, \mathcal{S}_c); \mu_c\big) + \delta_{\mathcal{S}_p^*} \cdot L\big(\hat{g}(\mathcal{S}_p^*, \mathcal{S}_c); \hat{\mu}_{\mathcal{S}_p^*}\big) \leq \operatorname*{argmin}_{h \in \mathcal{H}} \{L(h; \mu_c) + \delta_{\mathcal{S}_p^*} \cdot L(h; \hat{\mu}_{\mathcal{S}_p^*})\} + 2\epsilon'$$

$$= L\big(g(\delta_{\mathcal{S}_p^*}, \hat{\mu}_{\mathcal{S}_p^*}, \mu_c); \mu_c\big) + \delta_{\mathcal{S}_p^*} \cdot L\big(g(\delta_{\mathcal{S}_p^*}, \hat{\mu}_{\mathcal{S}_p^*}, \mu_c); \hat{\mu}_{\mathcal{S}_p^*}\big) + 2\epsilon'.$$

In addition, since the surrogate loss $\ell$ is $b$-strongly convex and the population risk is $\rho$-Lipschitz, we further know the clean risk of $\hat{g}(\mathcal{S}_p^*, \mathcal{S}_c)$ and $g(\delta_{\mathcal{S}_p^*}, \hat{\mu}_{\mathcal{S}_p^*}, \mu_c)$ is guaranteed to be close. Namely, with probability at least $1 - \delta'$, we have

$$\left|\operatorname{Risk}\big(\hat{g}(\mathcal{S}_p^*, \mathcal{S}_c); \mu_c\big) - \operatorname{Risk}\big(g(\delta_{\mathcal{S}_p^*}, \hat{\mu}_{\mathcal{S}_p^*}, \mu_c); \mu_c\big)\right| \leq \rho \cdot \big\|\hat{g}(\mathcal{S}_p^*, \mathcal{S}_c) - g(\delta_{\mathcal{S}_p^*}, \hat{\mu}_{\mathcal{S}_p^*}, \mu_c)\big\|_2$$

$$\leq 2\rho\sqrt{\frac{\epsilon'}{b}}.$$

Note that $\delta_{\mathcal{S}_p^*} \in [0, \epsilon]$ and $\operatorname{supp}(\hat{\mu}_{\mathcal{S}_p^*}) \subseteq \mathcal{C}$. Thus, according to the definition of $h_p^* = g(\delta^*, \mu_p^*, \mu_c)$, we further have

$$\operatorname{Risk}(h_p^*; \mu_c) \geq \operatorname{Risk}\big(g(\delta_{\mathcal{S}_p^*}, \hat{\mu}_{\mathcal{S}_p^*}, \mu_c); \mu_c\big) \geq \operatorname{Risk}\big(\hat{g}(\mathcal{S}_p^*, \mathcal{S}_c); \mu_c\big) - 2\rho\sqrt{\frac{\epsilon'}{b}}$$

$$= \operatorname{Risk}(\hat{h}_p^*; \mu_c) - 2\rho\sqrt{\frac{\epsilon'}{b}}. \tag{6}$$

So far, we have proven one direction of the asymptotic for results Theorem 4.3.

On the other hand, we can always construct a subset $\tilde{\mathcal{S}}_p$ with size $|\tilde{\mathcal{S}}_p| = \delta^* \cdot |\mathcal{S}_c|$ by i.i.d. sampling from $\mu_p^*$. Consider the empirical risk minimizer $\hat{g}(\tilde{\mathcal{S}}_p, \mathcal{S}_c)$ and the population risk minimizer $h_p^* = g(\delta^*, \mu_p^*, \mu_c)$. Similarly, since $\mathcal{H}$ satisfies the uniform convergence property, if $|\mathcal{S}_c| \geq m_{\mathcal{H}}(\epsilon', \delta')$, then with probability at least $1 - \delta'$, we have

$$L\big(\hat{g}(\tilde{\mathcal{S}}_p, \mathcal{S}_c); \mu_c\big) + \delta^* \cdot L\big(\hat{g}(\tilde{\mathcal{S}}_p, \mathcal{S}_c); \mu_p^*\big) \leq \underset{h \in \mathcal{H}}{\operatorname{argmin}} \left\{ L(h; \mu_c) + \delta^* \cdot L(h; \mu_p^*) \right\} + 2\epsilon'$$
$$= L\big(g(\delta^*, \mu_p^*, \mu_c); \mu_c\big) + \delta^* \cdot L\big(g(\delta^*, \mu_p^*, \mu_c); \mu_p^*\big) + 2\epsilon'.$$

According to the strong convexity of $\ell$ and the Lipschitz continuity of the population risk, we further have

$$\left| \operatorname{Risk}\big(\hat{g}(\tilde{\mathcal{S}}_p, \mathcal{S}_c); \mu_c\big) - \operatorname{Risk}\big(g(\delta^*, \mu_p^*, \mu_c); \mu_c\big) \right| \leq \rho \cdot \big\| \hat{g}(\tilde{\mathcal{S}}_p, \mathcal{S}_c) - g(\delta^*, \mu_p^*, \mu_c) \big\|_2$$
$$\leq 2\rho \sqrt{\frac{\epsilon'}{b}}.$$

Note that $\tilde{\mathcal{S}}_p \subseteq \mathcal{C}$ and $|\tilde{\mathcal{S}}_p| = \delta^* \cdot |\mathcal{S}_c| \leq \epsilon \cdot |\mathcal{S}_c|$. Thus according to the definition of $\hat{h}_p^* = \hat{g}(\mathcal{S}_p^*, \mathcal{S}_c)$, we have

$$\operatorname{Risk}(\hat{h}_p^*; \mu_c) \geq \operatorname{Risk}\big(\hat{g}(\tilde{\mathcal{S}}_p, \mathcal{S}_c); \mu_c\big) \geq \operatorname{Risk}\big(g(\delta^*, \mu_p^*, \mu_c); \mu_c\big) - 2\rho\sqrt{\frac{\epsilon'}{b}}$$
$$= \operatorname{Risk}(h_p^*; \mu_c) - 2\rho\sqrt{\frac{\epsilon'}{b}}. \tag{7}$$

Combining (6) and (7), we complete the proof of Theorem 4.3. $\qquad\square$

## A.2 Proof of Theorem 4.5

*Proof of Theorem 4.5.* We prove Theorem 4.5 by construction.

We start with the first condition $\operatorname{supp}(\mu_c) \subseteq \mathcal{C}$. Suppose $\delta^* < \epsilon$, since the theorem trivially holds if $\delta^* = \epsilon$. To simplify notations, define $h_p(\delta, \mu_p) = \operatorname{argmin}_{h \in \mathcal{H}} \left\{ L(h; \mu_c) + \delta \cdot L(h; \mu_p) \right\}$ for any $\delta$ and $\mu_p$. To prove the statement in Theorem 4.5, it is sufficient to show that there exists some $\mu_p^{(\epsilon)}$ based on the first condition such that

$$\operatorname{Risk}\big(h_p(\epsilon, \mu_p^{(\epsilon)}); \mu_c\big) = \operatorname{Risk}\big(h_p(\delta^*, \mu_p^*); \mu_c\big), \text{ and } \operatorname{supp}(\mu_p^{(\epsilon)}) \subseteq \mathcal{C}. \tag{8}$$

The above equality means we can always achieve the same maximum risk after poisoning with the full poisoning budget $\epsilon$. To proceed with the proof, we construct $\mu_p^{(\epsilon)}$ based on $\mu_c$ and $\mu_p^*$ as follows:

$$\mu_p^{(\epsilon)} = \frac{\delta^*}{\epsilon} \cdot \mu_p^* + \frac{\epsilon - \delta^*}{\epsilon(1 + \delta^*)} \cdot (\mu_c + \delta^* \cdot \mu_p^*)$$
$$= \frac{\epsilon - \delta^*}{\epsilon(1 + \delta^*)} \cdot \mu_c + \frac{\delta^*(1 + \epsilon)}{\epsilon(1 + \delta^*)} \cdot \mu_p^*.$$

We can easily check that $\mu_p^{(\epsilon)}$ is a valid probability distribution and $\operatorname{supp}(\mu_p^{(\epsilon)}) \subseteq \mathcal{C}$. In addition, we can show that

$$h_p(\epsilon, \mu_p^{(\epsilon)}) = \underset{h \in \mathcal{H}}{\operatorname{argmin}} \left\{ L(h; \mu_c) + \epsilon \cdot L(h; \mu_p^{(\epsilon)}) \right\}$$
$$= \underset{h \in \mathcal{H}}{\operatorname{argmin}} \left\{ \mathbb{E}_{(\boldsymbol{x}, y) \sim \mu_c} \ell(h; \boldsymbol{x}, y) + \epsilon \cdot \mathbb{E}_{(\boldsymbol{x}, y) \sim \mu_p^{(\epsilon)}} \ell(h; \boldsymbol{x}, y) \right\}$$
$$= \underset{h \in \mathcal{H}}{\operatorname{argmin}} \left\{ \frac{1 + \epsilon}{1 + \delta^*} \cdot \big( \mathbb{E}_{(\boldsymbol{x}, y) \sim \mu_c} \ell(h; \boldsymbol{x}, y) + \delta^* \cdot \mathbb{E}_{(\boldsymbol{x}, y) \sim \mu_p^*} \ell(h; \boldsymbol{x}, y) \big) \right\}$$
$$= h_p(\delta^*, \mu_p^*)$$

where the third equality holds because of the construction of $\mu_p^{(\epsilon)}$. Therefore, we have proven (8), which further implies the optimal attack performance can always be achieved with $\epsilon$-poisoning as long as the first condition is satisfied.

Next, we turn to the second condition of Theorem 4.5. Similarly, it is sufficient to construct some $\mu_p^{(\epsilon)}$ for the setting where $\delta^* < \epsilon$ such that

$$\text{Risk}\big(h_p(\epsilon, \mu_p^{(\epsilon)}); \mu_c\big) = \text{Risk}\big(h_p(\delta^*, \mu_p^*); \mu_c\big), \text{ and } \text{supp}(\mu_p^{(\epsilon)}) \subseteq \mathcal{C}.$$

We construct $\mu_p^{(\epsilon)}$ based on $\mu_p^*$ and the assumed data distribution $\mu$. More specifically, we construct

$$\mu_p^{(\epsilon)} = \frac{\delta^*}{\epsilon} \cdot \mu_p^* + \frac{\epsilon - \delta^*}{\epsilon} \cdot \mu. \tag{9}$$

By construction, we know $\mu_p^{(\epsilon)}$ is a valid probability distribution. In addition, according to the assumption of $\text{supp}(\mu) \subseteq \mathcal{C}$, we have $\text{supp}(\mu_p^{(\epsilon)}) \subseteq \mathcal{C}$. According to the assumption that for any $\boldsymbol{\theta}$, there exists a $\mu$ such that $\frac{\partial}{\partial \boldsymbol{\theta}} L(h_{\boldsymbol{\theta}}; \mu) = \mathbf{0}$, we know for any possible weight parameter $\boldsymbol{\theta}_p^*$ of $h_p(\delta^*, \mu_p^*)$, there also exists a corresponding $\mu$ such that the gradient is $\mathbf{0}$ and therefore, we have

$$\frac{\partial}{\partial \boldsymbol{\theta}_p^*}\big(L(h_p(\delta^*, \mu_p^*); \mu_c) + \epsilon \cdot L(h_p(\delta^*, \mu_p^*); \mu_p^{(\epsilon)})\big)$$

$$= \frac{\partial}{\partial \boldsymbol{\theta}_p^*}\big(L(h_p(\delta^*, \mu_p^*); \mu_c) + \delta^* \cdot L(h_p(\delta^*, \mu_p^*); \mu_p^*)\big)$$

$$= \mathbf{0}$$

where the last equality is based on the first-order optimality condition of $h_p(\delta^*, \mu_p^*)$ for convex losses. For simplicity, we also assumed $h_p(\delta^*, \mu_p^*)$ is obtained by minimizing the loss on $\mu_c + \delta^* \cdot \mu_p^*$ while in the case of $\text{supp}(\mu_c) \not\subseteq \mathcal{C}$, the victim usually minimizes the loss on $\bar{\mu}_c + \delta^* \cdot \mu_p^*$, where $\bar{\mu}_c$ is the "truncated" version of $\mu_c$ such that $\text{supp}(\bar{\mu}_c) \subseteq \mathcal{C}$. To conclude, we know $h_p(\epsilon, \mu_p^{(\epsilon)}) = h_p(\delta^*, \mu_p^*)$ holds for any possible $h_p(\delta^*, \mu_p^*)$ and we complete the proof of Theorem 4.5.

$\square$

### A.3 Proofs of the Statement about Linear Models in Remark 4.6

*Proof.* We provide the construction of $\mu$ with respect to the second condition of Theorem 4.5 for linear models and hinge loss. Since for any $h_{\boldsymbol{w},b} \in \mathcal{H}_L$ and any $(\boldsymbol{x}, y) \in \mathcal{X} \times \mathcal{Y}$, we have

$$\ell(h_{\boldsymbol{w},b}; \boldsymbol{x}, y) = \max\{0, 1 - y(\boldsymbol{w}^\top \boldsymbol{x} + b)\} + \frac{\lambda}{2}\|\boldsymbol{w}\|_2^2.$$

Let $\boldsymbol{\theta} = (\boldsymbol{w}, b)$, then the gradient with respect to $\boldsymbol{w}$ can be written as:

$$\frac{\partial}{\partial \boldsymbol{w}} \ell(h_{\boldsymbol{w},b}; \boldsymbol{x}, y) = \begin{cases} -y \cdot \boldsymbol{x} + \lambda \boldsymbol{w} & \text{if } y(\boldsymbol{w}^\top \boldsymbol{x} + b) \leq 1, \\ \lambda \boldsymbol{w} & \text{otherwise.} \end{cases}$$

Similarly, the gradient with respect to $b$ can be written as:

$$\frac{\partial}{\partial b} \ell(h_{\boldsymbol{w},b}; \boldsymbol{x}, y) = \begin{cases} -y & \text{if } y(\boldsymbol{w}^\top \boldsymbol{x} + b) \leq 1, \\ 0 & \text{otherwise.} \end{cases}$$

Therefore, for large input space $\mathcal{X} \times \mathcal{Y}$, we can simply construct $\mu$ by constructing a two-point distribution (with equal probabilities of label +1 and -1) that cancels out each other's gradient (for both $\boldsymbol{w}$ and $b$), so that $y(\boldsymbol{w}^\top \boldsymbol{x} + b) \leq 1$ and $-y \cdot \boldsymbol{x} + \lambda \boldsymbol{w} = \mathbf{0}$.

We may also generalize the construction of $\mu$ from linear models with hinge loss to general convex models if the victim minimizes the loss on $\bar{\mu}_c + \delta^* \cdot \mu_p^*$ to obtain $h_p(\delta^*, \mu_p^*)$, which is common in practice. In this case, we can simply set $\mu = \bar{\mu}_c + \delta^* \cdot \mu_p^*$, which guarantees

$$\frac{\partial}{\partial \boldsymbol{\theta}_p^*} L(h_p(\delta^*, \mu_p^*); \mu) = \mathbf{0}.$$

$\square$

# B  Proofs of Main Results in Section 5

## B.1  Proof of Theorem 5.3

To prove Theorem 5.3, we need to make use of the following three auxiliary lemmas, which are related to the maximum population hinge loss with $w = 1$ (Lemma B.1), the weight-flipping condition (Lemma B.2) and the risk behaviour of any linear hypothesis under (2) (Lemma B.3). For the sake of completeness, we present the full statements of Lemma B.1 and Lemma B.2 as follows. In particular, Lemma B.1, proven in Appendix C.1, characterizes the maximum achievable hinge loss with respect to the underlying clean distribution $\mu_c$ and some poisoned distribution $\mu_p$ conditioned on $w = 1$.

**Lemma B.1.** *Suppose the underlying clean distribution $\mu_c$ follows the Gaussian mixture model* (2) *with $p = 1/2$ and $\sigma_1 = \sigma_2 = \sigma$. Assume $|\gamma_1 + \gamma_2| \leq 2u$. For any $\epsilon \geq 0$, consider the following maximization problem:*

$$\max_{\mu_p \in \mathcal{Q}(u)} \left[ L(h_{1,b_p}; \mu_c) + \epsilon \cdot L(h_{1,b_p}; \mu_p) \right], \tag{10}$$

*where $b_p = \mathrm{argmin}_{b \in \mathbb{R}}[L(h_{1,b}; \mu_c) + \epsilon \cdot L(h_{1,b}; \mu_p)]$. There exists some $\alpha \in [0,1]$ such that the optimal value of* (10) *is achieved with $\mu_p = \nu_\alpha$, where $\nu_\alpha$ is a two-point distribution with some parameter $\alpha \in [0,1]$ defined according to* (3).

Lemma B.1 suggests that it is sufficient to study the extreme two-point distributions $\nu_\alpha$ with $\alpha \in [0,1]$ to understand the maximum achievable population hinge loss conditioned on $w = 1$. Lemma B.2, proven in Appendix C.2, characterizes the sufficient and necessary conditions in terms of $\epsilon$, $u$ and $\mu_c$, under which there exists a linear hypothesis with $w = -1$ that achieves the minimal value of population hinge loss with respect to $\mu_c$ and some $\mu_p$.

**Lemma B.2.** *Suppose the underlying clean distribution $\mu_c$ follows the Gaussian mixture model* (2) *with $p = 1/2$ and $\sigma_1 = \sigma_2 = \sigma$. Assume $|\gamma_1 + \gamma_2| \leq 2(u-1)$ for some $u \geq 1$. Let $g$ be an auxiliary function such that for any $b \in \mathbb{R}$,*

$$g(b) = \frac{1}{2}\Phi\left(\frac{b + \gamma_1 + 1}{\sigma}\right) - \frac{1}{2}\Phi\left(\frac{-b - \gamma_2 + 1}{\sigma}\right),$$

*where $\Phi$ is the CDF of standard Gaussian. For any $\epsilon > 0$, there exists some $\mu_p \in \mathcal{Q}(u)$ such that $\mathrm{argmin}_{h_{w,b} \in \mathcal{H}_L}[L(h_{w,b}; \mu_c) + \epsilon \cdot L(h_{w,b}; \mu_p)]$ outputs a hypothesis with $w = -1$, if and only if*

$$\max\{\Delta(-\epsilon), \Delta(g(0)), \Delta(\epsilon)\} \geq 0,$$

*where $\Delta(s) = L(h_{1,g^{-1}(s)}; \mu_c) - \min_{b \in \mathbb{R}} L(h_{-1,b}; \mu_c) + \epsilon(1 + u) - s \cdot g^{-1}(s)$, and $g^{-1}$ denotes the inverse of $g$.*

Lemma B.2 identifies sufficient and necessary conditions when a linear hypothesis with flipped weight parameter is possible. Note that we assume $\gamma_1 \leq \gamma_2$, thus flipping the weight parameter of the induced model from $w = 1$ to $w = -1$ is always favorable from an attacker's perspective. In particular, if the population hinge loss with respect to $\mu_c$ and some $\mu_p$ achieved by the loss minimizer conditioned on $w = 1$ is higher than that achieved by the loss minimizer with $w = -1$, then we immediately know that flipping the weight parameter is possible, which further suggests the optimal poisoning attack performance must be achieved by some poisoned victim model with $w = -1$.

Finally, we introduce Lemma B.3, proven in Appendix C.3, which characterizes the risk behavior of any linear hypothesis with respect to the assumed Gaussian mixture model (2).

**Lemma B.3.** *Let $\mu_c$ be the clean data distribution, where each example is sampled i.i.d. according to the data generating process specified in* (2). *For any linear hypothesis $h_{w,b} \in \mathcal{H}_L$, we have*

$$\mathrm{Risk}(h_{w,b}; \mu_c) = p \cdot \Phi\left(\frac{b + w \cdot \gamma_1}{\sigma_1}\right) + (1 - p) \cdot \Phi\left(\frac{-b - w \cdot \gamma_2}{\sigma_2}\right),$$

*where $\Phi$ denotes the CDF of standard Gaussian distribution $\mathcal{N}(0,1)$.*

Now we are ready to prove Theorem 5.3 using Lemmas B.1, B.2 and B.3.

*Proof of Theorem 5.3.* According to Theorem 4.5 and Remark 4.6, we note that the optimal poisoning performance in Definition 4.2 is always achieved with $\delta = \epsilon$. Therefore, we will only consider $\delta = \epsilon$ in the following discussions.

Since the optimal poisoning performance is defined with respect to clean risk, it will be useful to understand the properties of $\text{Risk}(h_{w,b}; \mu_c)$ such as monotonicity and range. According to Lemma B.3, for any $h_{w,b} \in \mathcal{H}_\text{L}$, we have

$$\text{Risk}(h_{w,b}; \mu_c) = \frac{1}{2}\Phi\left(\frac{b + w \cdot \gamma_1}{\sigma}\right) + \frac{1}{2}\Phi\left(\frac{-b - w \cdot \gamma_2}{\sigma}\right).$$

To understand the monotonicity of risk, we compute its derivative with respect to $b$:

$$\frac{\partial}{\partial b}\text{Risk}(h_{w,b}; \mu_c) = \frac{1}{2\sigma\sqrt{2\pi}}\left[\exp\left(-\frac{(b + w \cdot \gamma_1)^2}{2\sigma^2}\right) - \exp\left(-\frac{(b + w \cdot \gamma_2)^2}{2\sigma^2}\right)\right].$$

If $w = 1$, then $\text{Risk}(h_{w,b}; \mu_c)$ is monotonically decreasing when $b \in (-\infty, -\frac{\gamma_1+\gamma_2}{2})$ and monotonically increasing when $b \in (-\frac{\gamma_1+\gamma_2}{2}, \infty)$, suggesting that minimum is achieved at $b = -\frac{\gamma_1+\gamma_2}{2}$ and maximum is achieved when $b$ goes to infinity. To be more specific, $\text{Risk}(h_{1,b}; \mu_c) \in [\Phi(\frac{\gamma_1-\gamma_2}{2\sigma}), \frac{1}{2}]$. On the other hand, if $w = -1$, then $\text{Risk}(h_{w,b}; \mu_c)$ is monotonically increasing when $b \in (-\infty, \frac{\gamma_1+\gamma_2}{2})$ and monotonically decreasing when $b \in (\frac{\gamma_1+\gamma_2}{2}, \infty)$, suggesting that maximum is achieved at $b = \frac{\gamma_1+\gamma_2}{2}$ and minimum is achieved when $b$ goes to infinity. Thus, $\text{Risk}(h_{-1,b}; \mu_c) \in [\frac{1}{2}, \Phi(\frac{\gamma_2-\gamma_1}{2\sigma})]$.

Based on the monotonicity analysis of $\text{Risk}(h_{w,b}; \mu_c)$, we have the following two observations:

1. If there exists some feasible $\mu_p$ such that $h_{-1,b_p} = \text{argmin}_{h \in \mathcal{H}_\text{L}}\{L(h; \mu_c) + \epsilon L(h; \mu_p)\}$ can be achieved, then the optimal poisoning performance is achieved with $w = -1$ and $b$ close to $\frac{\gamma_1+\gamma_2}{2}$ as much as possible.
2. If there does not exist any feasible $\mu_p$ that induces $h_{-1,b_p}$ by minimizing the population hinge loss, then the optimal poisoning performance is achieved with $w = 1$ and $b$ far from $-\frac{\gamma_1+\gamma_2}{2}$ as much as possible (conditioned that the variance $\sigma$ is the same for the two classes).

Recall that we prove in Lemma B.2 specifies a sufficient and necessary condition for the existence of such $h_{-1,b_p}$, which is equivalent to the condition (4) presented in Lemma B.2. Note that according to Lemma C.1, $b = \frac{\gamma_1+\gamma_2}{2}$ also yields the population loss minimizer with respect to $\mu_c$ conditioned on $w = -1$. Thus, if condition (4) is satisfied, then we know there exists some $\alpha \in [0, 1]$ such that the optimal poisoning performance can be achieved with $\mu_p = \nu_\alpha$. This follows from the assumption $|\gamma_1 + \gamma_2| \le 2(u - 1)$, which suggests that for any $(x, y) \sim \nu_\alpha$, the individual hinge loss at $(x, y)$ will be zero. In addition, we know that the poisoned hypothesis induced by $\mathcal{A}_\text{opt}$ is $h_{-1,\frac{\gamma_1+\gamma_2}{2}}$, which maximizes risk with respect to $\mu_c$.

On the other hand, if condition (4) is not satisfied, we know that the poisoned hypothesis induced by any feasible $\mu_p$ has weight parameter $w = 1$. Based on our second observation, this further suggests that the optimal poisoning performance will always be achieved with either $\mu_p = \nu_0$ or $\mu_p = \nu_1$. According to the first-order optimality condition and Lemma C.1, we can compute the closed-form solution regarding the optimal poisoning performance. Thus, we complete the proof. $\square$

## B.2 Proof of Theorem 5.7

*Proof of Theorem 5.7.* Consider linear hypothesis class $\mathcal{H}$ and the poisoned distribution $\mu_p^*$ generated by the optimal poisoning adversary $\mathcal{A}_\text{opt}$ in Definition 4.1. Given clean distribution $\mu_c$, poisoning ratio $\epsilon$ and constraint set $\mathcal{C}$, the inherent vulnerability to indiscriminate poisoning is captured by the optimal attack performance $\text{Risk}(h_p^*; \mu_c)$, where $h_p^*$ denotes the poisoned linear model induced by $\mu_p^*$. For any $h \in \mathcal{H}$, we have

$$\text{Risk}(h_p^*; \mu_c) \le L(h_p^*; \mu_c) \le L(h_p^*; \mu_c) + \epsilon \cdot L(h_p^*; \mu_p^*) \le L(h; \mu_c) + \epsilon \cdot L(h; \mu_p^*) \quad (11)$$

where the first inequality holds because the surrogate loss is defined to be not smaller than the 0-1 loss, the second inequality holds because the surrogate loss is always non-negative, and the third inequality holds because $h_p^*$ minimizes the population loss with respect to both clean distribution

$\mu$ and optimally generated poisoned distribution $\mu_p^*$. Consider $h_c = \mathrm{argmin}_{h \in \mathcal{H}} L(h; \mu_c)$ (with weight parameter $\boldsymbol{w}_c$ and bias parameter $b_c$), which is the linear model learned from the clean data. Therefore, plugging $h = h_c$ into the right hand side of (11), we further obtain

$$\mathrm{Risk}(h_p^*; \mu_c) \leq L(h_c; \mu_c) + \epsilon \cdot L(h_c; \mu_p^*) \leq L(h_c; \mu_c) + \epsilon \cdot \ell_\mathrm{M}(\mathrm{Size}_{\boldsymbol{w}_c}(\mathcal{C})), \qquad (12)$$

where the last inequality holds because for any poisoned data point $(\boldsymbol{x}, y) \sim \mu_p^*$, the surrogate loss at $(\boldsymbol{x}, y)$ with respect to $h_c$ is $\ell_\mathrm{M}\big(y \cdot (\boldsymbol{w}_c^\top \boldsymbol{x} + b_c)\big)$, and $y \cdot (\boldsymbol{w}_c^\top \boldsymbol{x} + b_c) \leq \max_{(\boldsymbol{x},y) \in \mathcal{C}} |\boldsymbol{w}_c^\top \boldsymbol{x} + b_c|$. Under the condition that $\min_{(\boldsymbol{x},y) \in \mathcal{C}} \boldsymbol{w}_c^\top \boldsymbol{x} \leq -b_c \leq \max_{(\boldsymbol{x},y) \in \mathcal{C}} \boldsymbol{w}_c^\top \boldsymbol{x}$ which means the decision boundary of $h_c$ falls into the constraint set $\mathcal{C}$ when projected on to the direction of $\boldsymbol{w}_c$, we further have $\max_{(\boldsymbol{x},y) \in \mathcal{C}} |\boldsymbol{w}_c^\top \boldsymbol{x} + b_c| \leq \mathrm{Size}_{\boldsymbol{w}_c}(\mathcal{C})$, which implies the validity of (12). We remark that the condition $\min_{(\boldsymbol{x},y) \in \mathcal{C}} \boldsymbol{w}_c^\top \boldsymbol{x} \leq -b_c \leq \max_{(\boldsymbol{x},y) \in \mathcal{C}} \boldsymbol{w}_c^\top \boldsymbol{x}$ typically holds for margin-based loss in practice, since the support of the clean training data belongs to the constraint set for poisoning inputs (for either undefended victim models or models that employ some unsupervised data sanitization defense). Therefore, we leave this condition out in the statement of Theorem 5.7 for simplicity. $\qquad \square$

## C  Proofs of Technical Lemmas used in Appendix B.1

### C.1  Proof of Lemma B.1

To prove Lemma B.1, we need to make use of the following general lemma which characterizes the population hinge loss and its derivative with respect to clean data distribution $\mu_c$. For the sake of completeness, we provide the proof of Lemma C.1 in Appendix C.4.

**Lemma C.1.** *Let $\mu_c$ be data distribution generated according to* (2). *For any $h_{w,b} \in \mathcal{H}_\mathrm{L}$, the population hinge loss is:*

$$L(h_{w,b}; \mu_c) = p \int_{\frac{-b - w \cdot \gamma_1 - 1}{\sigma_1}}^{\infty} (b + w \cdot \gamma_1 + 1 + \sigma_1 z) \cdot \varphi(z) dz$$
$$+ (1 - p) \int_{-\infty}^{\frac{-b - w \cdot \gamma_2 + 1}{\sigma_2}} (-b - w \cdot \gamma_2 + 1 - \sigma_2 z) \cdot \varphi(z) dz,$$

*and its gradient with respect to $b$ is:*

$$\frac{\partial}{\partial b} L(h_{w,b}; \mu_c) = p \cdot \Phi\left(\frac{b + w \cdot \gamma_1 + 1}{\sigma_1}\right) - (1 - p) \cdot \Phi\left(\frac{-b - w \cdot \gamma_2 + 1}{\sigma_2}\right),$$

*where $\varphi$ and $\Phi$ denote the PDF and CDF of standard Gaussian distribution $\mathcal{N}(0, 1)$, respectively.*

Next, let us summarize several key observations based on Lemma C.1 (specifically for the setting considered in Lemma B.1). For any $w \in \{-1, 1\}$, $\frac{\partial}{\partial b} L(h_{w,b}; \mu_c)$ is a monotonically increasing with $b$, which achieves minimum $-\frac{1}{2}$ when $b$ goes to $-\infty$ and achieves maximum $\frac{1}{2}$ when $b$ goes to $\infty$. If $w = +1$, then $L(h_{w,b}; \mu_c)$ is monotonically decreasing when $b \in (-\infty, -\frac{\gamma_1 + \gamma_2}{2})$ and monotonically increasing when $b \in (-\frac{\gamma_1 + \gamma_2}{2}, \infty)$, reaching the minimum at $b = b_c^*(1) := -\frac{\gamma_1 + \gamma_2}{2}$. On the other hand, if $w = -1$, then $L(h_{w,b}; \mu_c)$ is monotonically decreasing when $b \in (-\infty, \frac{\gamma_1 + \gamma_2}{2})$ and monotonically increasing when $b \in (\frac{\gamma_1 + \gamma_2}{2}, \infty)$, reaching the minimum at $b = b_c^*(-1) := \frac{\gamma_1 + \gamma_2}{2}$.

As for the clean loss minimizer conditioned on $w = 1$, we have

$$L(h_{1, b_c^*(1)}; \mu_c) = \frac{1}{2} \int_{\frac{\gamma_2 - \gamma_1 - 2}{2\sigma}}^{\infty} \left(\frac{\gamma_1 - \gamma_2}{2} + 1 + \sigma z\right) \cdot \varphi(z) dz$$
$$+ \frac{1}{2} \int_{-\infty}^{\frac{\gamma_1 - \gamma_2 + 2}{2\sigma}} \left(\frac{\gamma_1 - \gamma_2}{2} + 1 - \sigma z\right) \cdot \varphi(z) dz$$
$$= \frac{(\gamma_1 - \gamma_2 + 2)}{2} \cdot \Phi\left(\frac{\gamma_1 - \gamma_2 + 2}{2\sigma}\right) + \frac{\sigma}{\sqrt{2\pi}} \cdot \exp\left(-\frac{(\gamma_1 - \gamma_2 + 2)^2}{8\sigma^2}\right),$$

whereas as for the clean loss minimizer conditioned on $w = -1$, we have

$$L(h_{-1,b_c^*(-1)};\mu_c) = \frac{1}{2}\int_{\frac{\gamma_1-\gamma_2-2}{2\sigma}}^{\infty}\left(\frac{\gamma_2-\gamma_1}{2}+1+\sigma z\right)\cdot\varphi(z)dz$$

$$+\frac{1}{2}\int_{-\infty}^{\frac{\gamma_2-\gamma_1+2}{2\sigma}}\left(\frac{\gamma_2-\gamma_1}{2}+1-\sigma z\right)\cdot\varphi(z)dz$$

$$=\frac{(\gamma_2-\gamma_1+2)}{2}\cdot\Phi\left(\frac{\gamma_2-\gamma_1+2}{2\sigma}\right)+\frac{\sigma}{\sqrt{2\pi}}\cdot\exp\left(-\frac{(\gamma_2-\gamma_1+2)^2}{8\sigma^2}\right).$$

Let $f(t) = t \cdot \Phi(\frac{t}{\sigma}) + \frac{\sigma}{\sqrt{2\pi}} \cdot \exp(-\frac{t^2}{2\sigma^2})$, we know $L(h_{1,b_c^*(1)};\mu_c) = f(\frac{\gamma_1-\gamma_2+2}{2})$ and $L(h_{-1,b_c^*(-1)};\mu_c) = f(\frac{\gamma_2-\gamma_1+2}{2})$. We can compute the derivative of $f(t)$: $f'(t) = \Phi(\frac{t}{\sigma}) \geq 0$, which suggests that $L(h_{1,b_c^*(1)};\mu_c) \leq L(h_{-1,b_c^*(-1)};\mu_c)$.

Now we are ready to prove Lemma B.1.

*Proof of Lemma B.1.* First, we prove the following claim: for any possible $b_p$, linear hypothesis $h_{1,b_p}$ can always be achieved by minimizing the population hinge loss with respect to $\mu_c$ and $\mu_p = \nu_\alpha$ with some carefully-chosen $\alpha \in [0,1]$ based on $b_p$.

For any $\mu_p \in \mathcal{Q}(u)$, according to the first-order optimality condition with respect to $b_p$, we have

$$\frac{\partial}{\partial b}L(h_{1,b_p};\mu_c) = -\epsilon \cdot \frac{\partial}{\partial b}L(h_{1,b_p};\mu_p) = -\epsilon \cdot \frac{\partial}{\partial b}\mathbb{E}_{(x,y)\sim\mu_p}\left[\ell(h_{1,b_p};\mu_p)\right] \in [-\epsilon,\epsilon], \qquad (13)$$

where the last inequality follows from $\frac{\partial}{\partial b}\ell(h_{w,b};x,y) \in [-1,1]$ for any $(x,y)$. Let $\mathcal{B}_p$ be the set of any possible bias parameters $b_p$. According to (13), we have

$$\mathcal{B}_p = \left\{b \in \mathbb{R} : \frac{\partial}{\partial b}L(h_{1,b};\mu_c) \in [-\epsilon,\epsilon]\right\}.$$

Let $b_c^*(1) = \text{argmin}_{b\in\mathbb{R}} L(h_{1,b};\mu_c)$ be the clean loss minimizer conditioned on $w = 1$. According to Lemma C.1 and the assumption $|\gamma_1 + \gamma_2| \leq 2u$, we know $b_c^*(1) = \frac{\gamma_1+\gamma_2}{2} \in [-u,u]$. For any $b_p \in \mathcal{B}_p$, we can always choose

$$\alpha = \frac{1}{2} + \frac{1}{2\epsilon} \cdot \frac{\partial}{\partial b}L(h_{1,b_p};\mu_c) \in [0,1], \qquad (14)$$

such that

$$h_{1,b_p} = \underset{b\in\mathbb{R}}{\text{argmin}}[L(h_{1,b};\mu_c) + \epsilon \cdot L(h_{1,b};\nu_\alpha)],$$

where $\nu_\alpha$ is defined according to (3). This follows from the first-order optimality condition for convex function and the closed-form solution for the derivative of hinge loss with respect to $\nu_\alpha$:

$$\frac{\partial}{\partial b}L(h_{1,b_p};\nu_\alpha) = \alpha \cdot \frac{\partial}{\partial b}\ell(h_{+1,b_p};-u,+1) + (1-\alpha) \cdot \frac{\partial}{\partial b}\ell(h_{+1,b_p};u,-1) = 1 - 2\alpha.$$

Thus, we have proven the claimed presented at the beginning of the proof of Lemma B.1.

Next, we show that for any $b_p \in \mathcal{B}_p$, among all the possible choices of poisoned distribution $\mu_p$ that induces $b_p$, choosing $\mu_p = \nu_\alpha$ with $\alpha$ defined according to (14) is the optimal choice in terms of the maximization objective in (10). Let $\mu_p \in \mathcal{Q}(u)$ be any poisoned distribution that satisfies the following condition:

$$b_p = \underset{b\in\mathbb{R}}{\text{argmin}}[L(h_{1,b};\mu_c) + \epsilon \cdot L(h_{1,b};\mu_p)].$$

According to the aforementioned analysis, we know that by setting $\alpha$ according to (14), $\nu_\alpha$ also yields $b_p$. Namely,

$$b_p = \underset{b\in\mathbb{R}}{\text{argmin}}[L(h_{1,b};\mu_c) + \epsilon \cdot L(h_{1,b};\nu_\alpha)].$$

Since the population losses with respect to $\mu_c$ are the same at the induced bias $b = b_p$, it remains to prove $\nu_\alpha$ achieves a larger population loss with respect to the poisoned distribution than that of $\mu_p$, i.e., $L(h_{1,b_p};\nu_\alpha) \geq L(h_{1,b_p};\mu_p)$.

Consider the following two probabilities with respect to $b_p$ and $\mu_p$:

$$p_1 = \mathbb{P}_{(x,y)\sim\mu_p}\left[\frac{\partial}{\partial b}\ell(h_{1,b_p}; x, y) = -1\right], \quad p_2 = \mathbb{P}_{(x,y)\sim\mu_p}\left[\frac{\partial}{\partial b}\ell(h_{1,b_p}; x, y) = 1\right].$$

Note that the derivative of hinge loss with respect to the bias parameter is $\frac{\partial}{\partial b}\ell(h_{w,b}; x, y) \in \{-1, 0, 1\}$, thus we have

$$\mathbb{P}_{(x,y)\sim\mu_p}\left[\frac{\partial}{\partial b}\ell(h_{1,b_p}; x, y) = 0\right] = 1 - (p_1 + p_2).$$

Moreover, according to the first-order optimality of $b_p$ with respect to $\mu_p$, we have

$$\frac{\partial}{\partial b}L(h_{1,b_p}; \mu_c) = -\epsilon \cdot \frac{\partial}{\partial b}L(h_{1,b_p}; \mu_p) = \epsilon \cdot (p_1 - p_2),$$

If we measure the sum of the probability of input having negative gradient and half of the probability of having zero gradient, we have:

$$p_1 + \frac{1 - (p_1 + p_2)}{2} = \frac{1}{2} + \frac{p_1 - p_2}{2} = \frac{1}{2} + \frac{1}{2\epsilon} \cdot \frac{\partial}{\partial b}L(h_{1,b_p}; \mu_c) = \alpha.$$

Therefore, we can construct a mapping $g$ that maps $\mu_p$ to $\nu_\alpha$: by moving any $(x, y) \sim \mu_p$ that contributes $p_1$ (negative derivative) and any $(x, y) \sim \mu_p$ that contributes $p_2$ (positive derivative) to extreme locations $(-u, +1)$ and $(u, -1)$, respectively, and move the remaining $(x, y)$ that has zero derivative to $(-u, +1)$ and $(u, -1)$ with equal probabilities (i.e., $\frac{1-p_1-p_2}{2}$), and we can easily verify that the gradient of $b_p$ with respect to $\mu_p$ is the same as $\nu_\alpha$.

In addition, note that hinge loss is monotonically increasing with respect to the $\ell_2$ distance of misclassified examples to the decision hyperplane, and the initial clean loss minimizer $b_c^*(1) \in [-u, u]$, we can verify that the constructed mapping $g$ will not reduce the individual hinge loss. Namely, $\ell(h_{1,b_p}; x, y) \leq \ell(h_{1,b_p}; g(x, y))$ holds for any $(x, y) \sim \mu_p$. Therefore, we have proven Lemma B.1. $\qquad\square$

## C.2   Proof of Lemma B.2

*Proof of Lemma B.2.* First, we introduce the following notations. For any $\mu_p \in \mathcal{Q}(u)$ and any $w \in \{-1, 1\}$, let

$$b_c^*(w) = \underset{b\in\mathbb{R}}{\mathrm{argmin}}\, L(h_{w,b}; \mu_c), \quad b_p(w; \mu_p) = \underset{b\in\mathbb{R}}{\mathrm{argmin}}[L(h_{w,b}; \mu_c) + \epsilon \cdot L(h_{w,b}; \mu_p)].$$

According to Lemma B.1, we know that the maximum population hinge loss conditioned on $w = 1$ is achieved when $\mu_p = \nu_\alpha$ for some $\alpha \in [0, 1]$. To prove the sufficient and necessary condition specified in Lemma B.2, we also need to consider $w = -1$. Note that different from $w = 1$, we want to specify the minimum loss that can be achieved with some $\mu_p$ for $w = -1$. For any $\mu_p \in \mathcal{Q}(u)$, we have

$$L(h_{-1,b_p(-1;\mu_p)}; \mu_c) + \epsilon \cdot L(h_{-1,b_p(-1;\mu_p)}; \mu_p) \geq \min_{b\in\mathbb{R}} L(h_{-1,b}; \mu_c) = L(h_{-1,b_c^*(-1)}; \mu_c). \quad (15)$$

According to Lemma C.1, we know $b_c^*(-1) = \frac{\gamma_1+\gamma_2}{2}$, which achieves the minimum clean loss conditioned on $w = -1$. Since we assume $\frac{\gamma_1+\gamma_2}{2} \in [-u + 1, u - 1]$, according to the first-order optimality condition, the equality in (15) can be attained as long as $\mu_p$ only consists of correctly classified data that also incurs zero hinge loss with respect to $b_c^*(-1)$ (not all correctly classified instances incur zero hinge loss). It can be easily checked that choosing $\mu_p = \nu_\alpha$ based on (3) with any $\alpha \in [0, 1]$ satisfies this condition, which suggests that as long as the poisoned distribution $\mu_p$ is given in the form of $\nu_\alpha$ and if the $w = -1$ is achievable (conditions on when this can be achieved will be discussed shortly), then the bias term that minimizes the distributional loss is equal to $b_c^*(-1)$, and is the minimum compared to other choices of $b_p(-1; \mu_p)$. According to Lemma B.1, it further implies the following statement: there exists some $\alpha \in [0, 1]$ such that

$$\nu_\alpha \in \underset{\mu_p\in\mathcal{Q}(u)}{\mathrm{argmax}}\left\{[L(h_{1,b_p(1;\mu_p)}; \mu_c) + \epsilon \cdot L(h_{1,b_p(1;\mu_p)}; \mu_p)]\right.$$

$$\left. - [L(h_{-1,b_p(-1;\mu_p)}; \mu_c) + \epsilon \cdot L(h_{-1,b_p(-1;\mu_p)}; \mu_p)]\right\}.$$

For simplicity, let us denote by $\Delta L(\mu_p; \epsilon, u, \mu_c)$ the maximization objective regarding the population loss difference between $w = 1$ and $w = -1$. Thus, a necessary and sufficient condition such that there exists a $h_{-1,b_p(-1;\mu_p)}$ as the loss minimizer is that $\max_{\alpha \in [0,1]} \Delta L(\nu_\alpha; \epsilon, u, \mu_c) \geq 0$. This requires us to characterize the maximal value of loss difference for any possible configurations of $\epsilon, u$ and $\mu_c$. According to Lemma C.1 and the definition of $\nu_\alpha$, for any $\alpha \in [0, 1]$, we denote the above loss difference as

$$\Delta L(\nu_\alpha; \epsilon, u, \mu_c) = \underbrace{L(h_{1,b_p(1;\nu_\alpha)}; \mu_c) + \epsilon \cdot L(h_{1,b_p(1;\nu_\alpha)}; \nu_\alpha)}_{I_1} - \underbrace{L(h_{-1,b_c^*(-1)}; \mu_c)}_{I_2}.$$

The second term $I_2$ is fixed (and the loss on $\nu_\alpha$ is zero conditioned on $w = -1$), thus it remains to characterize the maximum value of $I_1$ with respect to $\alpha$ for different configurations. Consider auxiliary function

$$g(b) = \frac{1}{2}\Phi\left(\frac{b + \gamma_1 + 1}{\sigma}\right) - \frac{1}{2}\Phi\left(\frac{-b - \gamma_2 + 1}{\sigma}\right).$$

We know $g(b) \in [-\frac{1}{2}, \frac{1}{2}]$ is a monotonically increasing function by checking with derivative to $b$. Let $g^{-1}$ be the inverse function of $g$. Note that according to Lemma C.1 and the first-order optimality condition of $b_p(1; \nu_\alpha)$, we have

$$\frac{\partial}{\partial b}L(h_{+1,b}; \mu_c)\Big|_{b=b_p(1;\nu_\alpha)} = g\big(b_p(+1; \nu_\alpha)\big) = -\epsilon \cdot \frac{\partial}{\partial b}L(h_{+1,b_p(1;\nu_\alpha)}; \nu_\alpha) = \epsilon \cdot (2\alpha - 1), \quad (16)$$

where the first equality follows from Lemma C.1, the second equality follows from the first-order optimality condition and the last equality is based on the definition of $\nu_\alpha$. This suggests that $b_p(1; \nu_\alpha) = g^{-1}\big(\epsilon \cdot (2\alpha - 1)\big)$ for any $\alpha \in [0, 1]$.

Consider the following two configurations for the term $I_1$: $0 \notin [g^{-1}(-\epsilon), g^{-1}(\epsilon)]$ and $0 \in [g^{-1}(-\epsilon), g^{-1}(\epsilon)]$. Consider the first configuration, which is also equivalent to $g(0) \notin [-\epsilon, \epsilon]$. We can prove that if $\gamma_1 + \gamma_2 < 0$ meaning that $b_c^*(1) > 0$, choosing $\alpha = 0$ achieves the maximal value of $I_1$; whereas if $\gamma_1 + \gamma_2 > 0$, choosing $\alpha = 1$ achieves the maximum. Note that it is not possible for $\gamma_1 + \gamma_2 = 0$ under this scenario. The proof is straightforward, since we have

$$I_1 = L(h_{1,g^{-1}(2\epsilon\alpha-\epsilon)}; \mu_c) + \epsilon \cdot L(h_{1,g^{-1}(2\epsilon\alpha-\epsilon)}; \nu_\alpha)$$
$$= L(h_{1,g^{-1}(2\epsilon\alpha-\epsilon)}; \mu_c) + \epsilon \cdot \big[1 + u + (1 - 2\alpha) \cdot g^{-1}(2\epsilon\alpha - \epsilon)\big]$$
$$= L(h_{1,t}; \mu_c) + \epsilon \cdot (1 + u) - t \cdot g(t),$$

where $t = g^{-1}(2\epsilon\alpha - \epsilon) \in [g^{-1}(\epsilon), g^{-1}(\epsilon)]$. In addition, we can compute the derivative of $I_1$ with respect to $t$:

$$\frac{\partial}{\partial t}I_1 = g(t) - g(t) - t \cdot g'(t) = -t \cdot g'(t),$$

which suggests that $I_1$ is a concave function with respect to $t$. If $0 \in [g^{-1}(-\epsilon), g^{-1}(\epsilon)]$, we achieve the global maximum of $I_1$ at $t = 0$ by carefully picking $\alpha_0 = \frac{1}{2} + \frac{1}{2\epsilon} \cdot g(0)$. If not (i.e., $g^{-1}(-\epsilon) > 0$ or $g^{-1}(\epsilon) < 0$), then we pick $t$ that is closer to 0, which is either $g(-\epsilon)$ or $g(\epsilon)$ by setting $\alpha = 0$ or $\alpha = 1$ respectively. Therefore, we can specify the sufficient and necessary conditions when the weight vector $w$ can be flipped from 1 to $-1$:

1. When $g(0) \notin [-\epsilon, \epsilon]$, the condition is

$$\max\{\Delta L(\nu_0; \epsilon, u, \mu_c), \Delta L(\nu_1; \epsilon, u, \mu_c)\} \geq 0.$$

2. When $g(0) \in [-\epsilon, \epsilon]$, the condition is

$$\Delta L(\nu_{\alpha_0}; \epsilon, u, \mu_c) \geq 0, \text{ where } \alpha_0 = \frac{1}{2} + \frac{1}{2\epsilon} \cdot g(0).$$

Plugging in the definition of $g$ and $\Delta L$, we complete the proof of Lemma B.2. $\qquad\square$

### C.3 Proof of Lemma B.3

*Proof of Lemma B.3.* Let $\mu_1, \mu_2$ be the probability measures of the positive and negative examples assumed in (2), respectively. Let $\varphi(z; \gamma, \sigma)$ be the PDF of Gaussian distribution $\mathcal{N}(\gamma, \sigma^2)$. For simplicity, we simply write $\varphi(z) = \varphi(z; 0, 1)$ for standard Gaussian. For any $h_{w,b} \in \mathcal{H}_L$, we know

$w$ can be either 1 or $-1$. First, let's consider the case where $w = 1$. According to the definition of risk and the data generating process of $\mu_c$, we have

$$
\begin{aligned}
\text{Risk}(h_{w,b}; \mu_c) &= p \cdot \text{Risk}(h_{w,b}; \mu_1) + (1 - p) \cdot \text{Risk}(h_{w,b}; \mu_2) \\
&= p \cdot \int_{-b}^{\infty} \varphi(z; \gamma_1, \sigma_1) dz + (1 - p) \cdot \int_{-\infty}^{-b} \varphi(z; \gamma_2, \sigma_2) dz \\
&= p \cdot \int_{\frac{-b-\gamma_1}{\sigma_1}}^{\infty} \varphi(z) dz + (1 - p) \cdot \int_{-\infty}^{\frac{-b-\gamma_2}{\sigma_2}} \varphi(z) dz \\
&= p \cdot \Phi\left(\frac{b + \gamma_1}{\sigma_1}\right) + (1 - p) \cdot \Phi\left(\frac{-b - \gamma_2}{\sigma_2}\right).
\end{aligned}
$$

Similarly, when $w = -1$, we have

$$
\begin{aligned}
\text{Risk}(h_{w,b}; \mu_c) &= p \cdot \int_{-\infty}^{b} \varphi(z; \gamma_1, \sigma_1) dz + (1 - p) \cdot \int_{b}^{\infty} \varphi(z; \gamma_2, \sigma_2) dz \\
&= p \cdot \int_{-\infty}^{\frac{b-\gamma_1}{\sigma_1}} \varphi(z) dz + (1 - p) \cdot \int_{\frac{b-\gamma_2}{\sigma_2}}^{\infty} \varphi(z) dz \\
&= p \cdot \Phi\left(\frac{b - \gamma_1}{\sigma_1}\right) + (1 - p) \cdot \Phi\left(\frac{-b + \gamma_2}{\sigma_2}\right).
\end{aligned}
$$

Combining the two cases, we complete the proof. $\qquad\square$

### C.4 Proof of Lemma C.1

*Proof of Lemma C.1.* We use similar notations such as $\mu_1$, $\mu_2$ and $\varphi$ as in Lemma B.3. For any $h_{w,b} \in \mathcal{H}_L$ with $w = 1$, then according to the definition of population hinge loss, we have

$$
\begin{aligned}
&L(h_{w,b}; \mu_c) \\
&= \mathbb{E}_{(x,y) \sim \mu_c}\left[\max\{0, 1 - y(x + b)\}\right] \\
&= p \int_{-b-1}^{\infty} (1 + b + z)\varphi(z; \gamma_1, \sigma_1) dz + (1 - p) \int_{-\infty}^{-b+1} (1 - b - z)\varphi(z; \gamma_2, \sigma_2) dz \\
&= p \int_{\frac{-b-1-\gamma_1}{\sigma_1}}^{\infty} (1 + b + \gamma_1 + \sigma_1 z)\varphi(z) dz + (1 - p) \int_{-\infty}^{\frac{-b+1-\gamma_2}{\sigma_2}} (1 - b - \gamma_2 - \sigma_2 z)\varphi(z) dz \\
&= p(b + \gamma_1 + 1)\Phi\left(\frac{b + \gamma_1 + 1}{\sigma_1}\right) + p\sigma_1 \frac{1}{\sqrt{2\pi}} \exp\left(-\frac{(b + \gamma_1 + 1)^2}{2\sigma_1^2}\right) \\
&\quad + (1 - p)(-b - \gamma_2 + 1)\Phi\left(\frac{-b - \gamma_2 + 1}{\sigma_2}\right) + (1 - p)\sigma_2 \frac{1}{\sqrt{2\pi}} \exp\left(-\frac{(-b - \gamma_2 + 1)^2}{2\sigma_2^2}\right).
\end{aligned}
$$

Taking the derivative with respect to parameter $b$ and using simple algebra, we have

$$
\frac{\partial}{\partial b} L(h_{w,b}; \mu_c) = p \cdot \Phi\left(\frac{b + \gamma_1 + 1}{\sigma_1}\right) - (1 - p) \cdot \Phi\left(\frac{-b - \gamma_2 + 1}{\sigma_2}\right).
$$

Similarly, for any $h_{w,b} \in \mathcal{H}_L$ with $w = -1$, we have

$$L(h_{w,b}; \mu_c)$$
$$= \mathbb{E}_{(x,y) \sim \mu_c}\big[\max\{0, 1 - y(-x + b)\}\big]$$
$$= p \cdot \int_{-\infty}^{b+1} (1 + b - z)\varphi(z; \gamma_1, \sigma_1)dz + (1 - p) \cdot \int_{b-1}^{\infty} (1 - b + z)\varphi(z; \gamma_2, \sigma_2)dz$$
$$= p \cdot \int_{-\infty}^{\frac{b+1-\gamma_1}{\sigma_1}} (1 + b - \gamma_1 - \sigma_1 z)\varphi(z)dz + (1 - p) \cdot \int_{\frac{b-1-\gamma_2}{\sigma_2}}^{\infty} (1 - b + \gamma_2 + \sigma_2 z)\varphi(z)dz$$
$$= p(b - \gamma_1 + 1)\Phi\left(\frac{b - \gamma_1 + 1}{\sigma_1}\right) + p\sigma_1 \frac{1}{\sqrt{2\pi}} \exp\left(-\frac{(b - \gamma_1 + 1)^2}{2\sigma_1^2}\right)$$
$$+ (1 - p)(-b + \gamma_2 + 1)\Phi\left(\frac{-b + \gamma_2 + 1}{\sigma_2}\right) + (1 - p)\sigma_2 \frac{1}{\sqrt{2\pi}} \exp\left(-\frac{(-b + \gamma_2 + 1)^2}{2\sigma_2^2}\right).$$

Taking the derivative, we have

$$\frac{\partial}{\partial b}L(h_{w,b}; \mu_c) = p \cdot \Phi\left(\frac{b - \gamma_1 + 1}{\sigma_1}\right) - (1 - p) \cdot \Phi\left(\frac{-b + \gamma_2 + 1}{\sigma_2}\right).$$

Combining the two scenarios, we complete the proof. $\qquad\square$

## D  Additional Experimental Results and Details

In this section, we provide details on our experimental setup (Appendix D.1) and then provide additional results (Appendix D.2).

### D.1  Details on Experimental Setup in Section 3

**Details on datasets and training configurations.** In the main paper, we used different public benchmark datasets including MNIST [27] digit pairs (i.e., 1–7. 6–9, 4–9) and also the Enron dataset, which is created by Metsis et al. [36], Dogfish [24] and Adult [14], which are all used in the evaluations of prior works except MNIST 6–9 and MNIST 4–9. In the appendix, we additionally present the results of the IMDB dataset [33], which has also been used in prior evaluations [24, 25]. We did not include the IMDB results in the main paper because we could not run the existing state-of-the-art poisoning attacks on IMDB because the computation time is extremely slow. Instead, we directly quote the poisoned error of SVM from Koh et al. [25] and then present the computed metrics. For the Dogfish and Enron dataset, we construct the constraint set $\mathcal{C}$ in the no defense setting by finding the minimum ($u_{\min}^i$) and maximum ($u_{\max}^i$) values occurred in each feature dimension $i$ for both the training and test data, which then forms a box constraint $[u_{\min}^i, u_{\max}^i]$ for each dimension. This way of construction is also used in the prior work [25]. Because we consider linear models, the training [38] of linear models and the attacks on them are stable (i.e., less randomness involved in the process) and so, we get almost identical results when feeding different random seeds. Therefore, we did not report error bars in the results. The regularization parameter $\lambda$ for training the linear models (SVM and LR) are configured as follows: $\lambda = 0.09$ for MNIST digit pairs, Adult, Dogfish, SVM for Enron; $\lambda = 0.01$ for IMDB, LR for Enron. Overall, the results and conclusions in this paper are insensitive to the choice of $\lambda$. The computation of the metrics in this paper are extremely fast and can be done on any laptop. The poisoning attacks can also be done on a laptop, except the Influence Attack [25], whose computation can be accelerated using GPUs.

**Attack details.** The KKT, MTA and Min-Max attacks evaluated in Section 3 require a target model as input. This target model is typically generated using some label-flipping heuristics [25, 46]. In practice, these attacks are first run on a set of carefully-chosen target models, then the best attack performance achieved by these target models is reported. We generate target models using the improved procedure described in Suya et al. [46] because their method is able to generate better target models that induce victim models with a higher test error compared with the method proposed in Koh et al. [25]. We generate target models with different error rates, ranging from $5\%$ to $70\%$ using the label-flipping heuristics, and then pick the best performing attack induced by these target models.

| Poison Ratio $\epsilon$ (%) | 0.0 | 0.1 | 0.2 | 0.3 | 0.5 | 0.7 | 0.9 | 1.0 | 2.0 | 3.0 |
|---|---|---|---|---|---|---|---|---|---|---|
| Train Error (%) | 0.1 | 0.8 | 1.2 | 1.8 | 2.6 | 3.1 | 3.3 | 3.6 | 5.3 | 6.5 |
| Test Error (%) | 0.8 | 9.5 | 12.8 | 13.8 | 17.8 | 20.5 | 21.0 | 20.5 | 27.3 | 31.8 |

Table 2: Comparisons of training and test errors of existing data poisoning attacks on Dogfish. The poisoned training and test errors are reported from the current best attacks.

| | | **Robust** | | | **Moderately Vulnerable** | | **Highly Vulnerable** | | |
|---|---|---|---|---|---|---|---|---|---|
| | Metric | MNIST 6–9 | MNIST 1–7 | Adult | Dogfish | MNIST 4–9 | F. Enron | Enron | IMDB |
| SVM | Error Increase | 2.7 | 2.4 | 3.2 | 7.9 | 6.6 | 33.1 | 31.9 | 19.1[†] |
| | Base Error | 0.3 | 1.2 | 21.5 | 0.8 | 4.3 | 0.2 | 2.9 | 11.9 |
| | Sep/SD | 6.92 | 6.25 | 9.65 | 5.14 | 4.44 | 1.18 | 1.18 | 2.57 |
| | Sep/Size | 0.24 | 0.23 | 0.33 | 0.05 | 0.14 | 0.01 | 0.01 | 0.002 |
| LR | Error Increase | 2.3 | 1.8 | 2.5 | 6.8 | 5.8 | 33.0 | 33.1 | - |
| | Base Error | 0.6 | 2.2 | 20.1 | 1.7 | 5.1 | 0.3 | 2.5 | - |
| | Sep/SD | 6.28 | 6.13 | 4.62 | 5.03 | 4.31 | 1.11 | 1.10 | 2.52 |
| | Sep/Size | 0.27 | 0.27 | 0.27 | 0.09 | 0.16 | 0.01 | 0.01 | 0.003 |

Table 3: Explaining disparate poisoning vulnerability under linear models by computing the metrics on the correctly classified clean test points. The top row for each model gives the increase in error rate due to the poisoning, over the base error rate in the second row. The error increase of IMDB (marked with [†]) is directly quoted from Koh et al. [25] as running the existing poisoning attacks on IMDB is extremely slow. LR results are missing as they are not contained in the original paper. The explanatory metrics are the scaled (projected) separability, standard deviation and constraint size.

Following the prior practice [25], we consider adversaries that have access to both the clean training and test data, and therefore, adversaries can design attacks that can perform better on the test data. This generally holds true for the Enron and MNIST 1–7 datasets, but for Dogfish, we find in our experiments that the attack "overfits" to the test data heavily due to the small number of training and test data and also the high dimensionality. More specifically, we find that the poisoned model tends to incur significantly higher error rates on the clean test data compared to the clean training data. Since this high error cannot fully reflect the distributional risk, when we report the results in Section 3 we report the errors on both the training and the testing data to give a better empirical sense of what the distributional risk may look like. This also emphasizes the need to be cautious on the potential for "overfitting" behavior when designing poisoning attacks. Table 2 shows the drastic differences between the errors of the clean training and test data after poisoning.

## D.2 Supplemental Results

In this section, we provide additional results results that did not fit into the main paper, but further support the observations and claims made in the main paper. We first show the results of IMDB and the metrics computed on the whole clean test data in Appendix D.2.2 to complement Table 1 in the main paper, then include the complete results of the impact of data sanitization defenses in Appendix D.2.3 to complement the last paragraph in Section 7. Next, we provide the metrics computed on selective benchmark datasets using a different projection vector from the clean model weight in Appendix D.2.4 to support the results in Table 1 in the main paper. Lastly, we show the performance of different poisoning attacks at various poisoning ratios in Appendix D.2.5, complementing Figure 1 in the main paper.

## D.2.1 IMDB results

Table 1 in the main paper presents the metrics that are computed on the correctly classified test samples by the clean model $w_c$. In Table 3, we additionally include the IMDB results to the Table 1 in the main paper. From the table, we can see that IMDB is still highly vulnerable to poisoning because its separability is low compared to datasets that are moderately vulnerable or robust, and impacted the most by the poisoning points compared to all other benchmark datasets. Note that, the increased error from IMDB is directly quoted from Koh et al. [25], which considers data sanitization

| Models | Metrics | Robust | | | Moderately Vulnerable | | Highly Vulnerable | | |
|--------|---------|--------|---------|-------|-----------------------|---------|-----------|-------|------|
| | | MNIST 6-9 | MNIST 1–7 | Adult | Dogfish | MNIST 4–9 | F. Enron | Enron | IMDB |
| SVM | Error Increase | 2.7 | 2.4 | 3.2 | 7.9 | 6.6 | 33.1 | 31.9 | 19.1[†] |
| | Base Error | 0.3 | 1.2 | 21.5 | 0.8 | 4.3 | 0.2 | 2.9 | 11.9 |
| | Sep/SD | 6.70 | 5.58 | 1.45 | 4.94 | 3.71 | 1.18 | 1.15 | 1.95 |
| | Sep/Size | 0.23 | 0.23 | 0.18 | 0.05 | 0.13 | 0.01 | 0.01 | 0.001 |
| LR | Error Increase | 2.3 | 1.8 | 2.5 | 6.8 | 5.8 | 33.0 | 33.1 | - |
| | Base Error | 0.6 | 2.2 | 20.1 | 1.7 | 5.1 | 0.3 | 2.5 | - |
| | Sep/SD | 5.97 | 5.17 | 1.64 | 4.67 | 3.51 | 1.06 | 1.01 | 1.88 |
| | Sep/Size | 0.26 | 0.26 | 0.16 | 0.08 | 0.15 | 0.01 | 0.01 | 0.002 |

Table 4: Explaining the different vulnerabilities of benchmark datasets under linear models by computing metrics on the whole data. The error increase of IMDB (marked with [†]) is directly quoted from Koh et al. [25].

| Dataset | Error Increase | | Base Error | | Sep/SD | | Sep/Size | |
|---------|------|------|------|------|------|------|------|------|
| | w/o | w/ | w/o | w/ | w/o | w/ | w/o | w/ |
| MNIST 1–7 (10%) | 7.7 | 1.0 | 1.2 | 2.4 | 6.25 | 6.25 | 0.23 | 0.43 |
| Dogfish (3%) | 8.7 | 4.3 | 0.8 | 1.2 | 5.14 | 5.14 | 0.05 | 0.12 |
| Enron (3%) | 31.9 | 25.6 | 2.9 | 3.2 | 1.18 | 1.18 | 0.01 | 0.11 |

Table 5: Understanding impact of data sanitization defenses on poisoning attacks. *w/o* and *w/* denote *without defense* and *with defense* respectively. MNIST 1–7 is evaluated at 10% poisoning ratio due to its strong robustness at $\epsilon = 3\%$ and Enron is still evaluated at $\epsilon = 3\%$ because it is highly vulnerable.

defenses. Therefore, we expect the attack effectiveness might be further improved when we do not consider any defenses, as in our paper.

### D.2.2 Metrics computed using all test data

Table 4 shows the results when the metrics are computed on the full test data set (including misclassified ones), rather than just on examples that were classified correctly by the clean model. The metrics are mostly similar to Table 3 when the initial errors are not high. For datasets with high initial error such as Adult, the computed metrics are more aligned with the final poisoned error, not the error increase.

### D.2.3 Explaining the impact of data sanitization defenses

We then provide additional results for explaining the effectiveness of the data sanitization defenses in improving dataset robustness, which is discussed in section 7. On top of the Enron result shown in the paper, which is attacked at 3% poisoning ratio, we also provide attack results of MNIST 1-7 dataset. We report the attack results when $\epsilon = 10\%$ and we considered a significantly higher poisoning ratio because at the original 3% poisoning, the dataset can well resist existing attacks and hence there is no point in protecting the dataset with sanitization defenses. This attack setting is just for illustration purpose and attackers in practice may be able to manipulate such a high number of poisoning points. Following the main result in the paper, we still compute the metrics based on the correctly classified samples in the clean test set, so as to better depict the relationship between the increased errors and the computed metrics. The results are summarized in Table 5 and we can see that existing data sanitization defenses improve the robustness to poisoning by majorly limiting $\mathrm{Size}_{\boldsymbol{w}_c}(\mathcal{C})$. For MNIST 1-7, equipping with data sanitization defense will make the dataset even more robust (robust even at the high 10% poisoning rate), which is consistent with the findings in prior work [45].

### D.2.4 Using different projection vectors

In the main paper, we used the weight of the clean model as the projection vector and found that the computed metrics are highly correlated with the empirical attack effectiveness observed for different benchmark datasets. However, there can also be other projection vectors that can be used for explaining the different vulnerabilities, as mentioned in Remark 5.8.

|            | Base Error (%) | Error Increase (%) | Sep/SD | | Sep/Size | |
|------------|----------------|--------------------|--------|---------|----------|---------|
|            |                |                    | $w_c$  | $w_U$   | $w_c$    | $w_U$   |
| MNIST 1–7  | 1.2            | 2.4                | 6.25   | 6.51    | 0.23     | 0.52    |
| Dogfish    | 0.8            | 7.9                | 5.14   | 4.43    | 0.05     | 0.19    |

Table 6: Using the projection vector that minimizes the upper bound on the risk of optimal poisoning attacks for general distributions. $w_c$ denotes the clean weight vector and $w_U$ denotes weight vector obtained from minimizing the upper bound.

| Metric          | MNIST 1–7 | Dogfish |
|-----------------|-----------|---------|
| Upper Bound (%) | 20.1      | 34.9    |
| Lower Bound (%) | 3.6       | 8.7     |
| Base Error (%)  | 1.2       | 0.8     |

Table 7: Non-trivial upper bounds on the limits of poisoning attacks across benchmark datasets on linear SVM. The *Upper Bound* on the limits of poisoning attacks is obtained by minimizing the upper bound Equation (17). The *Lower Bound* on the limit of poisoning attacks is obtained by reporting the highest poisoned error from the best empirical attacks. The datasets are sorted based on the lowest and highest empirical base error.

We conducted experiments that use the projection vector that minimizes the upper bound on optimal poisoning attacks, given in Equation 5. The upper-bound minimization corresponds to a min-max optimization problem. We solve it using the online gradient descent algorithm (alternatively updating the poisoning points and model weight), adopting an approach similar to the one used by Koh et al. for the i-Min-Max attack [25]. We run the min-max optimization for 30,000 iterations with learning rate of 0.03 for the weight vector update, and pick the weight vector that results in the lowest upper bound in Equation 5.

The results on two of the benchmark datasets, MNIST 1–7 and Dogfish, are summarized in Table 6. From the table, we can see that, compared to the clean model, the new projection vector reduces the projected constraint size (increases Sep/Size), which probably indicates the weight vector obtained from minimizing the upper bound focuses more on minimizing the term $\ell_M(\text{Size}_{w_c}(\mathcal{C}))$ in Equation 5. Nevertheless, projecting onto the new weight vector can still well explain the difference between the benchmark datasets.

Using the obtained vector $w_U$ (denote the corresponding hypothesis as $h_U$) above, we can also compute an (approximate) upper bound on the test error of the optimal indiscriminate poisoning attack, which is proposed by Steinhardt et al. [45] and is shown in (17). The upper bound is an approximate upper bound to the test error because $\max_{\mathcal{S}_p \subseteq \mathcal{C}} L(\hat{h}_p; \mathcal{S}_c)$ only upper bounds the maximum training error (of poisoned model $\hat{h}_p$) on $\mathcal{S}_c$ from poisoning and it is assumed that, with proper regularization, the training and test errors are roughly the same. We can view $h_U$ as the numerical approximation of the hypothesis that leads to the smallest upper bound on the (approximate) test error. Next, in Table 7, we compare the computed (approximate) upper bound and the lower bound on poisoning effectiveness achieved by best known attacks, and we can easily observe that, although the gap between the lower and upper bounds are relatively large (all datasets are vulnerable at 3% poisoning ratio when we directly check the upper bounds), the upper bounds still vary between datasets. We also speculate that the actual performance of the best attacks may not be as high as indicated by the upper bound, because when computing the upper bound, we used the surrogate loss to upper bound the 0-1 loss (exists some non-negligible gap) and maximized the surrogate loss in the poisoning setting, which potentially introduces an even larger gap between the surrogate loss and the 0-1 loss. Hence, we motivate future research to shrink the possibly large gap.

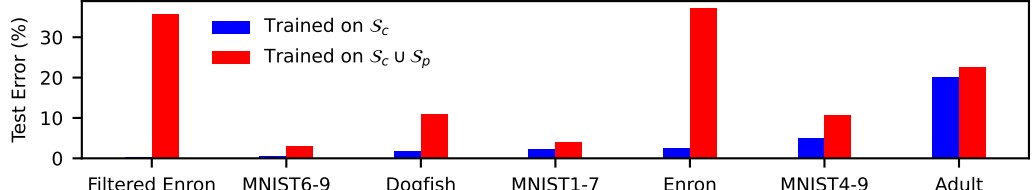

Figure 3: Performance of best current indiscriminate poisoning attacks with $\epsilon = 3\%$ across different benchmark datasets on LR. Datasets are sorted from lowest to highest base error rate.

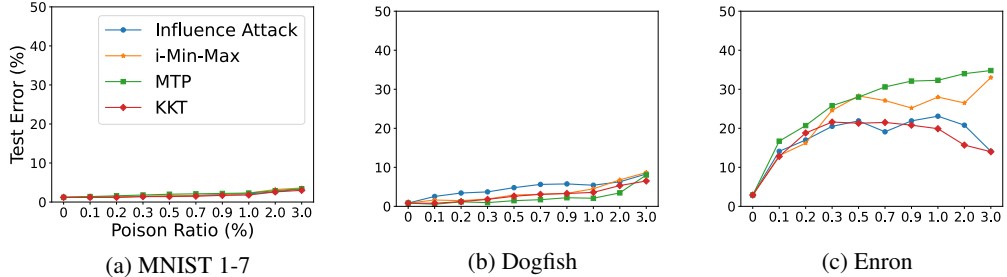

(a) MNIST 1-7                (b) Dogfish                (c) Enron

Figure 4: Comparisons of the attack performance of existing data poisoning attacks on different benchmark datasets. Poisoning ratios are 0.1%, 0.2%, 0.3%, 0.5%, 0.7%, 0.9%, 1%, 2%, 3%.

$$\max_{\mathcal{S}_p \subseteq \mathcal{C}} L(\hat{h}_p; \mathcal{S}_c) \leq \min_{h} \left( L(h; \mathcal{S}_c) + \epsilon \max_{(\boldsymbol{x}_p, y_p) \in \mathcal{C}} \ell(h; \boldsymbol{x}_p, y_p) + \frac{(1+\epsilon)\lambda}{2} \|\boldsymbol{w}\|_2^2 \right)$$

$$\leq L(h_U; \mathcal{S}_c) + \epsilon \max_{(\boldsymbol{x}_p, y_p) \in \mathcal{C}} \ell(h_U; \boldsymbol{x}_p, y_p) + \frac{(1+\epsilon)\lambda}{2} \|\boldsymbol{w}_U\|_2^2 \qquad (17)$$

### D.2.5 Performance of different attacks are similar

In this section, we first present the best attack results on different benchmark datasets on the LR model in Figure 3. Similar to the case of linear SVM as shown in Figure 1, the performance of best known attacks vary drastically across the benchmark datasets. Although we only report the performance of the best known attack from the candidate state-of-the-art poisoning attacks, the difference among these canndidate attacks do not vary much in many settings. In Figure 4 we show the attack performance of different attacks on the selected benchmark datasets of MNIST 1–7, Dogfish, and Enron and the main observation is that different attacks perform mostly similarly for a given dataset (but their performance varies a lot across datasets). Also, from the attack results on Enron (Figure 4c), we can see that several of the attacks perform worse at higher poisoning ratios. Although there is a chance that the attack performance can be improved by careful hyperparameter tuning, it also suggests that these attacks are suboptimal. Optimal poisoning attacks should never get less effective as the poisoning ratio increases, according to Theorem 4.5.

### D.2.6 Possible causal relationship of identified factors to poisoning vulnerability

In this section, we describe our preliminary experiment on showing the possible causal relationship from the identified two factors (i.e., Sep/SD and Sep/Size) to the empirically observed vulnerability against poisoning attacks. Given a dataset, we aim to modify it such that we can gradually change one factor of the dataset while (ideally) keeping the other one fixed, and then measure the changes in its empirical poisoning vulnerability. To cause changes to the factor of "Sep/SD", we need to modify the original data points in the dataset. This modification may make it hard to keep the factor of "Sep/Size" (or "Size") fixed, as the computation of "Size" depends on the clean boundary $\boldsymbol{w}_c$, which is again dependent on the (modified) clean distribution. Therefore, even if we use the same constraint set $\mathcal{C}$ for different modified datasets, the "Size" or "Sep/Size" can be different. We conducted preliminary experiments on MNIST 1–7 and the details are given below.

| Scale Factor $c$ | Error Increase (%) | Sep/Size |
|:---:|:---:|:---:|
| 1.0 | 2.4 | 0.23 |
| 2.0 | 3.6 | 0.12 |
| 3.0 | 4.9 | 0.08 |
| 5.0 | 7.8 | 0.05 |
| 7.0 | 9.8 | 0.03 |
| 10.0 | 15.6 | 0.02 |

Table 8: Impact of scale factor $c$ on poisoning effectiveness for $\mathcal{C}$ in the form of dimension-wise box-constraint as $[0, c]$ $(c > 1)$ for the MNIST 1–7 dataset. Originally, the dimension-wise box constraint is in $[0, 1]$. The Base Error is 1.2% and the Sep/SD factor is 6.25 and these two values will be the same for all settings of $c$ because the support set of the clean distribution is the strict subset of $\mathcal{C}$. Higher value of $c$ implies poisoning points can be more harmful.

| Number of Repetitions $x$ | Base Error (%) | Error Increase (%) | Sep/SD | Sep/Size |
|:---:|:---:|:---:|:---:|:---:|
| 0 | 1.2 | 2.4 | 6.25 | 0.23 |
| 1 | 1.0 | 4.4 | 5.78 | 0.21 |
| 2 | 1.0 | 5.7 | 5.48 | 0.19 |
| 4 | 1.0 | 6.2 | 5.02 | 0.17 |
| 6 | 1.2 | 6.8 | 4.75 | 0.15 |
| 8 | 1.4 | 8.0 | 4.62 | 0.13 |
| 10 | 1.1 | 8.1 | 4.55 | 0.12 |

Table 9: Changing the distribution of the MNIST 1–7 dataset by replacing $x \cdot 3\%$ of correctly classified points farthest from the clean boundary with 3% of correctly classified points (repeated $x$ times) closest to the boundary. Higher value of $x$ denotes that more near boundary points will be present in the modified training data.

**Change Sep/Size.,** In order to measure the impact of Sep/Size without impacting Sep/SD, we can choose $\mathcal{C}$ that covers even larger space so that the resulting constraint set will not filter out any clean points from the original training data, which then implies both the $\boldsymbol{w}_c$ and the corresponding metric of Sep/SD will not change. We performed a hypothetical experiment on MNIST 1–7 where we scale the original box-constraint from $[0, 1]$ to $[0, c], c >= 1$, where $c$ is the scale factor and $c = 1$ denotes the original setting for MNIST 1–7 experiment. The experiments are hypothetical because, in practice, a box-constraint of $[0, c]$ with $c > 1$ will result in invalid images. We experimented with $c = 2, 3, 5, 7, 10$ and the results are summarized in Table 8. From the table, we see that as $c$ becomes larger, the Size increases and correspondingly the Sep/Size decreases. These changes are associated with a gradual increase in the "Error Increase" from poisoning.

**Change Sep/SD.** As mentioned above, in order to change Sep/SD, we need to modify the original data points, and this modification makes it hard for us to keep Sep/Size fixed for MNIST 1–7 even when using the same $\mathcal{C}$. We leave the exploration of better ways to change data distributions with the Sep/Size fixed as future work. To modify the original data points, we first train a linear SVM on the original MNIST 1-7 dataset. Then, we pick 3% of all (training + test data) correctly classified points that are closest to the clean boundary. These points, once repeated $x$ times, replace $x \times 3\%$ of correctly classified training and testing data points that are furthest from the boundary. This process can be viewed as oversampling in the near-boundary region and undersampling in the region far from the boundary, and aim to create a dataset with a low separability ratio. We tested with $x = 1, 2, 4, 6, 8, 10$. We only deal with correctly classified points because we want to ensure the modified dataset still has high clean accuracy. The results are summarized in Table 9 and we can see that as we increase $x$, the Sep/SD gradually decreases, and the attack effectiveness (measured by the

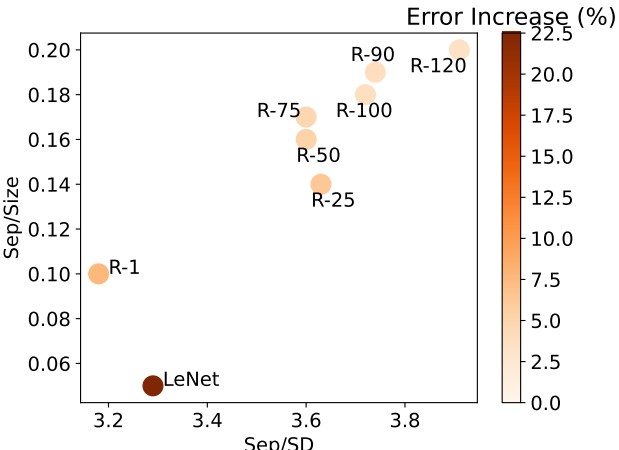

Figure 5: Impact of feature representations on the poisoning effectiveness. The training data of the pretrained model excludes the downstream data of classes "Truck" and "Ship".

| Experiments | Sep/SD vs Error Increase | Sep/Size vs Error Increase |
|---|---|---|
| New | -0.66 (-0.90) | -0.90 (-0.99) |
| Original | -0.98 (-0.97) | -0.98 (-0.97) |

Table 10: Pearson correlation between *Error Increase* and each of *Sep/SD* and *Sep/Size*. The value in the parentheses denotes the Pearson correlation coefficient when we exclude the LeNet structure and only consider ResNet-18 models.

error increase) increases. Interestingly, the Sep/Size also decreases even though it is computed with the same $\mathcal{C}$. Although the preliminary results above are not sufficient to prove a causation claim, they still show the possibility with the empirical evidence on the MNIST 1–7 dataset.

### D.2.7 Evaluating feature extractors with downstream data removed

The experiments in Figure 2 uses the entire CIFAR10 training data to train the pretrained feature extractor, which also includes the data of class "Truck" and "Ship" that are used in the downstream training and analysis. Although our original experiment aims to show how the victim can gain improved resistance to poisoning by using better feature extractors, for the particular experiment in Figure 2, a more realistic setting is to exclude the downstream training data used to train the (upstream) pretrained feature extractors. Therefore, we repeated the same experiment, but now removed the data points from the classes of "Ship" and "Truck" from the upstream training data and then train the (pretrained) models. The results are summarized in Figure 5 and the overall finding is similar to the one in Figure 2 that when we adopt a deep architecture and train it for more epochs, the quality of the obtained feature extractor increases and so, it appears to improve the resistance of downstream classifiers to poisoning attacks. The increased quality of the feature extractor is in general correlated to the increased Sep/SD and Sep/Size. To quantitatively measure the correlation of the error increase and each of the two factors, we show the Pearson correlation coefficient of the relevant metrics and the results are summarized in Table 10. From the table, we can see that when we directly measure the Pearson correlation with each individual factor, the correlation between the two factors and error increase is still very strong as the lowest (absolute) correlation coefficient of -0.66 still indicates a strong negative correlation. And if we further restrict ourselves to the ResNet-18 models only, then the (absolute) correlation coefficient is even higher, which again confirms the two factors are strongly (and negatively) correlated with the error increase.

In Table 10, we also compare the new experiment with the original experiment in Figure 2 and observe that the (absolute) Pearson correlation is reduced slightly, which is due to the fact that when the pretrained model does not involve any downstream data, the two factors may not consistently change in the same direction and undermines the Pearson correlation of each individual factor with

the error increase. However, if we compute the Pearson correlation between the number of epochs for training the ResNet-18 model (roughly incorporates the combined effects of both factors) and the error increase from poisoning, we observe an even stronger negative correlation (-0.99) compared to the original experiment (-0.95), which again supports the claim that training the deep architecture for more epochs seems to be one possible approach to improve resistance to poisoning. The slight increase in correlation in the new experiment results from the slightly higher covariance between the error increase and the number of epochs, but this difference is not significant since we only consider a limited number of samples when computing the correlation.

Developing general methods for learning meaningful feature extractors such that the two factors can be improved simultaneously in a controlled way would be an interesting research direction, which we plan to pursue in the future. However, we also note that this is a challenging task from the practical perspective since the goal is to extract useful feature representations for downstream applications, so the base error should also be considered in addition to the error increase from poisoning, especially when no downstream data are used in the upstream training. Our initial thought is to consider an intermediate few-shot learning setup [52], where a small number of (clean) downstream training data are allowed to be incorporated in training the feature extractor. This few-shot learning setup might correspond to an intermediate setting between the original experiment with full knowledge of the (clean) downstream training data and the new experiment with almost zero knowledge but still reflects certain real-world application scenarios. To conclude, all results in both the original and new experiments support our main claim that it is possible for the victim to choose better feature extractors to improve the resistance of downstream classifiers to indiscriminate poisoning attacks.

## E  Comparison to LDC and Aggregation Defenses

We first provide a more thorough discussion on the differences between our work and the Lethal Dose Conjecture (LDC) [50] from NeurIPS 2022, which had similar goals in understanding the inherent vulnerabilities of datasets but focused on targeted poisoning attacks (Appendix E.1). Then, we discuss how our results can also be related to aggregation based defenses whose asymptotic optimality on robustness against targeted poisoning attacks is implied by the LDC conjecture (Appendix E.2).

### E.1  Relation to LDC

As discussed in Section 1, LDC is a more general result and covers broader poisoning attack goals (including indiscriminate poisoning) and is agnostic to the learning algorithm, dataset and also the poisoning generation setup. However, this general result may give overly pessimistic estimates on the power of optimal injection-only poisoning attacks in the indiscriminate setting we consider. We first briefly mention the main conjecture in LDC and then explain why the LDC conjecture overestimated the power of indiscriminate poisoning attacks, followed by a discussion on the relations of the identified vulnerability factors in this paper and the key quantity in LDC.

**The main conjecture in LDC.** LDC conjectures that, given a (potentially poisoned) dataset of size $N$, the tolerable sample size for targeted poisoning attacks (through insertion and/or deletion) by any defenses and learners, while still predicting a known test sample correctly, is an asymptotic guarantee of $\Theta(N/n)$, where $n < N$ is the sample complexity of the most data-efficient learner (i.e., a learner that uses smallest number of clean training samples to make correct prediction). Although it is a conjecture on the asymptotic robustness guarantee, it is rigorously proven for cases of bijection uncovering and instance memorization, and the general implication of LDC is leveraged to improve existing aggregation based certified defenses against targeted poisoning attacks.

**Overestimating the power of indiscriminate poisoning.** LDC conjectures the robustness against targeted poisoning attacks, but the same conjecture can also be used in indiscriminate setting straightforwardly by taking the expectation over the tolerable samples for each of the test samples to get the expected tolerable poisoning size for the entire distribution (as mentioned in the original LDC paper) or by choosing the lowest value to give a worst case certified robustness for the entire distribution. The underlying assumption in the reasoning above is that individual samples are independently impacted by their corresponding poisoning points while in the indiscriminate setting, the entire distribution is impacted simultaneously by the same poisoning set. The assumption on the independence of the poisoning sets corresponding to different test samples might overestimate the power of indiscriminate

poisoning attacks as it might be impossible to simultaneously impact different test samples (e.g., test samples with disjoint poisoning sets) using the same poisoning set. In addition, the poisoning generation setup also greatly impacts the attack effectiveness—injection only attacks can be much weaker than attacks that modify existing points, but LDC provides guarantees against this worst case poisoning generation of modifying points. These general and worst-case assumptions mean that LDC might overestimate the power of injection-only indiscriminate poisoning attacks considered in this paper.

In practice, insights from LDC can be used to enhance existing aggregation based defenses. If we treat the fraction of (independently) certifiable test samples by the enhanced DPA [29] in Figure 2(d) (using $k = 500$ partitions) in the LDC paper as the certified accuracy (CA) for the entire test set in the indiscriminate setting, the CA against indiscriminate poisoning attack is 0% at the poisoning ratio of $\epsilon = 0.5\%$ (250/50000). In contrast, the best indiscriminate poisoning attacks [32] on CIFAR10 dataset reduces the model accuracy from 95% to 81% at the much higher $\epsilon = 3\%$ poisoning ratio using standard training (i.e., using $k = 1$ partition). Note that using $k = 1$ partition is a far less optimal choice than $k = 500$ as $k = 1$ will always result in 0% CA for aggregation based defenses. Our work explicitly considers injection only indiscriminate poisoning attacks so as to better understand its effectiveness.

While it is possible that current indiscriminate attacks for neural networks are far from optimal and there may exist a very strong (but currently unknown) poisoning attack that can reduce the neural network accuracy on CIFAR10 to 0% at a 0.5% poisoning ratio, we speculate such likelihood might be low. This is because, neural networks are found to be harder to poison than linear models [31, 32] while our empirical findings in the most extensively studied linear models in Section 3 indicate some datasets might be inherently more robust to poisoning attacks.

**Providing finer analysis on the vulnerability factors.** As mentioned above, LDC might overestimate the power of indiscriminate poisoning attacks. In addition, the key quantity $n$ is usually unknown and hard to estimate accurately in practice and the robustness guarantee is asymptotic while the constants in asymptotic guarantees can make a big difference in practice. However, the generic metric $n$ still offers critical insights in understanding the robustness against indiscriminate poisoning attacks. In particular, our findings on projected separability and standard deviation can be interpreted as the first step towards understanding the dataset properties that can be related to the (more general) metric $n$ (and maybe also the constant in $\Theta(1/n)$) in LDC for linear learners. Indeed, it is an interesting future work to identify the learning task properties that impact $n$ at the finer-granularity.

As for the projected constraint size (Definition 5.5), we believe there can be situations where it may be independent from $n$. The main idea is that in cases where changing $\mathcal{C}$ arbitrarily will not impact the clean distribution (e.g., when the support set of the clean distribution is a strict subset of $\mathcal{C}$, arbitrarily enlarging $\mathcal{C}$ will still not impact the clean distribution), the outcomes of learners trained on clean samples from the distribution will not change (including the most data-efficient learner) and hence $n$ will remain the same for different permissible choices of $\mathcal{C}$, indicating that the vulnerability of the same dataset remains the same even when $\mathcal{C}$ changes drastically without impacting the clean distribution. However, changes in $\mathcal{C}$ (and subsequently changes in the projected constraint size) will directly impact the attack effectiveness, as a larger $\mathcal{C}$ is likely to admit stronger poisoning attacks.

To illustrate how much the attack power can change as $\mathcal{C}$ changes, we conduct experiments on MNIST 1–7 and show that scaling up the original dimension-wise box-constraint from $[0, 1]$ to $[0, c]$ (where $c > 1$ is the scale factor) can significantly boost attack effectiveness. Table 11 summarizes the results and we can observe that, as the scale factor $c$ increases (enlarged $\mathcal{C}$, increased projected constraint size and reduced Sep/Size), the attack effectiveness also increases significantly. Note that this experiment is an existence proof and MNIST 1–7 is used as a hypothetical example. In practice, for normalized images, the box constraint cannot be scaled beyond [0,1] as it will result in invalid images.

### E.2   Relation to Aggregation-based Defenses

Aggregation-based (provable) defenses, whose asymptotic optimality is implied by the LDC, work by partitioning the potentially poisoned data into $k$ partitions, training a base learner on each partition and using majority voting to obtain the final predictions. These defenses provide certified robustness to poisoning attacks by giving the maximum number of poisoning points that can be tolerated to

| Scale Factor $c$ | Error Increase (%) | Sep/Size |
|:---:|:---:|:---:|
| 1.0 | 2.2 | 0.27 |
| 2.0 | 3.1 | 0.15 |
| 3.0 | 4.4 | 0.10 |

Table 11: Impact of scale factor $c$ on poisoning effectiveness for $\mathcal{C}$ in the form of dimension-wise box-constraint as $[0, c]$. Base Error is 1.2%. Base Error and Sep/SD will be the same for all settings because support set of the clean distribution is the strict subset of $\mathcal{C}$.

correctly predict a known test point, which can also be straightforwardly applied to indiscriminate setting by treating different test samples independently, as mentioned in the discussion of LDC.

Because no data filtering is used for each partition of the defenses, at the partition level, our results (i.e., the computed metrics) obtained in each poisoned partition may still be similar to the results obtained on the whole data without partition (i.e., standard training, as in this paper), as the clean data and $\mathcal{C}$ (reflected through the poisoning points assigned in each partition) may be more or less the same. At the aggregation level, similar to the discussion on LDC, these aggregation defenses may still result in overly pessimistic estimates on the effectiveness of injection only indiscriminate poisoning attacks as the certified accuracy at a particular poisoning ratio can be very loose, and the two possible reasons are: 1) straightforwardly applying results in targeted poisoning to indiscriminate poisoning might lead to overestimation and 2) considering the worst case adversary of modifying points might overestimate the power of injection only attacks in each poisoned partition. Therefore, our work can be related to aggregation defenses via reason 2), as it might be interpreted as the first step towards identifying factors that impact the attack effectiveness of injection only indiscriminate attacks in each poisoned partition, which may not always be highly detrimental depending on the learning task properties in each partition, while these aggregation defenses assume the existence of a single poisoning point in a partition can make the model in that partition successfully poisoned.

**Loose certified accuracy in indiscriminate setting.** Given a poisoning budget $\epsilon$, aggregation based defenses give a certified accuracy against indiscriminate poisoning attacks by first computing the tolerable fraction of poisoning points for each test sample and all the test samples with tolerable fraction smaller than or equal to $\epsilon$ are certifiably robust. Then, the fraction of those test samples to the total test samples gives the certified accuracy for the test set. Similar to the result of CIFAR10 shown in LDC, here, we provide an additional result of certified accuracy for neural networks trained on the MNIST dataset: the state-of-the-art finite aggregation (FA) method [51] gives a certified accuracy of 0% at 1% poisoning ratio (600/60,000) using $k = 1200$ partitions while at the much higher 3% poisoning ratio, the current state-of-the-art indiscriminate poisoning attack [32] can only reduce the accuracy of the neural network trained on MNIST without partitioning (i.e., $k = 1$, a far less optimal choice from the perspective of aggregation defenses) from over 99% to only around 90%.

# F  Extension to Multi-class Settings and Non-linear Learners

In this section, we first provide the high-level idea of extending the metric computation from binary setting to multi-class setting and then provide empirical results on multi-class linear models and show that these metrics can still well-explain the observations in multi-class linear classifiers (Appendix F.1). Then, we provide the idea of extending the metrics from linear models to neural networks (NNs) and also the accompanying experimental results (Appendix F.2). In particular, we find that, for the same learner (e.g., same or similar NN architecture), our metrics may still be able to explain the different dataset vulnerabilities. However, the extended metrics cannot explain the vulnerabilities of datasets that are under different learners (e.g., NN with significantly different architectures), whose investigation is a very interesting future work, but is out of the scope of this paper.

## F.1  Extension to Multi-class Linear Learners

Multi-class classifications are very common in practice and therefore, it is important to extend the computation of the metrics from binary classification to multi-class classification. For linear models, $k$-class classification ($k > 2$) is handled by treating it as binary classifications in the "one vs one"

| Datasets | Base Error (%) | Poisoned Error (%) | Increased Error (%) | Sep/SD | Sep/Size |
|----------|----------------|--------------------|--------------------|--------------|----------|
| SVM | 7.4% | 15.4% | 8.0% | **2.23**/2.73 | 0.06/**0.03** |
| LR | 7.7% | 30.6% | 22.9% | 1.15 | 0.02 |

Table 12: Results of Linear Models on MNIST using 3% poisoning ratio. The "Poisoned Error" is directly quoted from Lu et al. [31] and SVM one is quoted from Koh et al. [25]. SVM contains two values for Sep/SD and Sep/Size because there are two binary pairs with the lowest value for each of the two metrics (lowest value is made bold).

mode that results $k(k-1)/2$ binary problems by enumerating over every pair of classes or in the "one vs rest" mode that results in $k$ binary problems by picking one class as the positive class and the rest as the negative class. In practice, the "one vs rest" mode is preferred because it requires training smaller number of classifiers. In addition, the last classification layer of neural networks may also be roughly viewed as performing multi-class classification in "one vs rest" mode. Therefore, we only discuss and experiment with multi-class linear models trained in "one vs rest" mode in this section, consistent with the models in prior poisoning attacks [25, 32], but classifiers trained in "one vs one" mode can also be handled similarly.

**Computation of the metrics.** Although we consider linear models in "one vs rest" mode, when computing the metrics, we handle it in a way similar to the "one vs one" mode – when computing the metrics, given a positive class, we do not treat all the remaining k-1 classes (constitute the negative class) as a whole, instead, for each class in the remaining classes, we treat it as a "fake" negative class and compute the metrics as in the binary classification setting. Then from the $k-1$ metrics computed, we pick the positive and "fake" negative pair with smallest separability metric and use it as the metric for the current positive and negative class (includes all remaining $k-1$ classes)[2]. Once we compute the metrics for all the $k$ binary pairs, we report the lowest metrics obtained. The reasoning behind computing the metrics in (similar to) "one vs one" mode is, for a given positive class, adversaries may target the most vulnerable pair from the total $k-1$ (positive, negative) pairs to cause more damage using the poisoning budget. Therefore, treating the remaining $k-1$ pairs as a whole when computing the metrics will obfuscate this observation and may not fully reflect the poisoning vulnerabilities of a dataset.

We provide a concrete example on how treating the remaining classes as a whole can lead to wrong estimates on the dataset separability: we first train simple CNN models on the full CIFAR-10 [26] and MNIST datasets and achieve models with test accuracies of $\approx 70\%$ and $> 99\%$ respectively. When we feed the MNIST and CIFAR-10 test data through the model and inspect the feature representations, the t-SNE graph indicate that the CIFAR-10 dataset is far less separable than the MNIST, which is expected as CIFAR-10 has much lower test accuracy compared to MNIST. However, when we compute the separability metrics in our paper by considering all $k-1$ classes in the negative class, the separability of CIFAR-10 is similar to the separability of MNIST, which is inconsistent with drastic differences in the test accuracies of the respective CNN models. In contrast, if we treat each class in the remaining $k-1$ classes separately and pick the smallest value, we will again see the expected result that CIFAR-10 is far less separable than MNIST. Therefore, for the following experiments, we will compute the metrics by treating the remaining $k-1$ classes individually. We first provide the results of multi-class linear models for MNIST dataset below and then discuss our initial findings on the neural networks for CIFAR-10 and MNIST in Appendix F.2.

**Results on multi-class linear learners.** As explained above, when checking the $k$-binary pairs for a $k$-class problem, we report the lowest values for the Sep/SD and Sep/Size metrics. However, in some cases, the lowest values for the two metrics might be in two different pairs and in this case, we will report the results of both pairs. Table 12 summarizes the results, where the poisoned errors are directly quoted from the prior works—LR error is from Lu et al. [31] and the SVM error is from Koh et al. [25]. We can see that MNIST dataset is indeed more vulnerable than the selected MNIST 1–7 and MNIST 6–9 pairs because it is less separable and also impacted more by the poisoning points. We also note that the poisoned error of SVM is obtained in the presence of data sanitization defenses

---

[2]We still use the same projected constraint size for the $k-1$ positive and "fake" negative pairs because, the projected constraint size measures how much the decision boundary can be moved in presence of clean data points, which do not distinguish the points in the "fake" netgative class and the remaining $k-1$ classes.

| Datasets | Base Error (%) | Poisoned Error (%) | Increased Error (%) | Sep/SD | Sep/Size |
|---|---|---|---|---|---|
| MNIST | 0.8% | 1.9% | 1.1% | 4.58 | 0.10 |
| CIFAR-10 | 31.0% | 35.3% | 4.3% | 0.24 | 0.01 |

Table 13: Results on Simple CNN Models for MNIST and CIFAR-10 datasets using 3% poisoning ratio. The "Poisoned Error" of both datasets are directly quoted from Lu et al. [31]

| Datasets | Base Error (%) | Increased Error (%) | Sep/SD | Sep/Size |
|---|---|---|---|---|
| MNIST | 0.8% | 9.6% | 4.58 | 0.10 |
| CIFAR-10 | 4.8% | 13.7% | 6.36 | 0.24 |

Table 14: Results on Simple CNN Model for MNIST and ResNet18 model for CIFAR-10 datasets using 3% poisoning ratio. The "Poisoned Error" of both datasets are directly quoted from Lu et al. [32].

and hence, the poisoned error may be further increased when there are no additional defenses. We also see that, for SVM, although the lowest values for Sep/SD and Sep/Size are in two different pairs, their results do not differ much, indicating that either of them can be used to represent the overall vulnerability of MNIST.

### F.2 Extension to Multi-class Neural Networks

We first note that the insights regarding the separability and constraints set $\mathcal{C}$ can be general, as the the first metric measures the sensitivity of the dataset against misclassification when the decision boundary is perturbed slightly. The latter captures how much the decision boundary can be moved by the poisoning points once injected into the clean training data. The Sep/SD and Sep/Size metrics used in this paper are the concrete substantiations of the two metrics under linear models. Specific ways to compute the metrics in non-linear settings should still (approximately) reflect the high level idea above. Below, we use neural network (NN) as an example.

**High level idea.** We may partition the neural network into two parts of feature extractor and linear classification module and we may view the feature representations of the input data as a "new" data in the corresponding feature space, and so that we can convert the metric computations for non-linear neural network into metric computations (on feature space) for linear models. To be more concrete, we propose to use a fixed feature extractor, which can be extractors inherited from pretrained models (e.g., in transfer learning setting) or trained from scratch on the clean data, to map the input data to the feature space. Here, if the victim also uses the same pretrained feature extractor (as in the transfer learning setting), then our metrics can have higher correlation with the poisoned errors from existing attacks because the non-linear feature extractor is now independent from the poisoned points used in the victim's training. Below, we consider the from-scratch training case as it is more challenging.

**Computation of the metrics.** Although the feature extractor will also be impacted by the poisoning points now, in our preliminary experiment, we will still use the extractor trained on the clean data and leave the exploration of other better feature extractors as future work. Using the transformation from the clean feature extractor, the projected separability and standard deviation can be easily computed. But the computation of the projected constraint size can be tricky, because the set $\mathcal{C}$ after transforming through the feature extractor can be non-convex and sometimes, for complicated $\mathcal{C}$, computing such transformation can be very challenging (can be a case even for linear models), but is a very interesting direction to explore. For the simple forms of $\mathcal{C}$ such as the dimension-wise box constraints considered in this paper, we may leverage the convex outer polytope method [53] to bound the activation in each layer till the final feature layer so that we can obtain a final transformed convex set $\mathcal{C}'$ using the feature extractor, which is a set that contains the original $\mathcal{C}$. However, due to time limitation, when computing the projected constraint size in the following experiments, we simply set $\mathcal{C}$ as the dimension-wise box-constraints, whose minimum and maximum values are computed from the feature representations of the clean data points, similar to the transfer-learning experiment in Section 7.

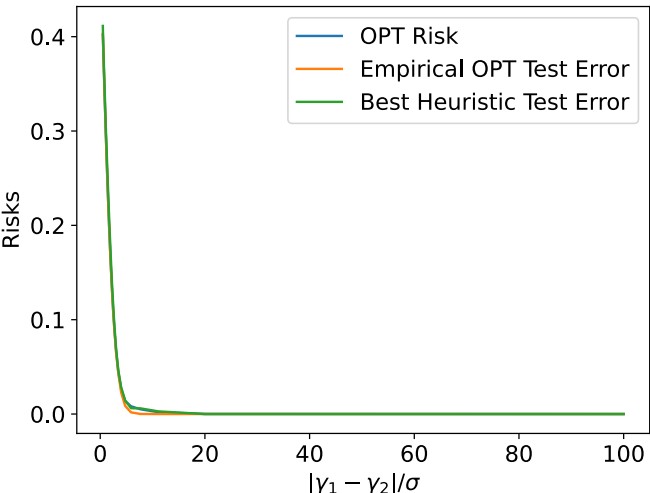

Figure 6: Measuring the performance of optimal poisoning attacks with different values of separability ratio $|\gamma_1 - \gamma_2|/\sigma$. *Best Heuristic* denotes the best performing attack in the literature.

**Results under similar learners.** For the experiments, we use the simple CNN models presented in Lu et al. [31] for MNIST and CIFAR-10 datasets (similar architecture). We directly quote the attack results of TGDA attack by Lu et al. [31] for both the CIFAR-10 and MNIST datasets. Note that, very recently, a stronger GC attack is also proposed by Lu et al. [32] and outperforms the TGDA attack. However, we could not include the newer result because the code is not published and the evaluation in the original paper also did not include the simple CNN for CIFAR-10 dataset. The results are shown in Table 13. From the table, we can see that CIFAR-10 tends to be more vulnerable than MNIST as the Sep/SD and Sep/Size metrics (in neural networks) are all much lower than those of MNIST. These significantly lower values of CIFAR-10 may also suggest that the poisoned error for CIFAR-10 with simple CNN maybe increased further (e.g., using the stronger attack in Lu et al. [32]).

**Results under different learners.** Above, we only showed results when the MNIST and CIFAR-10 datasets are compared under similar learners. However, in practical applications, one might use deeper architecture for CIFAR-10 and so, we computed the metrics for CIFAR-10 using ResNet18 model. Then we compare the metrics of MNIST under simple CNN and the metrics of CIFAR-10 under ResNet18 in Table 14, where the poisoned errors are quoted from the more recent GC attack [32] because the attack results are all available in the original paper. However, from the table we can see that, although MNIST is less separable and impacted more by the poisoning points, the error increase is still slightly smaller than CIFAR-10, which is not consistent with our metrics. If the current attacks are already performing well and the empirical poisoned errors are indeed good approximations to the inherent vulnerabilities, then we might have to systematically investigate the comparison of vulnerabilities of different datasets under different learners.

## G  Evaluation on Synthetic Datasets

In this section, we empirically test our theory on synthetic datasets that are sampled from the considered theoretical distributions in Section 5.1.

### G.1  Synthetic Datasets

According to Remark 5.4, there are two important factors to be considered: (1) the ratio between class separability and within-class variance $|\gamma_1 - \gamma_2|/\sigma$, denoted by $\beta$ for simplicity; (2) the size of constraint set $u$. We conduct synthetic experiments in this section to study the effect of these factors on the performance of (optimal) data poisoning attacks.

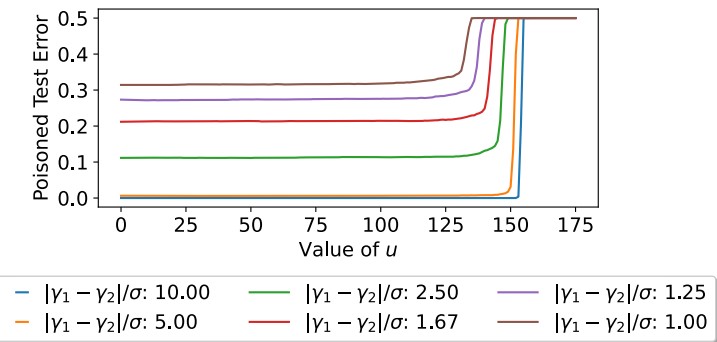

Figure 7: Impact of the (projected) constraint size $u$ on poisoning effectiveness. $u = 0$ means the test error without poisoning.

More specifically, we generate 10,000 training and 10,000 testing data points according to the Gaussian mixture model (2) with negative center $\gamma_1 = -10$ and positive center $\gamma_2 = 0$. Throughout our experiments, $\gamma_1$ and $\gamma_2$ are kept fixed, whereas we vary the variance parameter $\sigma$ and the value of $u$. The default value of $u$ is set as 20 if not specified. Evaluations of empirical poisoning attacks require training linear SVM models, where we choose $\lambda = 0.01$. The poisoning ratio is still set as 3%, consistent with evaluations on the benchmark datasets.

**Impact of $\beta$.** First, we show how the optimal attack performance changes as we increase the value of $\beta$. We report the risk achieved by the OPT attack based on Theorem 5.3. Note that we can only obtain approximations of the inverse function $g^{-1}$ using numerical methods, which may induce a small approximation error for evaluating the optimal attack performance. For the finite-sample setting, we also report the empirical test error of the poisoned models induced by the empirical OPT attack and the best current poisoning attack discussed in Section 3, where the latter is termed as *Best Heuristic* for simplicity. Since the poisoned models induced by these empirical attacks do not restrict $w \in \{-1, 1\}$, we normalize $w$ to make the empirical results comparable with our theoretical results.

Figure 6 summarizes the attack performance when we vary $\beta$. As the ratio between class separability and within-class variance increases, the risk of the OPT attack and empirical test errors of empirical OPT and best heuristic attacks gradually decrease. This is consistent with our theoretical results discussed in Section 5.1. Note that there exists a minor difference between these attacks when the value of $\beta$ is small, where the test error attained by the best current heuristic poisoning attack is slightly higher than that achieved by the empirical OPT attack. This is due to the small numerical error induced by approximating the inverse function $g^{-1}$.

**Impact of $u$.** Our theoretical results assume the setting where $w \in \{-1, 1\}$. However, this restriction makes the impact of the constraint set size $u$ less significant, as it is only helpful in judging whether flipping the sign of $w$ is feasible and becomes irrelevant to the maximum risk after poisoning when flipping is infeasible. In contrast, if $w$ is not restricted, the impact of $u$ will be more significant as larger $u$ tends to reduce the value of $w$, which in turn makes the original clean data even closer to each other and slight changes in the decision boundary can induce higher risks (further discussions on this are found in Appendix G.2).

To illustrate the impact of $u$ in a continuous way, we allow $w$ to take real numbers. Since this relaxation violates the assumption of our theory, the maximum risk after poisoning can no longer be characterized based on Theorem 5.3. Instead, we use the poisoning attack inspired by our theory to get an empirical lower bound on the maximum risk. Since $\gamma_1 + \gamma_2 < 0$, Theorem 5.3 suggests that optimal poisoning should place all poisoning points on $u$ with label $-1$ when $w \in \{1, -1\}$. We simply use this approach even when $w$ can now take arbitrary values. We vary the value of $u$ gradually and record the test error of the induced hypothesis, We repeat this procedure for different dataset configurations (i.e., fixing $\gamma_1$, $\gamma_2$ and varying $\sigma$).

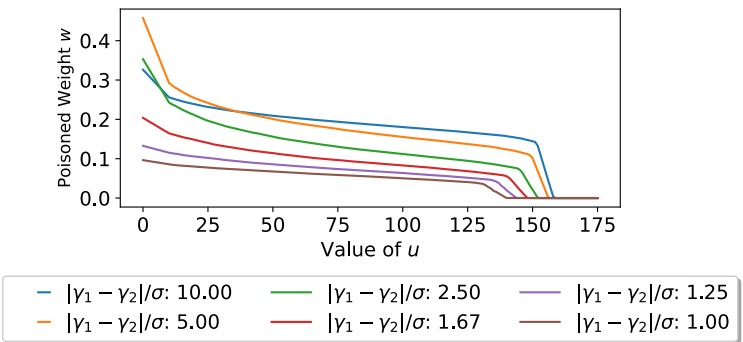

Figure 8: Impact of the (projected) constraint size $u$ on the value of $w$ after poisoning in 1-D Gaussian distribution.

The results are summarized in Figure 7. There are two key observations: (1) once $w$ is no longer constrained, if $u$ is large enough, the vulnerability of all the considered distributions gradually increases as we increase the value of $u$, and (2) datasets with smaller $\beta$ are more vulnerable with the increased value of $u$ compared to ones with larger $\beta$, which has larger increased test error under the same class separability and box constraint (choosing other values of $\beta$ also reveals a similar trend). Although not backed by our theory, it makes sense as smaller $\beta$ also means more points might be closer to the boundary (small margin) and hence small changes in the decision boundary can have significantly increased test errors.

## G.2 Relationship Between Box Constraint Size and Model Weight

Our theory assumes that the weight vector of $w$ can only take normalized value from $\{-1, 1\}$ for one-dimensional case, while in practical machine learning applications, convex models are trained by optimizing the hinge loss with respect to both parameters $w$ and $b$, which can result in $w$ as a real number. And when $w$ takes real numbers, the impact of $u$ becomes smoother: when poisoning with larger $u$, the poisoning points generated can be very extreme and forces the poisoned model to have reduced $w$ (compared to clean model $w_c$) in the norm so as to minimize the large loss introduced by the extreme points. Figure 8 plots the relationship between $u$ and $w$ of poisoned model, and supports the statement above. When the norm of $w$ becomes smaller, the original clean data that are well-separated also becomes less separable so that slight movement in the decision boundary can cause significantly increased test errors. This makes the existence of datasets that have large risk gap before and after poisoning more likely, which is demonstrated in Figure 7.

