# OpenReview forum: "What Distributions are Robust to Indiscriminate Poisoning Attacks for Linear Learners?"
_NeurIPS.cc/2023/Conference — NeurIPS 2023 poster_

### Official Review · Reviewer_oRg1 · 2023-06-15

**Soundness:** 3 good
**Presentation:** 3 good
**Contribution:** 2 fair
**Rating:** 5
**Confidence:** 4

**Summary:**

This paper explores poisoning attacks on linear classifiers.  Specifically, the authors explore why poisoning attacks may have high efficacy on one dataset but low efficacy on another.  The authors relate this phenomenon to two potential factors: (1) (projected) constraint set size and separability. The authors theoretically characterize an optimal poisoning attack w.r.t. Gaussian mixture data.

**Strengths:**

Previous work has repeatedly shown that poisoning attacks represent a real and serious threat to machine learning models.  Most work in the field focuses on how to design better attacks or better defenses.  An improved theoretical understanding of what mechanisms allow an attack to work could accelerate research for both attacks and defenses.  In my view, this represents a challenging, under-explored research topic. This paper makes an incremental step into improving understanding in this area.

**Weaknesses:**

I rate the paper as borderline reject and explain my rationale below. I want to make clear that with a thoughtful rebuttal and changes to different aspects, I could definitely see my score increasing.

While the paper has many things I like, it is my view that the paper still needs work.  Perhaps the biggest problem is that the writing tends to overclaim results or draw conclusions that, as written, are not well supported.  I discuss a couple of cases below under "Experimental Results." I also ask multiple questions about points I had concerns with in the "Questions" section below.

* A more harmless example of what I perceived as overclaiming is on lines 42-43, which state, "we consider indiscriminate poisoning attacks for linear models, *the most commonly studied victim models in the literature* (emphasis added). It is not my assessment that linear models are the most commonly studied victim model, particularly recently. Reading this sentence, it felt to me that the authors were stretching to over-justify the paper's restriction to linear models. This is despite the subsequent sentences being more than adequate to justify the choice, e.g., "linear models are still not well understood" and "high interpretability."

* A second example is on lines 35-36, which state "the reasons why attacks are sometimes ineffective have not been previously studied."  I do not know this statement to be true and expect given other work (e.g., the lethal dose conjecture) that this is not true.

**Experimental Results**

My biggest concerns are around the experimental results and the conclusions drawn from them.

* Lines 343-344 state, "Table 1 summarizes the results show that Sep/SD and Sep/Size can *largely* explain why datasets...are harder to poison than others." In my view, this statement is making primarily a causation claim. However, the results in the table provide primarily correlation results.  To make a causation claim (which I think is the appropriate type of claim for this paper to try to make), the experiments should externally vary the separability and measure how the performance changes.  I realize that is non-trivial, but I believe it is the right type of experiment here.

* Lines 390-392 state, "Although limited in effectiveness, the defense still mitigates the poisoning to some degree *mostly by shrinking the projected constraint size*" I do not see how the empirical results as presented can support such a strong claim.  As written, the text may barely support the claim that smaller projected constraint size *partially* helps (a weak, not that meaningful claim). To justify *mostly* a lot more is needed.

* Section 7 Better Feature Representation: Lines 363-364 state, "We train LeNet and ResNet18 on the CIFAR10 dataset..." From this sentence, it is my understanding that your pretraining was done on all 10 classes.  If so, your feature encoder goes beyond the transfer learning setting, and I believe this approach does *not* correspond with "the practical scenario where the victim has access to some *small number* of clean samples..."

* On lines 360-361, we present two ideas -- using better features might improve separability and using data filtering might reduce the projected constraint size." I would expect data filtering would both improve separability and reduce the constraint size.  The paper should explain how you are ensuring filtering is only affecting one criteria and not the other.

* I understand why the authors chose to lay out Figure 1 as they did.  However, I did not find the overlapping bars were the best format.  I was initially confused by the formatting, and it took me some time to figure out exactly what was being visualized since I expected multiple attacks, but only saw two bars.

**Miscellaneous Comments**

* Overall, I found section 5.1 notationally cumbersome and lacking a clear structure and flow. I had to re-read it twice to extract all the main points.  One possible fix is to provide more intuition about why each is being proposed and how the pieces relate.  It felt as though you gave a lot of disparate pieces of information to the reader without walking them through how they fit together as you went alone

* Lines 40-41 state, "An attack is considered ineffective if the increased risk from poisoning is roughly equal to or smaller than the injected poisoning rate."  This perspective makes sense, but it took me a minute to understand the definition as stated. I think this sentence could be written more intuitively.

**Questions:**

* Eq. (2) allows the two Gaussian distributions to have distinct variances (e.g., $\sigma_1$ and $\sigma_2$). In Theorem 5.3, you specify the two variances as equivalent. Is Theorem 5.3's equivariance specification for convenience in the proof or a necessity?

* On Line 155, you wrote, "*We chose 3% as the maximum poisoning rate." I found this confusing wording. I understanding "maximum poisoning rate" is terminology you use in the theoretical discussion.  When discussing experiments you perform, defining your poisoning rate w.r.t. a *maximum* implies to me you tested multiple poisoning rates. However, I do not get that indication elsewhere in the paragraph.  Please be precise concerning your experimental setup.

* Figure 1 uses the "Filtered Enron" dataset.  I am not familiar with the filtered version of the Enron dataset. Is this filtering procedure something you performed or an existing dataset? How do the results change if you use the vanilla Enron dataset?  If you filtered the dataset yourself, I believe the paper should explain why that filtering was done, not just how it was performed.

* Your preliminaries section specifies a few design choices. For example, you restrict consideration to primarily hinge loss. Moreover, you specify an L2 regularizer (Eq. (1)). As a reader, I am somewhat left to guess why these specific choices are made. For example, I assume you chose hinge loss because you want max margin.  Concerning regularizer, I was unsure if L2 was for convenience and whether a different convex regularizer (e.g., L1) could be used. If you are focusing on specific design choices, I believe it is better to explain why.
    * I do not recall any of the theorems requiring specifically the hinge loss as opposed to alternate convex losses.

* Lines 183-184 state, "Definition 4.1 *suggests* that no poisoning strategy can achieve a better attack performance than that achieved by $\mathcal{A}_{opt}$" When reading this, I was confused by the choice of the word "suggests," as given $\epsilon$, I found Def. 4.1 clearly stated no alternate strategy achieves better attack performance.

* One criterion for Theorem 4.5 to hold is that the support of $\mu_c$ should be a (proper) subset of $\mathcal{C}$.  I believe this criterion does not hold in the optimal Gaussian case given the definition of $\mathcal{Q}(u)$.  If so, please explain how that affects your results (including the proof). Ideally, this should be more explicit in the paper.
   * Searching through the paper, I did not see where $\mathcal{Q}(u)$ was used beyond line 246. Perhaps I missed it. If it is just for that one point, I think it is worth removing.

**Limitations:**

Better understanding what causes attacks to fail could lead to enable better attacks.  At the same time, it can lead to better defenses.  I see no serious ethical or societal risks for this work.

---

> ### Author Rebuttal · Authors · 2023-08-09
>
> We first respond to concerns about the experiments and conclusions we draw. To utilize space, we also respond to the last comment from the **Questions**. For other unaddressed comments, we will reflect them in the revised version.
>
> 1. **Statement on results in Table 1 is trying to make a causation claim, but results are correlation results**
>
> We should have been more careful in our writing, but did not mean to claim a causation result from Table 1, as it only shows the correlation between the factors and the empirical vulnerabilities to poisoning. We will revise our statement to make it explicit that we are showing the correlation, not causation. We will revise the writing to avoid any unfounded implications of causation, and to ensure we explicitly mention where needed that we found factors that are correlated to the disparate attack effectiveness across datasets.
>
> We also speculate our factors might cause vulnerability to poisoning due to the following reasons, but these are just some partial evidence and may not be sufficient to provide a definitive answer. We also clarify that we are not claiming our factors are the only factors that might cause the vulnerability. First, for the 1-D Gaussian example, we exactly characterized the optimal poisoning attacks and then provided an upper bound on the performance of optimal attacks for general distributions. And indeed, when the separability ratio (Sep/SD) is high and the projected constraint size (Size) is small, the distributions will be less prone to poisoning.
>
> Second, limited empirical results (due to computing constraints, running all the attacks on large number of datasets is infeasible) might also provide partial evidence for the possible causation from the two factors (Sep/SD and Sep/Size) to the poisoning vulnerability. In particular, as suggested by the reviewer, we vary one factor (ideally with another one fixed) externally and then measure the performance changes in the tested attacks.
>
> We conducted new experiments on MNIST 1-7. We changed the distributions of MNIST 1-7 such that the Sep/SD could change gradually while, ideally, the Sep/Size could remain the same. However, in practice, keeping Sep/Size (or Size) fixed is hard because the computation of Size depends on the clean boundary, which is again dependent on the (modified) clean distribution. Therefore, even if we use the same constraint set $\mathcal{C}$ for different modified datasets of MNIST 1-7, the Size or Sep/Size can be different. Below is the detail of our modification.
>
> We first train a linear SVM on the original MNIST 1-7 dataset. Then, we pick 3% of correctly classified points that are closest to the clean boundary. These points, once repeated $x$ times, replace $x \cdot$3% of correctly classified points that are furthest from the boundary. This process can be viewed as oversampling in the near-boundary region and undersampling in the region far from the boundary, and aim to create a dataset with a low separability ratio. We tested with $x = 1,2,4,6,8,10$ and applies to both the training and test data. We only deal with correctly classified points because we want to ensure the modified dataset still has high clean accuracy. The results are summarized in Table 1 in the attached PDF file in the global response. As mentioned above, even using the same $\mathcal{C}$ for all modified datasets, we still failed to keep the Sep/Size unchanged. We leave the exploration of better ways to change data distributions with the Sep/Size fixed as future work. From the table, we see that as we increase $x$, the Sep/SD gradually decreases, and the attack effectiveness (measured by the error increase) increases. Interestingly, the Sep/Size also decreases with the same $\mathcal{C}$.
>
> For Size (or Sep/Size), its impact can be measured more easily as we can change $\mathcal{C}$ without impacting the clean distribution, which means Sep/SD can be fixed. We performed a hypothetical experiment where we scale the original box-constraint from $[0,1]$ to $[0,c]$, where $c\geq 1$ is the scale factor and $c=1$ denotes the original setting for MNIST 1-7. The experiments are hypothetical because, in practice, a box-constraint of $[0,c]$ with $c > 1$ will result in invalid images. We experimented with $c=2,3,5,7,10$ and the results are summarized in Table 2 in the attached PDF file. From the table, we see as $c$ becomes larger, the Size increases and correspondingly the Sep/Size decreases. These changes are associated with a gradual increase in the *Error Increase* from poisoning.
>
> Although the preliminary results above are not sufficient to prove a causation claim, they still show the possibility with partial evidence.
>
>
> **(Questions) The first criterion of Thm 4.5 does not hold for 1-D Gaussian**
>
> This is a great point and we should have made it clear in the paper. For the 1-D Gaussian example in Sec 5.1, it holds based on the second criterion in Theorem 4.5 and will not impact the results and proofs. In particular, to satisfy the criterion, since $\boldsymbol{\theta}=(w,b)$ and $w\in$ { +1,-1}, given a $(w,b)$, we only need to concern with the gradient of $b$ w.r.t the hinge loss. If there is a valid point $(x,y)$ with zero gradient, then $\mu$ only contains that $(x,y)$. If no such point exists, then we pick any two valid points with different labels and $\mu$ has equal 50\% probability on each point.
>
> The remaining comments on the experimental results are related to the experiments in the discussion section (Section 7). We note that, the purpose of Section 7 is to identify possible implications from our work and support the claim through some preliminary results. This is not the major contribution of this work but rather provides partial evidence on leveraging the insights in this paper to build better defenses in the future. Therefore, we will revise the wording carefully to avoid making strong claims. **The remaining (detailed) responses are in the global response above.**

---

> > ### Comment · Reviewer_oRg1 · 2023-08-11
> > **Initial Reply**
> >
> > I have read the other reviews and the authors' rebuttals.  I noticed that the authors may have felt constrained by the length limitations on their rebuttal. If the authors feel the length limitation prevented them from addressing all points they wanted to make, I am fine with them making multiple official comments under my review. I will read all of them.
> >
> > Given the amount of points I raised and the length of the rebuttal, I cannot respond to most points tonight. I will address some of the points below and plan to reply more over the weekend.  Of course, the authors are welcome to rebut my partial response if they choose.
> >
> > > Why same variance in Theorem 5.3?
> >
> > Your explanation is reasonable. If there is not already a note explaining this choice, please consider adding one.
> >
> > > Is hinge loss specifically required?
> >
> > This updated explanation makes a lot more sense.  I recommend updating the manuscript to reflect your clearer, more detailed rationale above.
> >
> > > We will move part of our results in Section D of the appendix to the main paper in the revised version.
> >
> > In my view, I would leave the lethal dose conjecture (LDC) discussion as is in the main paper.  I understand the reviewer's statement "I don't think comparing with LDC is necessary as I clearly see the drastic difference."  However, LDC is a recent important work in this area and I think the paper is stronger when the contrast is clear and explicit -- even if some readers may find the contrast obvious.
> >
> > > Figure 1
> >
> > Would it be possible to add an updated Figure 1 to your one-page rebuttal for us to better visualize what you have in mind?  I am not sure what is possible or if you could also link an anonymized separate document.
> >
> > > Experiments on better feature representation do not correspond to the practical scenario of
> >
> > In short, I consider the feature extractor used in this experiment particularly unrealistic.  For example, since the feature extractor is trained on all of CIFAR10, the feature extractor's objective function specifically optimizes over the examples then used in your fine-tuning analysis.
> > A more meaningful experiment setup would train the feature extractor only on 8 of 10 classes (e.g., everything but "Truck" and "Ship"). If the feature extractor is trained over all instances with the target labels, then, in my view, we are no longer in the "practical scenario where the victim has access to some small number of clean samples."  Intuitively, an extractor that uses supervised pretraining to extract the best possible features will always be (far) better than when a victim who "has access to some small number of clean samples."

---

> > > ### Author Response · Authors · 2023-08-11
> > > **Additional response to unaddressed and new follow-up comments**
> > >
> > > We really appreciate your prompt response. Please find our additional responses to the follow-up comments and also the unaddressed comments in the initial rebuttal.
> > >
> > > > Same variance in Thm 5.3
> > >
> > > We will definitely add clarification in the revised version to make the presentation of the results clearer.
> > >
> > > > Hinge loss requirement
> > >
> > > We will also clarify this in the revised version, with provided details in the initial response. This will greatly improve the readability of the paper.
> > >
> > > > About results on LDC
> > >
> > > We did not mean to exclude the comparison to LDC from the main paper. We will work around spaces in other parts of the paper to make sure the LDC content is there while also including some interesting results from Section D. We acknowledge the importance of LDC in the area and that is why we included a detailed comparison to LDC in Appendix E.
> > >
> > > > Figure 1
> > >
> > > Please find the revised Figure 1 from this anonymous [link](https://drive.google.com/drive/folders/1KAwXQP-OtnDV89oF7UdbS2pi2TU0D_jk?usp=sharing). We were no longer allowed to update the attached PDF file in the global response.
> > >
> > > > Experiments on transfer learning experiment
> > >
> > > Thanks for the clarification. We now understand the concerns of the reviewer. We first note that the current experiment still shows the possibility of the victim selecting a better feature extractor free from poisoning to improve the resilience of the downstream classifiers to indiscriminate attacks. However, we agree with the reviewer that a more meaningful experimental setup should exclude the training data used in the downstream analysis when training the feature extractor. We are currently rerunning the experiments where the feature extractor is trained on the full CIFAR10 dataset except data points from classes of "Ship" and "Truck".
> > >
> > > We will post another response when the results are ready but decided to go with other comments first in case there are additional follow-up questions from the reviewer.
> > >
> > > Below, we provide responses to comments that are not addressed in the rebuttal.
> > >
> > >
> > > > linear models are not the most commonly studied victim against indiscriminate poisoning attacks, particularly recently
> > >
> > > Thanks for raising this point. In the original submission, we meant to say "most commonly" by the extensive number of concrete attack algorithms proposed for the indiscriminate setting that inject small amount of poisoning points, but realized this statement is indeed confusing and not general. We will simply remove this sentence in the revised version. Reviewer is also correct that, particularly recently, there are more focus towards using neural networks as the victim models and we also plan to perform more systematic investigation on non-linear models such as neural networks in our next step.
> > >
> > >
> > > > The statement of reasons on why attacks are sometimes ineffective have not been previously studied is not true
> > >
> > > In the original submission, we meant to say, for the previous works that propose concrete indiscriminate attacks, the reasons why sometimes their attacks are ineffective are not investigated. This is why this sentence appears right after the list of proposed indiscriminate poisoning attacks. We will make this statement clearer to avoid possible confusion.
> > >
> > >
> > > > Improve the presentation in Sec 5.1
> > >
> > > Thanks for raising this, and our presentation will be much clearer if the connections of the presented definitions and theorems were introduced verbally at the beginning of Sec 5.1. The proof sketch under Theorem 5.3 attempted to serve for this purpose, but we will now revise our paper by making these connections more explicit at the beginning of the section.
> > >
> > >
> > > > Definition of vulnerability should be stated more intuitively
> > >
> > > We will restate this definition as "Following definitions given in the existing works, an attack is considered ineffective if the increased error from poisoning is equal to or less than the injected poisoning ratio".
> > >
> > > > Why stating 3\% as the maximum poisoning ratio?
> > >
> > > The reviewer is correct that, in the experiments, we only use 3% as the poisoning ratio, not as the maximum poisoning ratio. Although we included results of different poisoning ratios in Appendix D, these results are not related to the main claims made in the paper using the 3\% poisoning ratio and we will revise the statement to make it precise.
> > >
> > > > Definition 4.1 suggests?
> > >
> > > We will replace the word "suggests" with "states".

---

> > > > ### Comment · Reviewer_oRg1 · 2023-08-12
> > > > **Further Reply**
> > > >
> > > > Thank you for providing the additional information I requested.  Overall you have addressed most of my concerns. I look forward to your updated experimental results.  I will reassess my score once those results are available.

---

> > > > > ### Author Response · Authors · 2023-08-12
> > > > > **New result on Figure 2**
> > > > >
> > > > > We are glad that our response addresses most of your concerns, and really appreciate your time and effort in reading our responses carefully. Below, we provide the new results on Figure 2 in the original submission.
> > > > >
> > > > > We repeated the experiments of Figure 2 in the paper with a new feature extractor whose training never witnesses data points from the classes of "Ship" and "Truck". The results are summarized in the updated figure available in this anonymized image: [link](https://drive.google.com/file/d/1umD2sv5fR9t1nK4Ed4N8fdX0YOsfaVrZ/view?usp=share_link).
> > > > >
> > > > > Each data point in the figure is marked with a text in the form of `F:(base,increase)`, where `F` denotes the feature extractor type. For the extractor types, similar to the description in the paper, `LeNet` means an extractor of a fully trained simple CNN model while `R-X` means the ResNet-18 model trained for `X` epochs. `base` denotes the base error (without poisoning) when we train a linear SVM model on the feature representations of "Ship" and "Truck" (these two classes are not used in the feature extractor training), and `increase` denotes the error increase resulting from the state-of-the-art poisoning attacks.
> > > > >
> > > > > Overall, the main message in the updated figure is similar to what we showed in Figure 2 in the original submission. In particular, when we adopt deep architecture and train it for more epochs, then the quality of the obtained feature extractor increases and so, it appears to improve the resistance of downstream classifiers to poisoning attacks. The increased quality of the feature extractor is in general correlated to increased Sep/SD and Sep/Size.
> > > > >
> > > > > An interesting (and somewhat surprising) observation is, the LeNet model, even though has the second-lowest base error, has the highest increased error among all tested datasets. We speculate one possible reason is that the search space of the poisoning points is relatively large, which makes the overall dataset more vulnerable, although further experiments would be necessary to confirm this. It would be interesting to perform further investigation and also to understand the differences between two model architectures (e.g., simple CNN and ResNet18).
> > > > >
> > > > > We note that there are also some differences compared to the results in Figure 2 in the original submission, where the feature extractor training additionally uses the downstream training data. First, because the feature extractor never sees images of "Ship" and "Truck", the base errors of the downstream linear models are high (the lowest base error in our new experiment is 12.8% while the lowest one reported in the original submission is only 1.0%). Second, the differences between adjacent epochs are no longer that distinguishable. For example, comparing R-90 and R-100, training for an additional 10 epochs does not appear to bring additional benefits.
> > > > >
> > > > > To conclude, our new results still show the possibility of the victim choosing a better feature extractor to improve the resistance of downstream classifiers to poisoning attacks. Based on our limited data points, one possible solution might be to adopt some deep architectures and train for a sufficient number of epochs. We leave a more systematic investigation along this direction for future work. We will also revise our statement to avoid drawing strong conclusions based on the preliminary experiments.

---

> > > > > > ### Comment · Reviewer_oRg1 · 2023-08-15
> > > > > > **Updated Score**
> > > > > >
> > > > > > Below I attempt to briefly summarize the takeaways from the revised experimental setup. If you disagree with any parts of my summary, you are more than to point them out.
> > > > > >
> > > > > > 1. Your latest response writes, "*The increased quality of the feature extractor is in general correlated to increased Sep/SD and Sep/Size*." I agree with that general sentiment. I recommend including, at minimum, the Pearson correlation in your rebuttal and the paper to make "generally correlated" more concrete.
> > > > > >
> > > > > > 2. It seems the correlation is weaker than in the original manuscript's version of the figure (visually, I cannot determine if the results constitute "substantially weaker"). A comparison of the Pearson correlation with and without the change would probably be appropriate for discussion here.
> > > > > >
> > > > > > Unfortunately, it seems the modified experiments (somewhat) weaken the strength of your overall claims.  If you have any thoughts on improving the experimental design to better support the claims, I would be interested in discussing more.  For now, I will update my score based on the new results and discussion so far.

---

> > > > > > > ### Author Response · Authors · 2023-08-17
> > > > > > > **Follow-up discussion on new results**
> > > > > > >
> > > > > > > Thanks for updating the score and we are happy to address your follow-up comments.
> > > > > > >
> > > > > > > We agree that Pearson correlation coefficients are good metrics to demonstrate the correlation quantitatively between our identified factors and error increase. Below, we show the computed coefficients with respect to each individual factor (i.e., Sep/SD or Sep/Size) for both Figure 2 in our original submission and the updated Figure 2 in the new experiment:
> > > > > > >
> > > > > > > | Figure 2 | Sep/SD vs Error Increase | Sep/Size vs Error Increase |
> > > > > > > |:--- | :----:        |    :----:   |
> > > > > > > | Original | -0.98 (-0.97)      | -0.98 (-0.97)       |
> > > > > > > | New | -0.66 (-0.90)  | -0.90 (-0.99) |
> > > > > > >
> > > > > > > The value in the parentheses denotes the Pearson correlation when we exclude the `LeNet` structure and only consider `ResNet-18` models. From the table, if we directly measure the Pearson correlation with each individual factor, it is true that the correlation between the two factors and error increase becomes weaker when the feature extractor no longer uses the downstream training data. However, it is worth noting that the lowest (absolute) correlation coefficient `-0.66` still indicates a fairly strong negative correlation. And if we further restrict ourselves to ResNet-18 models only, then the (absolute) correlation coefficient is even higher, which again confirms the two factors are strongly correlated with error increase.
> > > > > > >
> > > > > > > Importantly, we want to clarify that the computed Pearson correlation coefficients in the new experiment being not as high as those of the original Figure 2 does _NOT_ weaken our overall claim. Recall for our main experiments in Section 6, the major claim is that the two identified factors (i.e., Sep/SD and Sep/Size) are strongly correlated to the effectiveness of indiscriminate data poisoning attacks across different datasets, which is well-supported through the high degree of correlation indicated by the Pearson correlation coefficients computed for the new Figure 2 in the new experiment above. In addition, we note that there is likely to be a joint impact of the two factors, and it is also possible for the correlation to be non-linear as suggested by our theoretical results (e.g., see Theorem 5.3 for Gaussian distributions), which may not be well captured by the Pearson correlation with each individual factor, since they are used to characterize the linear correlation between two variables. We believe it would be an interesting yet challenging future work to characterize such relationship as comprehensively as possible between the two identified factors and error increase under the transfer learning setup. Our initial thought to gain a better understanding of such relationship is to produce more dots in the new Figure 2 by varying the feature extractor (e.g., varying model architecture or using pretrained ImageNet encoders), which may potentially uncover interesting patterns.
> > > > > > >
> > > > > > >
> > > > > > > In addition to not weakening our main claim, the Pearson correlation coefficients calculated for the updated Figure 2 also suggest similar possible practical implications on how to build more robust defenses against indiscriminate poisoning attacks, as what we explained in Section 7 of the original submission. Recall that the implication drawn from the original Figure 2 is that using deeper architecture and training for more epochs on datasets free from poisoning can potentially be an effective approach for enhancing the downstream model robustness by increasing both factors simultaneously. Since the Pearson correlation computed for the updated Figure 2 confirms that each individual factor is negatively correlated with the error increase, we can still conclude that increasing both Sep/SD and Sep/Size regrading the feature extractor might be a potential way for improving the resilience of downstream classifiers. Nevertheless, there exists a slight difference when we consider feature extractors that are not trained using any downstream data, as explained in our last response. The identified two factors no longer strictly increase simultaneously (e.g., compare R-25 and R-50 in the new experiment, where R-50 has a reduced Sep/SD but increased Sep/Size) as we train for more epochs. But, if we compute the Pearson correlation between the number of epochs (for ResNet-18 model) and the error increase from poisoning, we observe a even stronger correlation compared to the original Figure 2, which again supports the claim that training the deep architecture for more epochs seems to be one possible approach to improve resistance to poisoning. The slight increase in correlation in the new experiment results from the slightly higher covariance between the error increase and the number of epochs, but this difference is not significant since we only consider a limited number of samples when computing the correlation. The comparison results are summarized below:
> > > > > > >
> > > > > > > | Figure 2 | Epoch Number vs Increased Error |
> > > > > > > |:--- | :----:        |
> > > > > > > | Original  | -0.95      |
> > > > > > > | New  | -0.99 |

---

> > > > > > > > ### Author Response · Authors · 2023-08-17
> > > > > > > > **Continued follow-up discussion on new results**
> > > > > > > >
> > > > > > > > In summary, we believe the new results of Figure 2 still provide strong empirical evidence to support our main claim and suggest a similar future direction from the perspective of learning better feature representations to build better defenses against indiscriminate poisoning.
> > > > > > > >
> > > > > > > > Nevertheless, depending on different application scenarios (e.g., how much downstream data are used) and how the feature extractor is learned, the two factors may not consistently change in the same direction, as reflected in the new experiment. Therefore, developing general methods for learning meaningful feature extractors such that the two factors can be improved simultaneously in a controlled way would be an interesting research direction, which we plan to pursue in the future. However, we also note that this is a challenging task from the practical perspective since the goal is to extract useful feature representations for downstream applications, so the base error should also be considered in addition to error increase from poisoning, especially when no downstream data are used. Our initial thought is to consider an intermediate few-shot learning setup, where a small number of (clean) downstream training data are allowed to be incorporated in training the feature extractor. This few-shot learning setup might correspond to an intermediate setting between the original experiment with full knowledge of (clean) downstream training data and the new experiment with almost zero knowledge but still reflects certain real-world application scenarios.
> > > > > > > >
> > > > > > > > Thanks for your valuable comment. We will carefully discuss the implications with the computed quantitative metrics and also revise the statement in the original paper correspondingly.

---

### Official Review · Reviewer_Ya9d · 2023-06-26

**Soundness:** 2 fair
**Presentation:** 3 good
**Contribution:** 2 fair
**Rating:** 6
**Confidence:** 3

**Summary:**

This paper delves into the impact and effectiveness of indiscriminate data poisoning attacks on machine learning models, particularly linear ones. It highlights significant variations in attack effectiveness across different datasets and seeks to understand whether this is due to inherent dataset robustness or shortcomings in current attack strategies. The paper presents optimal poisoning definitions, characterizes these attacks under a theoretical Gaussian mixture model, and identifies learning task properties that affect attack performance. It concludes with suggestions for designing better defenses against such attacks.

**Strengths:**

- The question the paper is trying to answer, of whether some datasets are inherently robust to poisoning attacks or just resilient to state-of-the-art attacks, is original and interesting.
- The paper flows smoothly and is decently written, but contains some grammatical mistakes and typos (some of which I mention below). A careful proofread can increase the quality of writing quite a bit.
- The paper is sound, and the claims are proved on 1D toy Gaussian mixture model then extended to more complex data distributions.
- The empirical results support the theoretical results. Specifically, on various datasets, the results show that the datasets that are less vulnerable to poisoning have higher values for metrics defined by the paper (Sep/SD and Sep/Size).


**Weaknesses:**

- The contributions of the paper is not very significant as there is no clear practical implication of the findings of the paper. However, the paper investigates an interesting phenomenon of "inherent robustness" of some datasets which might inspire further investigations of this sort.
- The paper analysis is restricted to linear models.

**Questions:**

- The format of figure out is confusing.
  - Are the bars stacked?
  - Caption mentions “Datasets are sorted from lowest to highest base error rate”. Does this mean the error rate of the SVM trained on $S_c$?
  - Please change the color palette to more distinguishable colors.

- The paper analysis is restricted to linear models. Would any of these insights hold for more complex models? Did the authors do any (empirical) investigation there? In particular, it would be interesting if the authors find that high Sep/SD and Sep/Size metrics are indicative of lower vulnerability to poisoning in more general neural networks and on datasets such as CIFAR-10 and ImageNet.

- Are the percentage changes in the results paragraph in Line 157 absolute or relative? I believe it is absolute, but would be helpful to clarify that.

### Minor

- Line 123: What is a “theoretic game”?
- Line 278: an higher-> a higher


**Limitations:**

Yes.

---

> ### Author Rebuttal · Authors · 2023-08-09
>
> 1. **No clear practical implications from this paper**
>
> We appreciate that the reviewer agrees that our work might inspire the future investigation on the inherent robustness of datasets. As for the possible practical future implications, we present some preliminary results in the discussion section (Section 7) with a hope that our results may provide additional insights on building robust systems against poisoning attacks in the future.
>
> 2. **Analysis is only limited to linear models. Are there any empirical investigations on neural networks?**
>
> We do provide the preliminary results on neural networks (trained on the full MNIST and CIFAR10) in Appendix F and put a pointer to the appendix in the text right below Definition 5.6 in page 7. Our analysis show that, when the victim uses similar architectures for different datasets, the insights on linear models might also generalize, but more in-depth investigation is needed to make some definitive claims. However, this investigation is out of the scope of this paper and is an interesting next step.
>
> We also note that, although our analysis is limited to linear models, we think it is valuable to investigate linear models (as stated in the introduction) and much can still be learned from analyzing the simpler models. For example, attacks on linear models [1] can inspire the design of attacks on deep neural networks given proper adaptation and invention [2].
>
> [1] Koh et al., "Stronger Data Poisoning Attacks Break Data Sanitization Defenses", Machine Learning 2022.
>
> [2] Lu et al., "Exploring the Limits of Model-Targeted Indiscriminate Data Poisoning Attacks", ICML 2023.
>
> 3. **Other issues such as format of Figure 1 and absolute error increase**
>
> For Figure 1, yes, the bars were originally stacked with a hope to concisely represent all the information in one figure. Even though stacked, the height of each bar should be interpreted as the absolute value of the final errors (corresponding to absolute errors trained on $\mathcal{S}_c$ or $\mathcal{S}_c\cup\mathcal{S}_p$), not increased errors. The datasets are sorted based on the base error trained on $\mathcal{S}_c$ and because the base errors of LR and SVM have the same orders for the tested datasets, the potential ambiguity between two models is eliminated. We will clarify this in the figure caption. Further, we will divide Figure 1 into two figures that individually show the results of SVM and LR, and use two bars to show the errors of the models trained on $\mathcal{S}_c$ and $\mathcal{S}_c\cup \mathcal{S}_p$ respectively, and will choose the colors more wisely to make the bars more distinguishable.
>
> "Theoretic game" was intended to mean "game theoretic formulation", and we will simply name it as "game". The percentage changes in line 157 is the absolute change and we will add clarification to it.

---

### Official Review · Reviewer_J4a4 · 2023-07-06

**Soundness:** 4 excellent
**Presentation:** 4 excellent
**Contribution:** 4 excellent
**Rating:** 7
**Confidence:** 5

**Summary:**

Inspired by the variant performance of indiscriminate data poisoning attacks against different datasets, this paper studies the inherent robustness of datasets against them. Using an optimal attack designed for a binary Gaussian dataset, the authors discover three properties, namely class separability, variation, and constraint size are important factors to determine the robustness of a dataset. Experiments further validate the theory on several datasets.

**Strengths:**

(1) This paper studies an important problem, namely the dataset robustness against indiscriminate data poisoning attacks.

(2) The authors provide initial but valuable evidence on understanding the vulnerability of datasets by explicating the three properties. Although these characteristics match our intuitive understanding of the problem, they have not been systematically analyzed or studied.

(3) The empirical findings, especially the results in Section D, are nicely presented and very comprehensive. Moreover, I can find answers to most of my initial questions when reading the main paper in the appendix.

**Weaknesses:**

(1) I don't find the title a nice summarization of the entire paper as it is probably too general. I would suggest mentioning the dataset robustness in the title.

(2) I don't think comparing with LDC is necessary as I clearly see the drastic difference. I suggest the authors move this part to the appendix and present some more valuable content in the main paper, e.g., some experiments in Appendix D.

(3) Although it has been mentioned in the appendix, I think it is important to mention what attacks you use to generate Table 1. The authors should at least provide a pointer to the appendix.


**Questions:**

Questions are listed above.

**Limitations:**

Limitations are well addressed.

---

> ### Author Rebuttal · Authors · 2023-08-09
>
> We appreciate the reviewer for acknowledging our work for providing the initial but valuable step towards understanding the impact of distributional properties on robustness of linear learners against poisoning attacks. Please see our individual responses below.
>
> 1. **Set title to be more focused**
>
> Thanks for the suggestion and we are considering to change the title as *When can Linear Learners be Robust to Indiscriminate Poisoning: A Distributional Perspective*. We are inclined towards discussing the robustness with respect to distributions, instead of datasets, as distributional properties perhaps denote the underlying reasons that contribute to the dataset robustness in practice.
>
> 2. **Shrink the content on LDC and discuss more on experimental results**
>
> We will move part of our results in Section D of the appendix to the main paper in the revised version.
>
> 3. **What attacks are used to generate table 1**
>
> The poisoned error in Table 1 denotes the highest poisoned error returned from the four empirical attacks mentioned in the experimental setup in Section 3. We will clarify this part and add a pointer to the appendix in the main paper.

---

### Official Review · Reviewer_5jje · 2023-07-07

**Soundness:** 3 good
**Presentation:** 4 excellent
**Contribution:** 3 good
**Rating:** 6
**Confidence:** 3

**Summary:**

This paper explores the issue of data poisoning attacks on machine learning models, focusing on linear models. The authors find that the effectiveness of state-of-the-art poisoning strategies varies across different datasets. They introduce definitions of optimal poisoning attacks for finite-sample and distributional settings and prove performance convergence under certain conditions. They also characterize optimal poisoning attacks under a theoretical Gaussian mixture model and find that a larger projected constraint size is linked with higher vulnerability, while larger separability and smaller standard deviation provide less vulnerability. These findings explain the performance differences of indiscriminate poisoning attacks on linear models across various datasets and suggest ways to improve robustness against poisoning.

**Strengths:**

- The observation that the performance of state-of-the-art poisoning strategies significantly varies across different datasets is indeed insightful. It not only challenges the common understanding of indiscriminate poisoning attacks but also opens up new lines of inquiry into why these differences occur.
- The theoretical analysis under specific data distributions is a major strength of this paper. It's robust, reliable, and underpins a comprehensive understanding of how and why different datasets might be more or less vulnerable to poisoning attacks. By tying the theoretical analysis back to empirical attack performance, the authors effectively bridge the gap between theory and practice, enhancing the practical relevance of their findings.

**Weaknesses:**

- It's not explicitly clear from the paper whether the observation that "linear learners on some datasets are able to resist the best-known attacks even without any defenses" is novel or previously reported. The authors could benefit from addressing this explicitly.

- The authors' focus on linear classifiers is indeed a limitation. While the paper provides valuable insights into data poisoning attacks on linear models, these findings cannot be directly extrapolated to non-linear classifiers such as neural networks. Moreover, the paper does not provide enough information to definitively conclude whether linear classifiers are more robust than non-linear classifiers without additional defense strategies.

-  The paper doesn't sufficiently distinguish its analysis from existing works on data poisoning that also provided theoretical analysis on Gaussian mixture distributions and linear classifiers, such as [1]. Could you elaborate on this?



[1] Tao, Lue, et al. "Can Adversarial Training Be Manipulated By Non-Robust Features?." Advances in Neural Information Processing Systems 35 (2022): 26504-26518.

**Questions:**

- Is the resistance of some linear learners to attacks, even without defenses, a new finding or reported in previous studies?

- How applicable are these findings to non-linear classifiers like neural networks? Do they imply greater inherent robustness of linear classifiers against attacks?

**Limitations:**

In summary, this paper makes a substantial contribution to the field. Although its primary limitation is its exclusive focus on linear classifiers, it effectively elucidates the observed phenomena within this scope.

---

> ### Author Rebuttal · Authors · 2023-08-09
>
> 1. **Is the observation of "linear learners being robust to poisoning on some datasets" a novel observation or previously reported?**
>
> Yes, we are the first to explicitly report the results that linear models can resist the current state-of-the-art poisoning attacks on some datasets by running the state-of-the-art attacks on various benchmark datasets without considering additional defenses. In contrast, previous literature did not clearly show the robustness of linear models to poisoning without defenses.
>
> However, we note that the disparate effectiveness of current indiscriminate poisoning attacks across different datasets (in settings different from ours) is reported in prior work. For example, Steinhardt et al. [1] observed that datasets such as MNIST 1-7, when equipped with an oracle defense, can be highly robust to state-of-the-art poisoning attacks while the Enron dataset is highly vulnerable even with defenses. Recently, Lu et al. also observed that attack effectiveness might vary across datasets for neural networks in two recent works [2], [3]. However, these works either consider additional defenses and obfuscate the impact of original distributional properties on poisoning effectiveness [1], or the empirical observations are for neural networks, which are also found to be harder to poison than the linear models [3].
>
> 2. **How applicable are these findings to non-linear classifiers such as neural networks? Do they imply greater or lesser inherent robustness to attacks compared to linear models?**
>
> We merged the related questions mentioned in the "Weakness" and "Questions" sections to provide a more focused response.
>
> We provide the initial thoughts on extending the analysis on linear models to neural networks and some preliminary experiments in Appendix F (with a pointer to the appendix in page 7 in the main paper, right below Definition 5.6). Although our results show that the insights on linear models may also be relevant when the victim uses similar network architectures for different datasets (e.g., MNIST, CIFAR10), this conclusion still requires more in-depth investigation, which is beyond the scope of this paper.
>
> As for the robustness comparison between linear and non-linear models such as neural networks, we first clarify that our paper is not to make a claim that linear models are more robust than non-linear models such as neural networks without defenses. We only study the impact of distributional properties on poisoning effectiveness for linear models. In fact, prior literature actually shows the opposite that non-linear models such as neural networks are (currently) harder to poison than linear models. This is backed up by the fact that, indiscriminate attacks on neural networks are not very successful and some good progress is made only very recently [2], [3]. However, even with the state-of-the-art attack on neural networks, for the same dataset, neural networks are still harder to poison than the linear models. For details, please see the attack results on LR and NN that are trained on the full MNIST dataset at 3% poisoning ratio in Table 2 in Lu et al. [3], where the increased error of LR is around 23\% while the error increase of NN is only around 6\%. We will add a clarifying sentence in the revised version to avoid confusion.
>
>
> 3. **The distinction to the work by Tao et al.**
>
> We appreciate the reviewer for bringing this paper to our attention. We also discover another work on poisoning attacks that conducts rigorous analysis on Gaussian mixture models and then empirically validates the idea on benchmark datasets [4].
>
> Compared to the paper pointed out by the reviewer, the major distinction is, our work considers a threat model where the attacker generates poisoning points to impact the clean accuracy of the poisoned model that is obtained from natural (or standard) training, while the mentioned work considers the generation of poisoning points to compromise the robust accuracy of the model obtained through adversarial training, and the attack did not focus on reducing the model accuracy on clean test samples. Different from us, the work by Tao et al. [4] focuses on demonstrating how adversarial training can help defend against data poisoning attacks.
>
> Another distinction of our work from the two reference papers is, the reference papers consider the poisoning setup where the feature perturbations have bounded norms and the labels are still correct (i.e., the poisoned data points are still semantically meaningful), and the entire dataset can be modified by the attackers. However, our poisoning setup only allows attackers to inject a small fraction of poisoning points and the generated poisoning points do not have to be obtained by adding bounded perturbations to existing data points and their labels can also be freely assigned, as long as the generated poisoning points fall into the constraint set $\mathcal{C}$ defined in the paper. We will properly discuss the mentioned papers in our revised version.
>
> 4. **Focus on linear models is indeed a limitation**
>
> We think studying linear models is still valuable (as argued in the introduction) and much can still be learned from analyzing these simple models. For example, attacks on the convex (e.g., linear) models [5] can also inspire the design of attacks on non-convex models given some proper adaptation and invention [3].
>
> [1] Steinhardt et al., "Certified defenses against poisoning attacks", N(eur)IPS 2017.
>
> [2] Lu et al., "Indiscriminate Data Poisoning Attacks on Neural Networks", TMLR 2022.
>
> [3] Lu et al., "Exploring the Limits of Model-Targeted Indiscriminate Data Poisoning Attacks", ICML 2023.
>
> [4] Tao et al., "Better Safe Than Sorry: Preventing Delusive Adversaries with Adversarial Training", NeurIPS 2021.
>
> [5] Koh et al., "Stronger Data Poisoning Attacks Break Data Sanitization Defenses", Machine Learning 2022.

---

### Author Rebuttal · Authors · 2023-08-09

We continue to respond to reviewer oRg1 in the global response due to space constraints and also attach the new result tables as a single PDF. Comments from other reviewers are addressed in the individual rebuttals.

2. **For results on data sanitization defenses, to justify "mostly by ...", a lot more is needed. Also, Why does filtering only affect the projected constraint size?**

In the original submission, we meant to use "mostly" to indicate that the projected constraint size (Size or Sep/Size) is impacted more significantly compared to the separability ratio Sep/SD, but realized that this wording is misleading and may even give an impression of causation-like claims. We included detailed results in Table 3 in the attached PDF file (can also be found in Table 4 of Appendix D in the original submission), we can see that after filtering, the Sep/SD does not change much while the Sep/Size (or Size) is changed significantly. This also answers whether data filtering is going to impact both factors. At least for the tested datasets, the Sep/SD is minimally impacted while there is a significant impact on the Sep/Size.

We will revise our statement to indicate that the reduction in attack effectiveness is correlated with the significant reduction in the Size, and this high correlation provides possible explanations on why data sanitization defenses work, to avoid the issue of (strong) causation-like claims.

3. **Experiments on better feature representation do not correspond to the practical scenario of ...**

We are not absolutely sure if we get the reviewer's comment correctly. We will clarify based on our current understanding, so please correct us if we misunderstood your comment.

First, we clarify that we provide this transfer learning experiment mainly for the purpose of showing a potential implication of our work: it is possible for the victim to select a better feature extractor (assuming it is free from poisoning) to improve the resilience of the downstream classifiers to indiscriminate attacks.

The reviewer is correct that the whole CIFAR-10 dataset is used to train different feature encoders in this experiment as an illustration. In practice, the victim can go beyond this experimental setup to leverage any pretrained feature extractor (free from poisoning), which we believe is one typical transfer learning setting. We only use the CIFAR-10 dataset in this experiment instead of higher-resolution image datasets (e.g., ImageNet), just for the convenience in obtaining different feature extractors by training from scratch. In addition, we assume the victim has access to a small number of clean samples to set the box-constraints for the feature representation space. These clean samples are neither related to the pretrained feature extractor nor used to train the downstream classifier. While assuming a pretrained encoder free from poisoning may be strong for many practical applications, there are situations where there can be a one-time centralized effort to clean a large benchmark dataset to train some high-quality feature extractors free from poisoning and enable diverse downstream applications. Again, we want to emphasize that the presented results serve to point out potential implications of our work on building better defenses. It is an interesting future work to relax these assumptions to build more practical defenses for real-world scenarios.

4. **About Figure 1**

Please see the response to reviewer Ya9d for the revision plans. The poisoned error is the highest poisoned error among the errors from the four tested attacks. We did not include the results of multiple attacks because we are only concerned about the best-performing attack. The performances of individual attacks are given in Figure 4 in Appendix D, where they are similar in many cases.

Below are the remaining major concerns in the **Questions** section.

**Why same variance in Theorem 5.3?**

The same variance is assumed simply for convenience in the proof without involving minor but tedious calculations. Similar conclusions can be obtained using different variances.

**How is Filtered Enron obtained? Comparison to the Enron dataset?**

The Filtered Enron is obtained by first training a linear SVM on the original Enron dataset and then removing 3\% of points (in both the training and test data) that are closest to the boundary. As mentioned in Section 6, the purpose of this dataset (together with Adult) is to show one cannot trivially infer the vulnerability of a dataset to poisoning by checking its base error without poisoning. As shown in Figure 1 and Table 1 in the paper, both the Filtered Enron and the Enron are highly vulnerable, but the Filtered Enron has the lowest base error among all the tested datasets.

**Is hinge loss specifically required?**

Thanks for raising this question. Indeed, some of our statements are not very clear, so we provide clarifications below and will revise accordingly.

For all the theoretical results, we require the loss function for model training to be strongly convex, since we make use of the fact that the minimizer of a strongly convex loss function is unique to prove Theorems 4.3 and 4.5. Such a strong convexity condition is also used in Section 5, because we use Theorem 4.5 to justify the choice of the maximum poisoning budget. So, our results can be applied to any regularizer that can ensure the strong convexity of the loss function, including the typical $\ell_2$-regularization studied in this work and more broadly, the elastic net regularization.

The results in Section 4 apply to general convex loss (with the regularizer, the whole loss is strongly convex). In Section 5.1, we study hinge loss in particular for the convenience of mathematical derivations but believe similar results can be proven for other margin-based convex losses such as logistic loss (but maybe more tedious). In Section 5.2, we study any margin-based convex losses, not just the hinge loss.

---

### Author Response · Authors · 2023-08-20
**Thanks for your constructive feedback**

We greatly appreciate the thoughtful comments from the reviewers that have helped improve our paper. We will reflect on them carefully in the revised version of our paper. If there are any remaining questions or concerns, we are more than happy to address them before the rebuttal deadline.

---

### Decision · Program_Chairs · 2023-09-21

**Decision:**

Accept (poster)

**Comment:**

Most reviewers were generally happy with this paper from the start, only Reviewer oRg1 was unconvinced. Through a significant back and forth, the reviewer and the authors settled their disagreements. The reviewer was pleased with the engagement from the authors, and trusts them to make the changes that they promised to all the reviewers. In my own evaluation, I tend to agree -- that trying to understand which datasets are or aren't poisonable is a very interesting direction, and the authors take a promising step in this direction. Thus, I recommend the paper for acceptance.